# Spurious Local Minima Provably Exist for Deep Convolutional Neural Networks

## Abstract

In this paper, we prove that a general family of infinitely many spurious local minima exist in the loss landscape of deep convolutional neural networks with squared loss or cross-entropy loss. Our construction of spurious local minima is general and applies to practical dataset and CNNs containing two consecutive convolutional layers. We develop some new techniques to solve the challenges in construction caused by convolutional layers. We solve a combinatorial problem to show that a differentiation of data samples is always possible somewhere in feature maps. Empirical risk is then decreased by perturbation of network parameters that can affect different samples in different ways. Despite filters and biases are tied in each feature map, in our construction this perturbation only affects the output of a single ReLU neuron. We also give an example of nontrivial spurious local minimum in which different activation patterns of samples are explicitly constructed. Experimental results verify our theoretical findings.

## 1 Introduction

Convolutional neural networks (CNNs) (e.g. Lecun et al. (1998); Krizhevsky et al. (2012); Simonyan & Zisserman (2015); Szegedy et al. (2015); He et al. (2016); Huang et al. (2017)), one of the most important models in deep learning, have been successfully applied to many domains. Spurious local minima, whose losses are greater than that of global minimum, play an important role in the training of deep CNNs and understanding of deep learning models. It is widely believed that spurious local minima exist in the loss landscape of CNNs which is thought to be highly non-convex, as evidenced by some experimental studies (e.g. Dauphin et al. (2014); Goodfellow et al. (2015); Liao & Poggio (2017); Freeman & Bruna (2017); Draxler et al. (2018); Garipov et al. (2018); Li et al. (2018); Mehmeti-Gopel et al. (2021)). However, the existence of spurious local minima for deep CNNs caused by convolutions has never been proved mathematically before.

In this paper, we prove that infinite spurious local minima exist in the loss landscape of deep CNNs with squared loss or cross-entropy loss. This is in contrast to the "no spurious local minima" property of deep linear networks. The construction of spurious local minima in this paper is general and applies to practical dataset and CNNs containing two consecutive convolutional layers, which is satisfied by popular CNN architectures.

The idea is to construct a local minimum $\theta$ at first, and then construct another point $\theta'$ in parameter space which has the same empirical risk as $\theta$ and there exist regions around $\theta'$ with less empirical risks. However, the construction of spurious local minima for CNNs faces some technical challenges, and the construction for fully connected deep networks cannot be directly extended to CNNs. Our main contribution in this paper is to tackle these technical challenges. In the construction of spurious local minima for fully connected deep ReLU networks (He et al. (2020); Ding et al. (2019); Goldblum et al. (2020); Liu et al. (2021)), in order to construct $\theta'$ and perturb around it, data samples are split into some groups according to the inputs of a specific ReLU neuron such that each group will behave differently under the perturbation of network parameters so as to produce a lower risk. This technique relies on data split and parameter perturbation, and cannot be directly applied to CNNs due to the following difficulties. Every neuron in CNN feature maps has limited receptive field that covers partial pixels in an input image (take image as an example), and hence the inputs to a ReLU neuron can be identical even for distinct samples, making them hard to distinguish. This data split issue is further complicated by the nonlinear ReLU activations that truncate negative inputs, and the

activation status can vary from place to place and from sample to sample. Moreover, the filters and biases of CNNs are shared by all neurons in the same feature map, and thus adjusting the output of a ReLU neuron by perturbing these tied parameters will also affect other neurons in the same feature map.

We solve these challenges by developing some new techniques in this paper. By taking limited receptive fields and possible distinct activation status for different locations and samples into account, we solve a combinatorial problem to show that a split of data samples is always possible somewhere in feature maps. We then present a construction of CNN parameters ($\theta'$) that can be perturbed to achieve lower losses for general local minima $\theta$. Despite the parameters are tied, our construction can perturb the outputs of samples at a single neuron in feature maps without affecting other locations. We also give a concrete example of spurious local minima for CNNs. To our best knowledge, this is the first work showing existence of spurious local minima in deep CNNs introduced by convolutional layers.

This paper is organized as follows. Section 1.1 is related work. Section 2 describes convolutional neural networks, and gives some notations used in this paper. In section 3, our general results on spurious local minima are given with some discussions. In section 4, we present an example of nontrivial spurious local minima for CNNs. Section 5 presents experimental results to verify our theoretical findings. Finally, conclusions are provided. More lemmas, experimental details and all proofs are given in appendices.

## 1.1 RELATED WORK

For some neural networks and learning models, it has been shown that there exist no spurious local minima. These models include deep linear networks ( Baldi & Hornik (1989); Kawaguchi (2016); Lu & Kawaguchi (2017); Laurent & von Brecht (2018); Yun et al. (2018); Nouiehed & Razaviyayn (2018); Zhang (2019)), matrix completion and tensor decomposition (e.g., Ge et al. (2016)), one-hidden-layer networks with quadratic activation (Soltanolkotabi et al. (2019); Du & Lee (2018)), deep linear residual networks (Hardt & Ma (2017)) and deep quadratic networks (Kazemipour et al. (2020)).

Existence of spurious local minima for one-hidden-layer ReLU networks has been demonstrated, by constructing examples of networks and data samples, in Safran & Shamir (2018); Swirszcz et al. (2016); Zhou & Liang (2018); Yun et al. (2019); Ding et al. (2019); Sharifnassab et al. (2020); He et al. (2020); Goldblum et al. (2020) etc.

For deep ReLU networks, He et al. (2020); Ding et al. (2019); Goldblum et al. (2020); Liu et al. (2021) showed that spurious local minima exist for fully connected deep neural networks with some general loss functions. For these spurious local minima, all ReLU neurons are active and deep neural networks are reduced to linear predictors. Spurious local minima for CNNs are not treated in these works. In comparison, we deal with spurious local minima for CNNs in this work, and the constructed spurious local minima can be nontrivial in which nonlinear predictors are generated and some ReLU neurons are inactive.

Du et al. (2018); Zhou et al. (2019); Brutzkus & Globerson (2017) showed the existence of spurious local minima for one-hidden-layer CNNs with a single non-overlapping filter, Gaussian input and squared loss. In contrast, we discuss practical deep CNNs with multiple filters of overlapping receptive fields and arbitrary input, for both squared and cross-entropy loss. Given non-overlapping filter and Gaussian input, the population risk with squared loss function can be formulated analytically with respect to the single filter $\mathbf{w}$, which facilitates the analysis of loss landscape. Thus, the techniques used in Du et al. (2018); Zhou et al. (2019); Brutzkus & Globerson (2017) cannot be extended to the general case of empirical risk with arbitrary input samples discussed in this paper. Nguyen & Hein (2018) showed that a sufficiently wide CNN (include a wide layer which has more neurons than the number of training samples followed by a fully connected layer) has a well-behaved loss surface with almost no bad local minima. Liu (2022) explored spurious local minima for CNNs introduced by fully connected layers. Du et al. (2019); Allen-Zhu et al. (2019) explored the local convergence of gradient descent for sufficiently over-parameterized deep networks including CNNs.

## 2 PRELIMINARIES

### 2.1 NOTATIONS

We use $M_{i,\cdot}$ and $M(i,\cdot)$ to denote the $i$th row of a matrix $M$, and use $M_{\cdot,i}$ and $M(\cdot,i)$ to denote the $i$th column of $M$. $M_{i,j}$ and $M_{ij}$ are the $(i,j)$ entry of $M$. The $i$th component of a vector $\mathbf{v}$ is denoted as $v_i$, $v^i$ or $v(i)$. $[N]$ is equivalent to $\{1, 2, \cdots, N\}$. $i : j$ means $i, i+1, \cdots, j$. $v(i : end)$ stands for the components of a vector $\mathbf{v}$ from the $i$th one to the last one. $\mathbf{1}_n$ denotes a vector of size $n$ whose components are all 1s.

### 2.2 CONVOLUTIONAL NEURAL NETWORKS

A typical CNN includes some convolutional layers, pooling layers and fully connected layers. Because spurious local minima caused by fully connected layers in CNNs can be treated in the same way as in fully connected networks, we will focus on spurious local minima caused by convolutional layers.

Convolutional layers can take advantage of the translational invariance inherent in data, and are defined as follows. Suppose the $l$th layer is a convolutional layer, neighboring neurons at layer $(l-1)$ are grouped into patches for the convolution operation. Let $P_l$ and $s_l$ be, respectively, the number of patches and the size of each patch in the $l$th layer, and denote by $T_l$ the number of convolutional filters (or the number of feature maps) in the $l$th layer. Denote $\left\{ \mathbf{o}_1^l, \cdots, \mathbf{o}_{P_l}^l \right\} \in \mathbb{R}^{s_l}$ as the set of patches in the $l$th layer. If there are multiple feature maps in layer $(l-1)$, a patch will include corresponding neighboring neurons from every feature map. The number of neurons in each feature map in the $l$th layer is $P_{l-1}$. The number of neurons in the $l$th layer is then $n_l = T_l P_{l-1}$. Given the $p$th ($p \in [P_{l-1}]$) patch $\mathbf{o}_p^{l-1}$ and the $t$th ($t \in [T_l]$) filter $\mathbf{w}_t^l \in \mathbb{R}^{s_{l-1}}$, the output of corresponding neuron in layer $l$ is given as follows,

$$o^l(h) = \sigma(\mathbf{w}_t^l \cdot \mathbf{o}_p^{l-1} + b_t^l), \tag{1}$$

where $h = (t-1)P_{l-1} + p$. $\sigma(x) = max(0, x)$ is the ReLU activation function. There is a single bias $b_t^l$ for each feature map.

For ease of presentation, we will assume that the strides for convolutions are equal to one and the input feature maps are zero padded such that for each convolutional layer, the sizes of input and output feature maps are equal. The convolution operation can then be reformulated as a matrix-vector product. We explain this by considering the following one-dimensional example. Given a one-dimensional input $\mathbf{o}^{l-1} = (a, b, c, d, e)^T$ and a filter $\mathbf{w}_1 = (w_1^1, w_1^2, w_1^3)^T$, the input becomes $(0, a, b, c, d, e, 0)^T$ after padding with zeros. Here, $T_l = 1$, $s_{l-1} = 3$, $P_{l-1} = 5$ after zero-padding, $n_l = T_l P_{l-1} = 5$. The convolution operation is equivalent to

$$\mathbf{o}^l = \sigma \left[ \begin{pmatrix} w_1^2 & w_1^3 & 0 & 0 & 0 \\ w_1^1 & w_1^2 & w_1^3 & 0 & 0 \\ 0 & w_1^1 & w_1^2 & w_1^3 & 0 \\ 0 & 0 & w_1^1 & w_1^2 & w_1^3 \\ 0 & 0 & 0 & w_1^1 & w_1^2 \end{pmatrix} \begin{pmatrix} a \\ b \\ c \\ d \\ e \end{pmatrix} + \begin{pmatrix} b_1^l \\ b_1^l \\ b_1^l \\ b_1^l \\ b_1^l \end{pmatrix} \right]. \tag{2}$$

One can set $(w_1^1 = 0, w_1^2 = 1, w_1^3 = 0)$ in 2 to forward $\mathbf{o}^{l-1}$ unchanged (before adding bias and ReLU activation). This is equivalent to setting the weight matrix as a identity one. For the case of two-dimensional input, the input can be converted into a vector and the convolution operation can be reformulated as a matrix-vector product using similar idea.

We will use matrix-vector product to represent convolution operations. Denote the weight matrix and bias vector of the $l$th layer as $W^l$ and $\mathbf{b}^l$, respectively, with $W^l \in \mathbb{R}^{n_l \times n_{l-1}}$ and $\mathbf{b}^l \in \mathbb{R}^{n_l}$, then the output of the $l$th layer is $\mathbf{o}^l = \sigma(W^l \mathbf{o}^{l-1} + \mathbf{b}^l)$. This matrix-vector product formulation can also be used to represent fully connected layers.

We only consider average pooling layers. If layer $l$ is a average pooling layer, each patch used in the pooling operation will include only the neighboring neurons within a single feature map in layer $(l-1)$. Let $P_{l-1}$ be the number of patches in a single feature map of the $(l-1)$th layer, and hence the size of each feature map in layer $l$ is $P_{l-1}$. For average pooling, the output of the $p$th neuron $o^l(p)$ in each feature map of layer $l$ is computed by

$$o^l(p) = \text{mean}(o_p^{l-1}(1), \cdots, o_p^{l-1}(s_{l-1})), \ p \in [P_{l-1}], \tag{3}$$

where $o_p^{l-1}(i)$ is the $i$th element in the $p$th patch of layer $(l-1)$ within a feature map. The average pooling operation is a linear operation and thus can can be represented by a matrix-vector product. The exact forms of parameter matrices for average pooling and fully connected operations will be given in Appendix A in detail.

Consider a training set $\{(\mathbf{x}_1, \mathbf{y}_1), (\mathbf{x}_2, \mathbf{y}_2), \ldots, (\mathbf{x}_N, \mathbf{y}_N)\}$, where $\mathbf{x}_i \in \mathbb{R}^{d_x}, \mathbf{y}_i \in \mathbb{R}^{d_y}$ $(i \in [N])$ are, respectively, the input and target output. Let $L$ be the number of layers in a CNN, and let $\mathbf{o}^{l,i}$ be the output $\mathbf{o}^l$ for the $i$th data sample (denote $\mathbf{o}^{0,i} := \mathbf{x}_i$). The output vector of a CNN is

$$\mathbf{o}_i := \mathbf{o}^{L,i} = W^L \mathbf{o}^{L-1,i} + \mathbf{b}^L, \ \ i \in [N].$$

We introduce a diagonal matrix $I^{l,i} \in \mathbb{R}^{n_l \times n_l}$ for each sample and each ReLU layer to represent the activation status of ReLU neurons, whose diagonal entries are $I_{k,k}^{l,i} = 1$ if $W^l(k, \cdot)\mathbf{o}^{l-1,i} + b_k^l > 0$ and $I_{k,k}^{L,i} = 0$ otherwise. Consequently, the output of a convolutional layer can be written as $\mathbf{o}^{l,i} = I^{l,i}(W^l\mathbf{o}^{l-1,i} + \mathbf{b}^l)$. The CNN output can be written as

$$\mathbf{o}_i = W^L I^{L-1,i} \left( W^{L-1} \cdots \left( W^1 \mathbf{x}_i + \mathbf{b}^1 \right) + \cdots + \mathbf{b}^{L-1} \right) + \mathbf{b}^L. \tag{4}$$

The empirical risk (training loss) is defined as

$$R(W^1, \mathbf{b}^1, \cdots, W^L, \mathbf{b}^L) = \frac{1}{N} \sum_{i=1}^N l(\mathbf{o}_i, \mathbf{y}_i), \tag{5}$$

where $l$ is the loss function, such as the widely used cross-entropy loss for classification problems. Our main goal in this paper is to construct spurious local minima for the empirical risk function of deep CNNs given in (5).

## 3 Spurious Local Minima

In this section, we first construct a general local minimum with parameters $\theta := (W^l, \mathbf{b}^l)_{l=1}^L$. Then, we present another point $\theta'$ in parameter space which has an equal empirical risk with $\theta$. We further perturb $\theta'$ and show that empirical risk can be decreased. Therefore, $\theta$ is a spurious local minimum in parameter space.

Our construction is general and can be applied to CNNs with two consecutive convolutional layers, which is easily satisfied by most practical CNN models, such as AlexNet (Krizhevsky et al. (2012)), VGG (Simonyan & Zisserman (2015)) and GoogleNet (Szegedy et al. (2015)).

### 3.1 Construction of local minima

We first give a general construction of nontrivial local minima. Given a local minimum for a subnetwork of a CNN, we show that it will induce a local minimum of the whole CNN if the subnetwork is embedded into the CNN appropriately. Local minima for subnetworks of a CNN can be constructed using any method. Our construction of spurious local minima is general and does not rely on the concrete form of local minima for subnetworks. In section 4, we will give an explicit construction of exemplar nontrivial local minima for subnetworks.

Our construction of spurious local minima is general and will utilize two consecutive convolutional layers, whereever there are in the CNN. For ease of presentation, first we will assume that the last two hidden layers before the output layer (the $L$th layer, which is fully connected) are two consecutive convolutional layers. The general case of having two consecutive convolutional layers in other places will be discussed in section 3.3.

When layers $L-1$ and $L-2$ are two consecutive convolutional layers, we will fix the parameters of the last two convolutional layers in a way such that the output $\mathbf{o}^{L-3,i}$ is passed through layers $L-2$ and $L-1$ unchanged. Two feature maps in layers $L-2$ and $L-1$, respectively, are reserved for the perturbation described later. Without loss of generality, let them be the first and second feature maps in these two layers, respectively. Fig.1(a) shows the top layers of the CNN architeture with some associated parameters. Note that the size of a single output feature map in the $l$th layer is $P_{l-1}$, and $P_{L-2} = P_{L-3} = P_{L-4}$ after padding. We have the following lemma to informally descibe the construction of local minima $\theta$, and its formal version is given Lemma 6 in Appendix C.

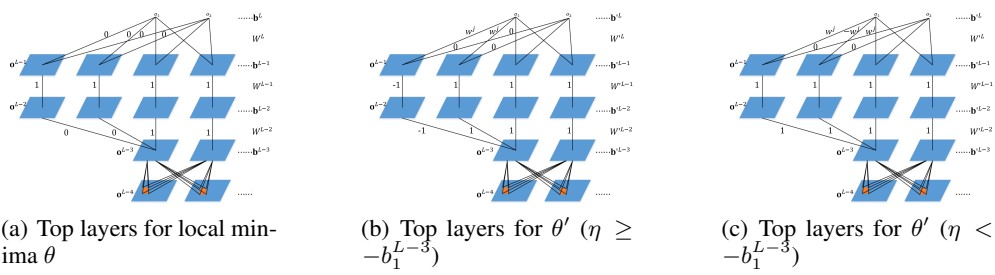

(a) Top layers for local minima $\theta$

(b) Top layers for $\theta'$ ($\eta \geq -b_1^{L-3}$)

(c) Top layers for $\theta'$ ($\eta < -b_1^{L-3}$)

Figure 1: The top layers and associated parameters for constructing local minima and spurious local minima. In general, our construction can be applied to CNNs with two consecutive convolutional layers, like layers $L-1$ and $L-2$ in this figure.

**Lemma 1.** *(informal). Given a CNN whose numbers of filters in the last two convolutional layers satisfy $T_{L-1} \geq 3$, $T_{L-2} \geq 3$, the parameters $W^L(\cdot, 1:2P_{L-2})$, $W^{L-1}, \mathbf{b}^{L-1}$ and $W^{L-2}, \mathbf{b}^{L-2}$ of the last three layers are set to propagate $\mathbf{o}^{L-3,i}$ unchanged to the output neurons, and the first and second feature maps in both layer $L-2$ and layer $L-1$ have no contributions to the final output. The remaining parameters, including $W^L(\cdot, 2P_{L-2}+1:end)$, $\mathbf{b}^L$ and $(W^l, \mathbf{b}^l)_{l=1}^{L-3}$, constitute a subnetwork, and if they locally minimize the training loss when fixing other parameters, then point $\theta := (W^l, \mathbf{b}^l)_{l=1}^{L}$ is a local minimum in parameter space.*

In this lemma, $W^L(\cdot, 1:2P_{L-2})$ means the weights of connections from output neurons to the first two feature maps in layer $L-1$. The identity forward propagations in layers $L-1$ and $L-2$ (see Fig.1(a)) can be implemented by setting corresponding submatrices in $W^{L-1}$ and $W^{L-2}$ to identity ones, which is equivalent to setting corresponding filters to the form of $(0, \cdots, 1, \cdots, 0)^T$ ( with a single nonzero entry in the middle) as explained in section 2.2, and setting biases $\mathbf{b}^{L-1}, \mathbf{b}^{L-2}$ to zeros.

The embedding scheme to construct local minima or saddle points has also been used for fully connected networks by Fukumizu et al. (2019); Fukumizu & Amari (2000); He et al. (2020); Liu et al. (2021); Zhang et al. (2022). However, embedding for CNNs needs to tackle the problems of limited receptive fields, arbitrary activation status at different locations, and tied weights and biases.

### 3.2 CONSTRUCTION OF SPURIOUS LOCAL MINIMA

Our construction of spurious local minima relies on the following assumption.

**Assumption 1**. At local minimum $\theta$, the inputs to the final fully connected layer are distinct for different samples, i.e., $\forall i, j \in [N]$ and $i \neq j$, $\mathbf{o}^{L-1,i} \neq \mathbf{o}^{L-1,j}$.

For practical CNNs and datasets with rich contents, the number of neurons $n_{L-1}$ input to the final fully connected layer is big and the chance of $\mathbf{o}^{L-1,i}$ (note that $\mathbf{o}^{L-1,i} \neq \mathbf{0}$ since $I^{L-1,i} \neq 0$ by the nondegenerate requirement in Lemma 6, the formal version of Lemma 1) being identical for different samples is very small, thus Assumption 1 is reasonable and enforces a very mild restriction on local minimum $\theta$. This assumption is used to prevent distinct samples from becoming indistinguishable when passing through ReLU layers and being truncated by them. For fully connected deep neural networks, (He et al. (2020); Liu et al. (2021)) assumed that all data points are distinct, i.e., $\forall i, j \in [N]$ and $i \neq j$, $\mathbf{x}_i \neq \mathbf{x}_j$. Under this condition, we show in Lemma 7 in Appendix C that for practical datasets and CNNs there always exists local minimum $\theta$ for which Assumption 1 holds. An exception is the neural collapse phenomenon that during the terminal phase of training, the features of the final hidden layer tend to collapse to class feature means. As a result, our construction of spurious local minima does not include those that may arise during the ending training phase when the loss is driving towards zero.

### 3.2.1 DATA SPLIT

In the following, we will show that given Assumption 1, there must exist at least one location in feature maps where the outputs of all samples can be split into two parts using a threshold such that the two parts will behave differently under perturbations. This is not a easy task due to the following difficulties introduced by convolutional layers. Firstly, the receptive field of each neuron in a convolutional layer is smaller than the input image, thus at each location in feature maps, the effective inputs (pixels in the receptive field) from distinct samples can be identical and indistinguishable. Secondly, the activation status of hidden ReLU neurons can vary from place to place and from sample to sample. Assumption 1 will help us to distinguish different samples. Thirdly, the filters and biases are shared by all locations in a feature map, and one cannot perturb these parameters at a location without affecting other places.

We assume that training loss at $\theta$ satisfies $R(\theta) > 0$. By Lemma 8 given in Appendix C, for training data that cannot be fit by linear models (also assumed by He et al. (2020); Liu et al. (2021)) and popular CNN architectures, there always exists point $\theta$ such that both Assumption 1 and $R(\theta) > 0$ hold. Moreover, by Lemma 11 in Appendix E, for squared and cross-entropy loss function $l$, $\frac{\partial l(\mathbf{o}(\mathbf{x}_i), \mathbf{y}_i)}{\partial \mathbf{o}}$ cannot equal to zero for all samples if $R(\theta) > 0$, where $\mathbf{o}(\mathbf{x}_i)$ is the output function of input $\mathbf{x}_i$ given by 4. Without loss of generality, assume $u_i := \frac{\partial l(\mathbf{o}(\mathbf{x}_i), \mathbf{y}_i)}{\partial o(1)} \neq 0$ for some $i \in [N]$, where $o(1)$ is the first component of output vector $\mathbf{o}$. By optimality of $\theta := (W^l, \mathbf{b}^l)_{l=1}^L$, we have $\frac{\partial R}{\partial b_1^L} = \frac{1}{N} \sum_{i=1}^N \frac{\partial l(\mathbf{o}(\mathbf{x}_i), \mathbf{y}_i)}{\partial o(1)} = \frac{1}{N} \sum_{i=1}^N u_i = 0$.

We will split the outputs of all samples at layer $L-3$. Given CNN parameters $\theta := (W^l, \mathbf{b}^l)_{l=1}^L$ and data samples $\{\mathbf{x}_1, \mathbf{x}_2, \cdots, \mathbf{x}_N\}$, we consider the outputs $o^{L-3,i}(j)$ $(i \in [N], j \in [M])$ in feature maps of layer $L-3$, where $M$ is the number of neurons (locations) in all feature maps of layer $L-3$ used in Lemma 1. For all $j \in [M]$, $i \in [N]$, we denote

$$v_i^j := W^{L-3}(j, \cdot) \mathbf{o}^{L-4,i}, \qquad w^j := W^L(1, 2P_{L-2} + j). \tag{6}$$

$v_i^j$ is the output at layer $L-3$ before adding bias and ReLU activation. Thus, at each location $j \in [M]$, there is a list $(v_1^j, v_2^j, \cdots, v_N^j)$. Let $\tilde{\mathbf{x}}_i^j$ be the vector composed of pixels of $\mathbf{x}_i$ in the receptive field of the $j$th location. $\tilde{\mathbf{x}}_i^j$ can be identical for distinct samples, resulting in identical $v_i^j$s. Identical $v_i^j$s can also be resulted from ReLU activations that remove the difference between input samples.

In general, at each location $j$ in the feature maps, $(v_1^j, v_2^j, \cdots, v_N^j)$ are organized in groups according to their magnitudes. Assume there are $g_j$ groups in the list $(v_1^j, v_2^j, \cdots, v_N^j)$. Without loss of generality, we can re-index the samples in descending order of $v_i^j$ and write the ordered list as $v_1^j = v_2^j = \cdots v_{m_1}^j > v_{m_1+1}^j = \cdots v_{m_2}^j > \cdots > v_{m_{g_j-1}+1}^j = \cdots = v_{m_{g_j}}^j = v_N^j$.

Let $\mathbf{u} = (u_1, u_2, \cdots, u_N)^T$ and note that $\mathbf{u} \neq \mathbf{0}$ and $\sum_{i=1}^N u_i = 0$. In the following lemma, we show that there is at least one location where the split of data samples exists. This is a combinatorial problem which needs to deal with multiple locations and multiple ordered groups at each location, and the groupings of samples can be arbitrary at each location.

**Lemma 2.** *Assume that training loss* $R((W^l, \mathbf{b}^l)_{l=1}^L) > 0$ *for squared or cross-entropy loss. Under Assumption 1, there exist some locations* $j \in [M]$ *where the ordered list* $v_1^j = v_2^j = \cdots v_{m_1}^j > v_{m_1+1}^j = \cdots v_{m_2}^j > \cdots > v_{m_{g_j-1}+1}^j = \cdots = v_{m_{g_j}}^j = v_N^j$ *can be splitted into two parts* $\{v_1^j, v_2^j, \cdots, v_n^j\}$ *and* $\{v_{n+1}^j, \cdots, v_N^j\}$ *by a threshold* $\eta$ *such that*

$$v_1^j, v_2^j, \cdots, v_n^j > \eta, \quad v_{n+1}^j, v_{n+2}^j, \cdots, v_N^j < \eta, \quad \sum_{i=1}^n u_i \neq 0. \tag{7}$$

There may be more than one location where such data split exists, and we choose the one with the biggest threshold $\eta$ for later usage. Let $h$ be the location with the biggest split threshold, and without loss of generality, assume it is located in the first feature map of layer $L-3$ (that is why we put a special emphasis on the first and second feature maps in layers $L-2$ and $L-1$ in Lemma 1 and

Lemma 4). Since $\eta$ for the $h$th location is the biggest threshold among all locations $j \in [M]$, by 7, $\sum_{i=1}^n u_i \neq 0$ are not satisfied at locations whose thresholds are less than $\eta$, thus we have

$$\forall j \in [M], \ j \neq h, \sum_{i \in \{1,2,\cdots,N | v_i^j > \eta\}} u_i = 0. \tag{8}$$

**An Auxiliary Lemma**  At each location $j \in [M]$, samples are organized into groups according to $v_i^j$ ($i \in [N]$). $v_i^j$s in each group are equal. Let $I_q^j$ be the set of indices of samples in the $q$th group of the $j$th location. We have the following lemma to show that $\sum_{i \in I_q^j} u_i$ cannot equal to zero for all groups and all locations.

**Lemma 3.** *Under Assumption 1, for CNNs with squared or cross-entropy loss, if $R\big((W^l, \mathbf{b}^l)_{l=1}^L\big) > 0$, then there must exist some $j \in [M]$ and $q \in [g_j]$ such that $\sum_{i \in I_q^j} u_i \neq 0$.*

Lemma 3 will be used by Lemma 2 to find data split that satisfies (7). We will prove Lemma 3 by induction. The number of groups $g_j$ at each location and the value of $v_i^j$ for each group can be arbitrary. We will prove that if Lemma 3 holds for $m$ samples, then for all possible configurations of current groups and all ways of adding a new sample, it also holds for $m + 1$ samples.

### 3.2.2 CONSTRUCTION OF $\theta'$

Given the CNN parameters $\theta := (W^l, \mathbf{b}^l)_{l=1}^L$ specified in Lemma 6 and the threshold $\eta$ in Lemma 2, we will give the point $\theta' := \big(W'^l, \mathbf{b}'^l\big)_{l=1}^L$ in parameter space and show that training loss at $\theta'$ is equal to that at $\theta$. We only change the parameters that are related with the first and second feature maps in layers $L - 2$ and $L - 1$, respectively, to obtain $\theta'$, and remaining parameters are fixed. The purpose of these parameter setting is to make $R(\theta') = R(\theta)$ and enable different behaviors under perturbation of parameters for different parts of samples. Different settings are designed depending on whether $\eta \geq -b_1^{L-3}$ or not. The setting for the case of $\eta \geq -b_1^{L-3}$ is illustrated in Fig.1(b), and the setting for $\eta < -b_1^{L-3}$ is shown in Fig.1(c). The following lemma informally describe the parameter settings and its formal version is given in Lemma 9 in Appendix C).

**Lemma 4.** *(informal). Given the CNN parameters $\theta := (W^l, \mathbf{b}^l)_{l=1}^L$ specified in Lemma 1 (formally Lemma 6 in Appendix C), let $\theta' := \big(W'^l, \mathbf{b}'^l\big)_{l=1}^L$, and set $W'^L, \mathbf{b}^L, W'^{L-1}, \mathbf{b}^{L-1}$ and $W'^{L-2}, \mathbf{b}^{L-2}$ in a way such that the value $w^j \cdot \sigma(v_i^j + b_1^{L-3}) + b_1^L$ (the original output of the first output neuron, contributed by location $j$ in the first feature map of layer $L - 3$) is not changed for every location $j$ and each sample $i \in [N]$, no matter how big $v_i^j$ is relative to $\eta$ and $b_1^{L-3}$. There are three possible paths through which a value $\sigma(v_i^j + b_1^{L-3})$ can propagate, i.e., the paths that connect the first, second, and third feature maps, respectively, in layers $L - 2$ and $L - 1$ (see Fig.1). If $\eta \geq -b_1^{L-3}$, a positive value of $\sigma(v_i^j + b_1^{L-3})$ will pass through either the first or the second path. If $\eta < -b_1^{L-3}$, a positive value of $\sigma(v_i^j + b_1^{L-3})$ will pass through all three paths and the outputs from the first and second paths counteract. The differentiation of paths will make it possible for different parts of samples to behave differently under perturbation of parameters. Remaining parameters in $\theta$ keep fixed. Then, $R(\theta') = R(\theta)$.*

### 3.2.3 PERTURBATION OF $\theta'$

Next, we will show that by perturbing $\theta'$, we can decrease the training loss. We demand that only the final output contributed by location $h$ in the first feature map of layer $L - 3$, where $v_1^j, v_2^j, \cdots, v_n^j > \eta$ and $\sum_{i=1}^n u_i \neq 0$ happens, will be affected by the perturbation so as to decrease the training loss. The perturbation is then designed such that a single bias in the path each $v_i^j$ (with $v_i^j > \eta$) passes through is perturbed. The final output is not affected by locations other than $h$ even with $v_i^j > \eta$ due to $\sum_{i=1}^n u_i = 0$. The following lemma informally describe our parameter perturbation scheme and its formal version is given in Lemma 10 in Appendix C).

**Lemma 5.** *(informal). If $\eta \geq -b_1^{L-3}$, perturb $b_2'^{L-2}$. If $\eta < -b_1^{L-3}$, perturb $b_1'^{L-2}$. Remaining parameters in $\theta'$ keep fixed. Then under Assumption 1, a training loss lower than $R(\theta')$ is obtained under this perturbation.*

### 3.3 MAIN RESULTS AND DISCUSSION

Combining Lemma 1, Lemma 4 and Lemma 5, we have the following theorem.

**Theorem 1.** *Under Assumption 1, the local minima $\theta := \left(W^l, \mathbf{b}^l\right)_{l=1}^{L}$ given in Lemma 1 are spurious if $R(\theta) > 0$ for squared or cross-entropy loss. Due to nonnegative homogeneity of ReLU activation, infinitely many spurious local minima exist by scaling the parameters of different layers in $\theta$.*

In the general case when the two consecutive convolutional layers we utilized are not located at the top and there are some convolutional, average pooling, or fully connected layers between them and the output layer, we can still construct spurious local minima using similar idea. The only difference is the setting of parameters for layers above the two consecutive convolutional layers, and we set them such that the output of the two consecutive convolutional layers are propagated unchanged (except pooling operations in possible subsequent pooling layers) to the first fully connected layer, which plays the role of layer $L$ in Lemmas 1 and 4, and then the output of the first fully connected layer is forwarded unchanged to the output neurons. For such constructed $\theta$ and $\theta'$, we can show that $\theta$ is still a local minimum, there is still $R(\theta') = R(\theta)$ and the empirical risk can be decreased by perturbing $\theta'$. The details will be given in Appendix D. The requirement of having two consecutive convolutional layers may be further relaxed, and we leave it to our future work.

**Remark 1:** Our construction of spurious local minima shows that for practical datasets and CNNs each local minimum of the subnetwork is associated with a spurious local minimum. Since the output of a CNN with ReLU activations is a piece-wise linear function, and from the perspective of fitting data samples with piece-wise linear output (Liu (2022)), the local minima of subnetworks of a CNN (and consequently the spurious local minima of the CNN) may be common due to the abundance of different fitting patterns. Furthermore, as suggested by Xiong et al. (2020), CNNs have more expressivity than fully connected NNs per parameter in terms of the number of linear regions produced. The ability of producing more linear regions implies more fitting patterns, thus we conjecture that CNNs are more likely to produce spurious local minima than fully connected NNs of the same size. We leave the exploration of these ideas to our future work.

## 4 AN EXAMPLE OF NONTRIVIAL SPURIOUS LOCAL MINIMA

In this section, we will construct an example of nontrivial local minimum in which some neurons are inactive, which can serve as the subnetwork in Lemma 1. By Theorem 1, the associated local minimum for the CNN that contains this subnetwork is spurious. In comparison, in He et al. (2020); Ding et al. (2019); Goldblum et al. (2020); Liu et al. (2021) for deep ReLU networks, spurious local minima are trivial since all ReLU neurons are active and deep neural networks are reduced to linear predictors.

The idea is to split data samples into two groups using a hyperplane that is perpendicular to an axis, and then fit each group of samples using different predictors. We design a CNN architecture with appropriate parameters to generate these two predictors, and show that perturbing parameters will increase or at least maintain the loss.

Given data samples $\{\mathbf{x}_1, \mathbf{x}_2, \cdots, \mathbf{x}_N\}$, we use a hyperplane with normal $\mathbf{p}$ to split the data samples. Choose any index $k \in [d_x]$, we set $\mathbf{p} = (0, 0, \cdots, 1, 0, \cdots, 0)^T$ where only the $k$th component of $\mathbf{p}$ is 1. Without loss of generality, assume the $k$th component of input $\mathbf{x}$ is in located in its first channel. Let $I = \left\{i | x_i^k = \max_{j \in [N]} x_j^k\right\}$, i.e., $I$ is the set of indices of samples having the largest $k$th component. Choose one element $i^*$ in set $I$, and denote by $j^*$ the index of any sample with the largest $x^k$ among $x_n^k$ $(n \in [N], n \notin I)$, then we set

$$c_p = -\frac{1}{2}\left(x_{i_*}^k + x_{j_*}^k\right). \tag{9}$$

For any point $\mathbf{x}$ on the positive side of hyperplane $\mathbf{p}$, we have

$$\sigma(\mathbf{p} \cdot \mathbf{x} + c_p) = \sigma(x^k - \frac{1}{2}(x_{i^*}^k + x_{j^*}^k)) = x^k - \frac{1}{2}(x_{i^*}^k + x_{j^*}^k) > 0. \tag{10}$$

We fit the two groups of samples in different ways using two subnetworks. Concatenating the two subnetworks, we obtain a full CNN. The following theorem gives an informal description of its architecture and parameters. The formal description will be given in Appendix B.

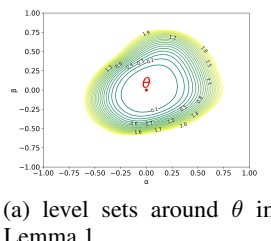

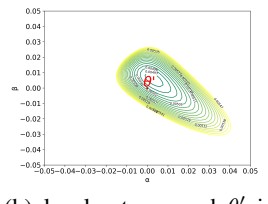

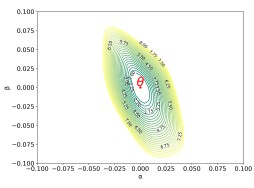

(a) level sets around $\theta$ in Lemma 1

(b) level sets around $\theta'$ in Lemma 3

(c) level sets around $\theta_1$ in Theorem 2

Figure 2: The visualization of level sets of empirical loss around specific locations in parameter space. (a) and (b) demonstrate that $\theta$ is a spurious local minimum since $R(\theta') = R(\theta)$. (c) shows that $\theta_1$ constructed in Theorem 2 is a local minimum.

**Theorem 2.** *(informal). Given a CNN with two subnetworks, its parameters $\theta_1 = \left(W^l, \mathbf{b}^l\right)_{l=1}^L$ are set as follows. The parameters of the first subnetwork are set such that the $k$th component of sample $\mathbf{x}_i$ ($i \in I$) is forward propagated to the final fully connected layer, and that of sample $\mathbf{x}_i$ ($i \notin I$) is blocked. The parameters of the second subnetwork are optimized to minimize the empirical loss for samples $\{\mathbf{x}_j | j \in [N], j \notin I\}$. Then, the point $\theta_1$ is a local minimum in parameter space.*

For this local minimum, the ReLU units in the first subnetwork that correspond to the $k$th component of input are only activated by samples $\mathbf{x}_j$ ($j \in I$) and are inactive for remaining samples. Thus, the local minimum $\left(W^l, \mathbf{b}^l\right)_{l=1}^L$ is nontrivial for which different activation patterns exist and the resulting predictor is nonlinear. When using the local minimum $\theta_1$ constructed in Theorem 2 as the subnetwork of $\theta$ in Lemma 1, in order to further construct $\theta'$ shown in Lemma 4, we need to make sure that Assumption 1 and $R(\theta) > 0$ hold. With the same ideas as dicussed in section 3.2.1, we can construct the second subnetwork such that there always exists point $\theta_1$ (and consequently $\theta$) for which both Assumption 1 and $R(\theta) > 0$ hold for popular datasets and CNN architectures.

## 5 Experimental Results

We conduct experiments on CIFAR-10 image set to verify the correctness of Theorem 1 and Theorem 2. We use two approaches to show the existence of spurious local minima. The first one is to visualize the loss landscape around local minima $\theta$ using the technique given in Li et al. (2018). Given two random directions $\delta$ and $\eta$, we compute the empirical losses $R(\theta + \alpha\delta + \beta\eta)$ at grid points in the two-dimensional plane specified by $\delta$ and $\eta$, where $(\alpha, \beta)$ are the coodinates of each grid point. Then, the level sets of empirical loss are depicted using empirical losses at these grid points. The second approach is to compute the empirical losses around $\theta$ along many random directions and see whether losses lower than $R(\theta)$ exist. Experimental details are given in Appendix F.

Fig.2 shows the results of the level set visualization approach. Fig.2(a) and Fig.2(b) demonstrate that the local minimum $\theta$ constructed in Lemma 6 is spurious since $R(\theta') = R(\theta)$ and local minimum with loss lower than $R(\theta')$ exists. Fig.2(c) shows that the point $\theta_1$ constructed in Theorem 2 is a local minimum. Fig.3 shows the results of random direction approach and will be given in Appendix F.

## 6 Conclusion

We have proved that convolutional layers can introduce infinite spurious local minima in the loss landscape of deep convolutional neural networks with squared loss or cross-entropy loss. To show this, we developed new techniques to solve the challenges introduced by convolutional layers. We solved a combinatorial problem to demonstrate that a split of outputs of data samples is always possible somewhere in feature maps. In this combinatorial problem, we overcame the difficulty of arbitrary groupings of outputs of data samples caused by limited receptive fields and arbitrary activation status of hidden neurons. We also solved the tied parameters problem, giving perturbations of filters and biases to decrease the training loss that affect only the output of a single neuron in the feature map.

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

# A    PARAMETER MATRICES FOR AVERAGE POOLING AND FULLY CONNECTED OPERATIONS

We give a one-dimensional example to show how to formulate the average pooling operation as a matrix-vector product. Given a one-dimensional input $\mathbf{o}^{l-1} = (a, b, c, d)^T$, suppose the patches for pooling are non-overlapping, which is usually the case, and the patch size is 2. The output is then $\mathbf{o}^l = (\frac{1}{2}(a+b), \frac{1}{2}(c+d))^T$, which is equivalent to the following matrix-vector product

$$\mathbf{o}^l = \frac{1}{2} \begin{pmatrix} 1 & 1 & 0 & 0 \\ 0 & 0 & 1 & 1 \end{pmatrix} \begin{pmatrix} a \\ b \\ c \\ d \end{pmatrix}. \tag{11}$$

Thus, $W^l = \frac{1}{2} \begin{pmatrix} 1 & 1 & 0 & 0 \\ 0 & 0 & 1 & 1 \end{pmatrix}$. Parameter matrix for average pooling operation in two-dimensional case can be obtained using similar idea by converting the feature maps in input layer into a vector, and the nonzero entries in each row of $W^l$ are not all adjacent.

For fully connected layers, there are no constraints on the entries of parameter matrix $W^l \in \mathbb{R}^{n_l \times n_{l-1}}$.

# B    AN EXAMPLE OF NONTRIVIAL LOCAL MINIMA FOR CNNS

We fit the two groups of samples separated by the hyperplane with normal $\mathbf{p}$ in different ways. For samples $\{\mathbf{x}_j | j \in [N], j \notin I\}$, we use a subnetwork with parameter matrix $W_2$ to fit them, i.e., the output of this subnetwork is

$$\mathbf{o}_j = W_2 \hat{\mathbf{x}}_j, \ \forall j \in [N], j \notin I, \tag{12}$$

where $\hat{x}_j$ is the input augmented with scalar 1 (at the same time weight matrices are augmented to include biases (denoted as $\hat{W}^l$) to make notations simple).

We use a subnetwork with parameter matrix $W_1$ to fit data samples $\{\mathbf{x}_j | j \in I\}$, and $W_1$ is given by

$$W_1(i, \cdot)\hat{\mathbf{x}}_j = W_2(i, \cdot)\hat{\mathbf{x}}_j + \lambda_i(\mathbf{p}^T \mathbf{x_j} + c_p), \ j \in I, \ i \in [d_y]. \tag{13}$$

Then, the output is

$$\mathbf{o}_j = W_1 \hat{\mathbf{x}}_j = W_2 \hat{\mathbf{x}}_j + (\mathbf{p} \cdot \mathbf{x}_j + c_p) \cdot \left(\lambda_1, \lambda_2, \ldots, \lambda_{d_y}\right)^T, \ j \in I. \tag{14}$$

$\{\lambda_i, \ i \in [d_y]\}$ are constants determined as follows. If there is only one sample $\mathbf{x}_{i^*}$ in set $I$ with target output $\mathbf{y}_{i^*}$ and the loss function is the squared loss, there must exist $\tilde{\mathbf{y}}_{i^*}$ such that $l\left(\tilde{\mathbf{y}}_{i^*}, \mathbf{y}_{i^*}\right) = 0$. Let $\mathbf{o}_{i^*} = \tilde{\mathbf{y}}_{i^*}$, then we have $\tilde{y}_{i^*}(j) = W_2(j, \cdot)\hat{\mathbf{x}}_{i^*} + \lambda_j (\mathbf{p} \cdot \mathbf{x}_{i^*} + c_p), \forall j \in [d_y]$, which leads to

$$\lambda_j = \frac{\tilde{y}_{i^*}(j) - W_2(j, \cdot)\hat{\mathbf{x}}_{i^*}}{x_{i^*}^k + c_p}, \ \forall j \in [d_y]. \tag{15}$$

If there are multiple elements in set $I$ or the loss function is the cross-entropy loss, the parameters $\{\lambda_j, j \in [d_y]\}$ are determined by

$$\{\lambda_j, \ j \in [d_y]\} = \text{argmin}_{\{\mu_j, \ j \in [d_y]\}} \sum_{i \in I} l\left(W_2 \hat{\mathbf{x}}_i + \left(x_i^k + c_p\right)\left(\mu_1, \mu_2, \cdots, \mu_{d_y}\right)^T, \mathbf{y}_i\right). \tag{16}$$

Concatenating the two subnetworks, we obtain a full CNN. The following theorem gives its architecture and parameters $\left(W^l, \mathbf{b}^l\right)_{l=1}^{L}$, and shows that $\theta_1 := \left(W^l, \mathbf{b}^l\right)_{l=1}^{L}$ is a local minimum. For ease of presentation, we ignore pooling layers at first.

**Theorem 3.** *(formal version of Theorem 2). Given a CNN, assume the number of filters in each convolutional layer satisfies $T_l \geq 2$ ($l \in [L-1]$), and the size of each convolutional layer satisfies $n_l - P_{l-1} \geq d_y$ ($l \in [L-1]$). Its parameters are given as follows. The weight matrix of the first filter in the first convolutional layer is*

$$W^1(1 : P_0, 1 : P_0) = I_{d_{P_0 \times P_0}}, \tag{17}$$

where $I_{d_{P_0 \times P_0}}$ is the identity matrix of size $P_0 \times P_0$, $P_0$ is the number of patches in the input used for convolution. The corresponding bias is

$$b_1^1 = c_p. \tag{18}$$

The filters and biases of higher convolution layers are set as

$$W^l(1 : P_{l-1}, 1 : P_{l-1}) = I_{d_{P_{l-1} \times P_{l-1}}}, \ \ b_1^l = 0, \ \ l \in \{2, 3, \cdots, L-1\} \tag{19}$$

to forward propagate the first feature map of $o^1(\boldsymbol{x}_i)$ unchanged.

The subnetwork with parameter matrix $W_2$ is formed by the part of CNN that removes the first filter and all activation units in each layer, and its output is given as follows. For all $j \in [N]$ and $j \notin I$,

$$\begin{aligned}
\mathbf{o}_j = W_2\hat{\mathbf{x}}_j :=& W^L(\cdot, P_{L-2}+1:n_{L-1})[W^{L-1}(P_{L-2}+1:n_{L-1}, P_{L-3}+1:n_{L-2})\cdots \\
& (W^1(P_0+1:n_1, \cdot)\mathbf{x}_j + b^1(P_0+1:n_1)) + \cdots + b^{L-1}(P_{L-2}+1:n_{L-1})] + \mathbf{b}^L.
\end{aligned} \tag{20}$$

$W_2$ is optimized to minimize $\sum_{\substack{i \in [N], \\ i \notin I}} l(W_2\hat{\mathbf{x}}_i, \mathbf{y}_i)$.

The final fully connected layer is set as

$$W^L(i, 1 : P_{L-2}) = (0, \cdots, \lambda_i, \cdots, 0), \ i \in [d_y], \tag{21}$$

where only the $k$th component connecting to the first subnetwork is nonzero and equals to $\lambda_i$.

Sufficiently large positive constants $c_l$ ($l \in [L-1]$) are added to the biases of each convolutional layer in the second subnetwork such that the involved ReLU units are all activated. The biases of top fully connected layer $b_i^L$ ($i \in [d_y]$) are then substituted by $b_i^L - \Delta o^L(i)$, where $\Delta o^L(i)$ is recursively computed as

$$\begin{aligned}
\Delta o^1(i) =& c_1, \ \ i \in [P_0 + 1 : n_1], \\
\Delta o^l(i) =& \sum_{j \in [P_{l-2}+1:n_{l-1}]} W^l(i, j)\Delta o^{l-1}(j) + c_l, \ i \in [P_{l-1}+1:n_l], \ l \in \{2, 3, \cdots, L-1\}, \\
\Delta o^L(i) =& \sum_{j \in [P_{L-2}+1:n_{L-1}]} W^L(i, j)\Delta o^{L-1}(j), \ i \in [d_y].
\end{aligned} \tag{22}$$

All remaining parameters are set to zeros. Then, the point $\theta_1 = (W^l, \mathbf{b}^l)_{l=1}^L$ specified above is local minimum in parameter space.

If pooling layers are used in CNNs, we can adjust the parameters in Theorem 3 to reflect the effects of parameter matrices of pooing layers. However, the adjusted $(W^l, \mathbf{b}^l)_{l=1}^L$ is still a local minimum.

The condition for the number of filters is used to contain two subnetworks. The condition for the size of each convolutional layers is used to make sure $\theta_1$ presented in Theorem 3 meets the full rank requirement. Because the size of each convolutional layer satisfies $n_l - P_{l-1} \geq d_y$ ($l \in [L-1]$), the matrix $W_2 \in \mathbb{R}^{d_y \times d_x}$ for the second predictor is full rank around $\theta_1$ (see the proof of Theorem 3 in Appendix G.9). These conditions are easily satisfied by most CNNs used in practice.

## C  LEMMAS FOR CONSTRUCTION OF SPURIOUS LOCAL MINIMA

The following Lemma 6 desribe the construction local minima $\theta$, and it is the formal version of Lemma 1 in section 3.1.

**Lemma 6.** *Given a CNN whose numbers of filters in the last two convolutional layers satisfy $T_{L-1} \geq 3$, $T_{L-2} \geq 3$, the parameters of its last three layers are set as follows to propagate $\mathbf{o}^{L-3}(\mathbf{x})$ to the output layer. The first and second feature maps in both layer $L-2$ and layer $L-1$ have no*

*contributions to the final output.*

$$W^L\left(\cdot, 1 : 2P_{L-2}\right) = 0,$$
$$W^{L-1}\left(i, i\right) = 1,\ i \in \left[min\left(n_{L-1}, n_{L-2}\right)\right];\quad W^{L-1}\left(i, j\right) = 0,\ i \neq j,$$
$$W^{L-2}\left(i, \cdot\right) = 0,\ i \in \left[2P_{L-3}\right];\quad W^{L-2}\left(2P_{L-3} + i, i\right) = 1,\ i \in \left[min\left(n_{L-2} - 2P_{L-3}, n_{L-3}\right)\right],$$
$$W^{L-2}\left(2P_{L-3} + i, j\right) = 0,\ i \neq j,\ i \in \left[min\left(n_{L-2} - 2P_{L-3}, n_{L-3}\right)\right],$$
$$\mathbf{b}^{L-1} = \mathbf{0},\ \mathbf{b}^{L-2} = \mathbf{0}.$$

$$(23)$$

*The remaining parameters, including $W^L(\cdot, 2P_{L-2} + 1 : end)$, $\mathbf{b}^L$ and $\left(W^l, \mathbf{b}^l\right)_{l=1}^{L-3}$, constitute a subnetwork, and they locally minimize the following training loss when fixing those parameters in (23),*

$$\tilde{R}(\left(W^l, \mathbf{b}^l\right)_{l=1}^{L-3}, W^L(\cdot, 2P_{L-2} + 1 : end),\ \mathbf{b}^L)$$
$$:= \frac{1}{N}\sum_{i=1}^{N} l\left(W^L I^{L-1,i}\left(W^{L-1} \cdots \left(W^1 \mathbf{x}_i + \mathbf{b}^1\right) + \cdots + \mathbf{b}^{L-1}\right) + \mathbf{b}^L, \mathbf{y}_i\right).$$

$$(24)$$

*Assume that $\hat{I}^{l,i} \neq 0$ for all $l \in [L-1]$ and $i \in [N]$. Then, the point $\theta := \left(W^l, \mathbf{b}^l\right)_{l=1}^{L}$ obtained above is a local minimum in parameter space.*

The requirement in Lemma 6 that $\hat{I}^{l,i} \neq 0$ for all $l \in [L-1]$ and $i \in [N]$ is used to prevent degenerate cases in which all ReLU neurons in a layer are deactivated for some samples. The nondegenerate cases always exist since we can fix sufficiently big biases such that all ReLU neurons are turned on and then train the subnetwork.

The following Lemma 7 shows that for practical datasets and popular CNN architectures, there always exists point $\theta$ in parameter space where Assumption 1 holds.

**Lemma 7.** *For CNNs whose number of feature maps in each convolutional and average pooling layer is greater than or equals to the number of input channels $n^0$ and width of each fully connected layer is bigger than or equals to that of output layer, and datasets in which all data points are distinct, i.e., $\forall i, j \in [N]$ and $i \neq j$, $\mathbf{x}_i \neq \mathbf{x}_j$, if the distinctness of samples is preserved after each pooling operation of CNNs when directly applied to the input, then there always exists point $\theta$ for which Assumption 1 holds.*

For popular CNN architectures, such as AlexNet and VGG used in CIFAR10 and ImageNet image classifiction tasks, the number of feature maps in each layer is greater than the number of input channels, and the output layer is indeed narrower than hidden fully connected layers. For practical datasets, the distinctness of samples is usually preserved after several pooling operations. Thus, the conditions in Lemma 7 are reasonable in practice.

*Proof.* We consider the general case in which the two consecutive convolutional layers we utilized are not located at the top, and there are some convolutional, pooling, or fully connected layers between them and the final output layer. For CNNs whose number of feature maps in each convolutional and average pooling layer is greater than or equals to the number of input channels, we can set the parameters $\left(W^l, \mathbf{b}^l\right)_{l=1}^{L}$ such that each convolutional layer forward propagates its input unchanged using the first $n^0$ feature maps and the remaining feature maps are set to zeros using zero filters and biases. Also, since the width of each fully connected layer is bigger than or equals to that of output layer, we use the first $d_y$ neurons in the first fully connected layer to perform the fully connection operation, and subsequent fully connected layers forward propagate their inputs unchanged to the final output neurons. Sufficiently large biases are used in the first convolutional layer and the first fully connected layer such that all ReLU neurons in the first $n^0$ feature maps of convolutional and pooling layers and the first $d_y$ ReLU neurons in hidden fully connected layers are turned on, and their effects can be cancelled at the final output layer (as done for the second subnetwork in 22 in Theorem 3).

By doing so, the training loss of the CNN can be expressed as $R(\theta) = \frac{1}{N}\sum_{i=1}^{N} l\left(W^{fc}W^{pool}\mathbf{x}_i, \mathbf{y}_i\right)$, where $W^{pool}$ is a constant matrix that characterizes the average pooling operations from input to the first fully connected layer, and $W^{fc}$ characterizes the computation in the first fully connected layer. The CNN has been reduced to a linear classifier and the corresponding point $\theta$ in parameter space becomes a local minimum when minimizing $\frac{1}{N}\sum_{i=1}^{N} l\left(W^{fc}W^{pool}\mathbf{x}_i, \mathbf{y}_i\right)$ with respect to $W^{fc}$.

The only operation that can affect the distinctness of $\mathbf{o^{L-1,i}}$ ($i \in [N]$) at layer $L-1$ (or more generally, the two consecutive convolutional layers used in the construction of spurious local minima) is then the pooling operation. If the distinctness of samples $\mathbf{x}_i$ ($i \in [N]$) is preserved after each pooling operation of the CNN when directly applied to $\mathbf{x}_i$ (i.e., $W^{pool}\mathbf{x}_i$), then $\forall i, j \in [N]$ and $i \neq j$, $\mathbf{o}^{L-1,i} \neq \mathbf{o}^{L-1,j}$, hence Assumption 1 holds for $\theta$.

$\square$

The following Lemma 8 shows that for training data that cannot be fit by linear models and popular CNN architectures, point $\theta$ in parameter space with training loss $R(\theta) > 0$ always exists.

**Lemma 8.** *For training data that cannot be fit by linear models, and CNNs whose number of feature maps in each convolutional and average pooling layer is greater than or equals to the number of input channels $n^0$ and width of each fully connected layer is bigger than or equals to that of output layer, there always exists a point $\theta = \left(W^l, \mathbf{b}^l\right)_{l=1}^{L}$ in parameter space such that training loss $R(\theta) > 0$.*

The point $\theta = \left(W^l, \mathbf{b}^l\right)_{l=1}^{L}$ in Lemma 8 can be the same as that in Lemma 7. Therefore, combining the two lemmas, there exists point $\theta$ where both Assumption 1 and $R(\theta) > 0$ hold.

Most practical datasets, such as CIFAR10 and ImageNet, are complex and cannot be fit by linear models. Thus, the conditions in Lemma 8 are reasonable in practice.

*Proof.* Let $W^\star \in \mathbb{R}^{d_y \times d_x}$ is a local minimizer of $\frac{1}{N}\sum_{i=1}^{N} l\left(W\mathbf{x}_i, \mathbf{y}_i\right)$, where $l$ is the loss function. For training data that cannot be fit by linear models, we have $\frac{1}{N}\sum_{i=1}^{N} l\left(W^\star\mathbf{x}_i, \mathbf{y}_i\right) > 0$. We use the same parameter settings as in the proof of Lemma 7, and then the training loss of CNN is expressed as $R(\theta) = \frac{1}{N}\sum_{i=1}^{N} l\left(W^{fc}W^{pool}\mathbf{x}_i, \mathbf{y}_i\right)$. Since $\frac{1}{N}\sum_{i=1}^{N} l\left(W^\star\mathbf{x}_i, \mathbf{y}_i\right) = \min_{W \in \mathbb{R}^{d_y \times d_x}} \frac{1}{N}\sum_{i=1}^{N} l\left(W\mathbf{x}_i, \mathbf{y}_i\right) > 0$, we have $R(\theta) > 0$. $\square$

The following Lemma 9 descibe the construction the point $\theta'$, and it is the formal version of Lemma 4 in section 3.2.2.

**Lemma 9.** *Given the CNN parameters $\theta := (W^l, \mathbf{b}^l)_{l=1}^{L}$ specified in Lemma 6, let $\theta' := \left(W'^l, \mathbf{b}'^l\right)_{l=1}^{L}$. If $\eta \geq -b_1^{L-3}$, set*

$$W'^L\left(1, P_{L-2}+1 : 2P_{L-2}\right) = W^L\left(1, 2P_{L-2}+1 : 3P_{L-2}\right),$$

$$W'^L\left(1, 1 : P_{L-2}\right) = W^L\left(1, 2P_{L-2}+1 : 3P_{L-2}\right), \quad W'^L\left(1, 2P_{L-2}+1 : 3P_{L-2}\right) = \mathbf{0}^T,$$

$$W'^L\left(2 : d_y, 1 : 2P_{L-2}\right) = 0,$$

$$W'^{L-1}(i,i) = -1, \ i \in [P_{L-3}]; \qquad W'^{L-1}(i,i) = +1, \ i \in \{P_{L-3}+1 : 2P_{L-3}\},$$

$$b_1'^{L-1} = \eta + b_1^{L-3}, \qquad b_2'^{L-1} = \eta + b_1^{L-3}, \tag{25}$$

$$W'^{L-2}(i,i) = -1, \ i \in [P_{L-4}]; \qquad W'^{L-2}(P_{L-3}+i, i) = +1, \ i \in [P_{L-4}],$$

$$b_1'^{L-3} = -\min_{i \in [N], \, j \in [P_{L-4}]} v_i^j, \qquad b_1'^{L-2} = \eta + b_1'^{L-3};$$

$$b_2'^{L-2} = -\eta - b_1'^{L-3}; \qquad b_1'^L = b_1^L - \sum_{j=1}^{P_{L-2}} w^j\left(\eta + b_1^{L-3}\right).$$

*If $\eta < -b_1^{L-3}$, set*

$$
\begin{aligned}
&W'^L(1, 1:P_{L-2}) = W^L(1, 2P_{L-2}+1:3P_{L-2}), \\
&W'^L(1, P_{L-2}+1:2P_{L-2}) = -W^L(1, 2P_{L-2}+1:3P_{L-2}), \quad W'^L(2:d_y, 1:2P_{L-2}) = 0; \\
&W'^{L-1}(i,i) = 1, \ i \in [P_{L-3}]; \qquad\qquad W'^{L-1}(i,i) = 1, \ i \in [P_{L-3}+1:2P_{L-3}]; \\
&W'^{L-1}(i,i) = 1, \ i \in [2P_{L-3}+1:3P_{L-3}], \\
&W'^{L-2}(i,i) = 1, \ i \in [P_{L-4}]; \qquad\qquad W'^{L-2}(P_{L-3}+i,i) = 1, \ i \in [P_{L-4}], \\
&b_1'^{L-3} = -\min_{i \in [N], \ j \in [P_{L-4}]} v_i^j, \qquad b_1'^{L-2} = -\eta - b_1'^{L-3}, \qquad b_2'^{L-2} = -\eta - b_1'^{L-3}, \\
&b_3'^{L-2} = -\eta - b_1'^{L-3}, \qquad b_1'^{L-1} = 0, \qquad b_2'^{L-1} = 0, \qquad b_3'^{L-1} = \eta + b_1'^{L-3}; \qquad b_i'^L = b_i^L, \ i \in [d_y].
\end{aligned}
$$
(26)

*Remaining parameters in $\theta$ keep fixed. Then, $R(\theta') = R(\theta)$.*

The purpose of introducing $b_1'^{L-3}$ is to make $\sigma(v_i^j + b_1'^{L-3}) \geq 0$ for all $i \in [N]$, $j \in [P_{L-4}]$.

Next, we show that by perturbing $\theta'$, we can decrease the training loss. We use different perturbation schemes for different cases of $\eta$. The following Lemma 10 descibe the perturbation scheme, and it is the formal version of Lemma 5 in section 3.2.3.

**Lemma 10.** *If $\eta \geq -b_1'^{L-3}$, perturb $b_2'^{L-2} = -\eta - b_1'^{L-3}$ as*

$$
b_2'^{L-2} \to b_2'^{L-2} + \delta b = -\eta - b_1'^{L-3} + \delta b.
$$
(27)

*If $\eta < -b_1'^{L-3}$, perturb $b_1'^{L-2} = -\eta - b_1'^{L-3}$ as*

$$
b_1'^{L-2} \longrightarrow b_1'^{L-2} + \delta b = -\eta - b_1'^{L-3} + \delta b.
$$
(28)

*Remaining parameters in $\theta'$ keep fixed. Then under Assumption 1, a training loss lower than $R(\theta')$ is obtained under this perturbation with a proper sign of $\delta b$.*

## D    CONSTRUCTION OF SPURIOUS LOCAL MINIMA: THE GENERAL CASE

When the two consecutive convolutional layers we utilized are not located at the top, and there are some convolutional, pooling, or fully connected layers between the two consecutive convolutional layers used in our construction and the final output layer, we can still construct spurious local minima using settings similar to Lemmas 6, 9 and 10. We let the first fully connected layer play the role of layer $L$ in Lemmas 6 and 9, and set the parameters of layers above the two consecutive convolutional layers such that the output of the two consecutive convolutional layers are propagated unchanged (except the pooling operations in subsequent pooling layers) to the first fully connected layer, whose output is then forwarded invariantly to the final output neurons. Similar to the proof of Lemma 7, this is always possible by utilizing the minimal widths among subsequent convolutional layers and fully connected layers, respectively.

Then, by reserving the first two feature maps in layers starting from the two consecutive convolutional layers and minimizing the loss of the remaining subnetwork, we can obtain a local minimum $\theta$. By setting the parameters connected to the two consecutive convolutional layers as in Lemma 9, we can obtain $\theta'$. The positive value of $\sigma(v_i^j + b_1^{L-3})$ may pass through different paths as before for different location $j$. If there are average pooling layers above the two consecutive convolutional layers, since average pooling operation is linear, the average poolings in feature maps of different paths are finally aggregated in the first fully connected layer, producing an output equal to that of $\theta$. Therefore, there is still $R(\theta') = R(\theta)$. The perturbation scheme is the same as in Lemma 10, and the perturbation will persist after passing through the linear average pooling operations, thus the empirical risk can be decreased by perturbing $\theta'$ as before.

# E    More Auxiliary Lemmas

Denote $u_i := \frac{\partial l(\mathbf{o}(\mathbf{x}_i), \mathbf{y}_i)}{\partial o(1)}$ and $o_i := o^{L,i}(1) = \sum_{j=1}^{M} w^j \sigma \left( v_i^j + b_{t_j}^{L-3} \right) + b_1^L$, where $b_{t_j}^{L-3}$ is the bias associated with the feature map in which location $j$ resides. We have the following lemma for squared and cross-entropy loss.

**Lemma 11.** *For convolutional neural networks with squared loss or cross-entropy loss, the followings can be achieved through perturbing $W^L(1, \cdot)$ under Assumption 1, where $o_i'$ and $u_i'$ denote, respectively, $o_i$ and $u_i$ after perturbation.*

*1. $o_i' \neq o_j'$ if $o_i = o_j$  $(i, j \in [N], \ i \neq j)$.*

*2. $u_i' \neq 0$ if $u_i = 0$  $(i \in [N])$.*

*3. $u_i' \neq u_j'$ or $u_i' + u_j' \neq 0$ if $u_i = u_j$ or $u_i + u_j = 0$, respectively,  $(i, j \in [N], \ i \neq j)$.*

*4. $\mathbf{u} = (u_1, u_2, \cdots, u_N)^T \neq \mathbf{0}$ if $R = \frac{1}{N} \sum_{i=1}^{N} l(\mathbf{o}_i, \mathbf{y}_i) > 0$.*

*5. Nonzero gaps between $o_i'$s or $u_i'$s can still exist after subsequent perturbations.*

The maintenance of nonzero gaps between $o_i'$s or $u_i'$s guarantees that $o_i' \neq o_j'$ or $u_i' \neq u_j'$ etc. still hold after subsequent perturbations.

In the following, we absorb biases into weights and the CNN output is written as $\mathbf{o}_i = W^L I^{L-1,i} W^{L-1} \cdots I^{1,i} W^1 \mathbf{x}_i$. Here, we omit the hat on augmented variables for notational simplicity, and the exact meaning of involved symbols is clear from context. The following two lemmas will be used to prove $u_i' \neq u_j'$ or $u_i' + u_j' \neq 0$ $(i, j \in [N], \ i \neq j)$ in Lemma 11 for cross-entropy loss.

**Lemma 12.** *For cross-entropy loss, if $u_i = u_j$ for two samples $\mathbf{x}_i, \mathbf{x}_j$ $(i \neq j)$ and $I^{L-1,i} W^{L-1} \cdots W^1 \mathbf{x}_i = \alpha I^{L-1,j} W^{L-1} \cdots I^{1,j} W^1 \mathbf{x}_j$ $(\alpha \neq 0, \alpha \neq 1)$, then under Assumption 1, $u_i' \neq u_j'$ can be achieved through perturbing $W^L(1, \cdot)$.*

**Lemma 13.** *For cross-entropy loss, if $u_i + u_j = 0$ for two samples $\mathbf{x}_i, \mathbf{x}_j$ $(i \neq j)$ and $I^{L-1,i} W^{L-1} \cdots W^1 \mathbf{x}_i = \alpha I^{L-1,j} W^{L-1} \cdots I^{1,j} W^1 \mathbf{x}_j$ $(\alpha \neq 0, \alpha \neq 1)$, then under Assumption 1, $u_i' + u_j' \neq 0$ can be achieved through perturbing $W^L(1, \cdot)$.*

# F    Experimental Details

We use CIFAR-10 image set to train CNNs, which consists of 10 classes and 50000 training images of size $32 \times 32$. The CNN used in our experiments has 7 convolutional layers, with the number of channels being $64, 64, 128, 128, 256, 256, 256$, respectively, and no pooling layers are used. The convolution filters are all $3 \times 3$. Each convolutional layer is followed by a ReLU layer. The subnetwork in Lemma 6 is trained by Adam optimizer with 150 epochs, with a learning rate of 0.001 and weight decay of 0.0005. The subnetwork $W_2$ in Theorem 3 is trained by Adam optimizer with 500 epochs, with a learning rate of 0.0001 and weight decay of 0.0005. Both subnetworks use the Kaiming initialization (He et al. (2015)). The biggest threshold $\eta$ in Lemma 2 is found to be 6.8915.

Fig.3 shows the results of random direction approach.

# G    Missing Proofs

For the proofs in the following sections, we assume that the perturbation of network parameters for a differentiable local minimum $\theta = \left( W^l, \mathbf{b}^l \right)_{l=1}^{L}$ is very small such that the activation patterns $I^{l,i}$ $(l \in [L-1], i \in [N])$ will keep constant.

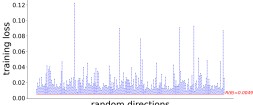
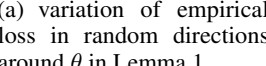
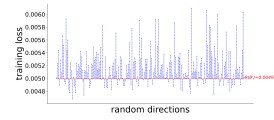
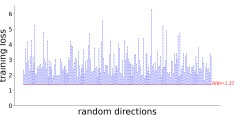

(a) variation of empirical loss in random directions around $\theta$ in Lemma 1

(b) variation of empirical loss in random directions around $\theta'$ in Lemma 3

(c) variation of empirical loss in random directions around $\theta_1$ in Theorem 2

Figure 3: The variation of empirical loss in 200 random directions around specific locations in parameter space. In (b), some directions have losses lower than $R(\theta') = R(\theta)$. (a) and (b) together demonstrate that $\theta$ is a spurious local minimum. (c) shows that $\theta_1$ constructed in Theorem 2 is a local minimum.

### G.1 PROOF OF LEMMA 6

*Proof.* When parameters are perturbed as $\left(W^l, \mathbf{b}^l\right)_{l=1}^{L} \longrightarrow \left(W^l + \delta W^l, \mathbf{b}^l + \delta \mathbf{b}^l\right)_{l=1}^{L}$, the output after perturbation is as follows,

$$
\begin{aligned}
\mathbf{o}'\left(\mathbf{x}_i\right) := \mathbf{o}'^{L,i} &= \left(\hat{W}^L + \delta \hat{W}^L\right) \hat{I}^{L-1,i} \left(\hat{W}^{L-1} + \delta \hat{W}^{L-1}\right) \hat{I}^{L-2,i} \cdots \left(\hat{W}^1 + \delta \hat{W}^1\right) \hat{\mathbf{x}}_i \\
&= (\hat{W}^L \hat{I}^{L-1,i} \hat{W}^{L-1} \hat{I}^{L-2,i} \cdots \hat{W}^1 + \delta F^i)\hat{\mathbf{x}}_i, \quad \forall i \in [N],
\end{aligned}
\tag{29}
$$

where $\delta F^i = \delta \hat{W}^L \hat{I}^{L-1,i} \hat{W}^{L-1} \hat{I}^{L-2,i} \cdots \hat{W}^1 + \hat{W}^L \hat{I}^{L-1,i} \delta \hat{W}^{L-1} \hat{I}^{L-2,i} \cdots \hat{W}^1 + \cdots + \delta \hat{W}^L \hat{I}^{L-1,i} \delta \hat{W}^{L-1} \hat{I}^{L-2,i} \cdots \delta \hat{W}^1$.

The training loss after perturbation is

$$
\begin{aligned}
R((W^l + \delta W^l, \mathbf{b}^l + \delta \mathbf{b}^l)_{l=1}^{L}) &= \frac{1}{N} \sum_{i=1}^{N} l\left(\mathbf{o}'\left(\mathbf{x}_i\right), \mathbf{y}_i\right) \\
&= \frac{1}{N} \sum_{i=1}^{N} l\left(\hat{W}^L \hat{I}^{L-1,i} \hat{W}^{L-1} \hat{I}^{L-2,i} \cdots \hat{W}^1 \hat{\mathbf{x}}_i + \delta F^i \hat{\mathbf{x}}_i, \mathbf{y}_i\right).
\end{aligned}
\tag{30}
$$

Since for every sample the output of each layer is nondegenerate, when minimizing $\tilde{R}\left(\left(W^l, \mathbf{b}^l\right)_{l=1}^{L-3}, W^L(\cdot, 2P_{L-2} + 1 : end), \mathbf{b}^L\right) := \frac{1}{N} \sum_{i=1}^{N} l\left(\hat{W}^L \hat{I}^{L-1,i} \hat{W}^{L-1} \hat{I}^{L-2,i} \cdots \hat{W}^1 \hat{\mathbf{x}}_i, \mathbf{y}_i\right)$ in (24), the space of $\mathbf{o}\left(\mathbf{x}_i\right)$ has been fully explored by $\hat{W}^L \hat{I}^{L-1,i} \hat{W}^{L-1} \cdots \hat{W}^1$ in the neighborhood during minimization. Therefore, $R((W^l + \delta W^l, \mathbf{b}^l + \delta \mathbf{b}^l)_{l=1}^{L})$ cannot be lower than $R((W^l, \mathbf{b}^l)_{l=1}^{L})$ despite the perturbation in $\delta F^i$, and $\left(W^l, \mathbf{b}^l\right)_{l=1}^{L}$ is thus a local minimum in parameter space.

$\square$

### G.2 PROOF OF LEMMA 2

*Proof.* At a location $j \in [M]$ (more specifically, $M := \min\left(n_{L-1} - 2P_{L-2}, n_{L-2} - 2P_{L-3}, n_{L-3}\right)$), if there exist some groups in the ordered list of $v_i^j$ ($i \in [N]$) such that $\sum_{i \in I_q^j} u_i \neq 0$, and suppose $G_1 = \left\{v_{m_q+1}^j, v_{m_q+2}^j, \cdots, v_{m_{q+1}}^j\right\}$ is the first such group in the ordered list, we can set $\eta = \frac{1}{2}\left(v_{m_{q+1}+1}^j + v_{m_{q+1}}^j\right)$, which is the midpoint of the gap between group $G_1$ and the next group, and set $n = m_{q+1}$. We then have $\sum_{i=1}^{n} u_i = \sum_{i=1}^{m_q} u_i + \sum_{i \in G_1} u_i = \sum_{i \in G_1} u_i \neq 0$. The requirements in (7) are satisfied and we have found a split for data samples.

If for every group $q \in [g_j]$ at location $j$, there is $\sum_{i \in I_q^j} u_i = 0$, we have to explore other locations in feature maps to see whether there exist such splits. According to Lemma 3, $\sum_{i \in I_q^j} u_i = 0$

cannot happen for every group $q$ and every location $j$, and there must exist some locations where $\sum_{i \in I_q^j} u_i \neq 0$ for some groups. As a result, we can split data samples as before at these locations.

There may be more than one location where such data split exists, and we choose the one with the biggest threshold $\eta$ and use it in the perturbation of training loss. Let $h$ be the location having the biggest split threshold, we then have

$$\forall j \in [M], j \neq h, \sum_{i \in \{1,2,\cdots,N \mid v_i^j > \eta\}} u_i = 0. \tag{31}$$

$\square$

### G.3  PROOF OF LEMMA 9

*Proof.* There are two possible cases for $\eta$ : $\eta \geq -b_1^{L-3}$ or $\eta < -b_1^{L-3}$, which will be treated differently in the following.

In order to compare the training losses at $\theta$ and $\theta'$, we need to compute the output $\mathbf{o}'(\mathbf{x}_i) := \mathbf{o}'^{L,i}$ obtained by $\theta'$. We will give the computation of its first component $o_1'(\mathbf{x}_i)$ in detail. We only need to compute those terms in $o_1'(\mathbf{x}_i)$ that are affected by the parameter settings in (25) and (26), and denote by $\tilde{o}'(\mathbf{x}_i)$ the sum of such terms.

If $\eta \geq -b_1^{L-3}$, only the first and the second feature maps in layers $L-1$ and $L-2$ are involved. Let $\mathbf{o}'^{L-3,i} := \sigma(W^{L-3}(1:P_{L-4},\cdot)\mathbf{o}^{L-4,i} + b_1'^{L-3}\mathbf{1}_{P_{L-4}})$, where $\mathbf{1}_{P_{L-4}}$ is the vector of size $P_{L-4}$ whose components are all 1s. $\mathbf{o}'^{L-3,i}$ is the output of the first feature map at layer $L-3$ using the new bias $b_1'^{L-3}$. Since $W^{L-3}(j,\cdot)\mathbf{o}^{L-4,i} + b_1'^{L-3} \geq 0$ for all $i \in [N]$, $j \in [P_{L-4}]$, there is $\mathbf{o}'^{L-3,i} = W^{L-3}(1:P_{L-4},\cdot)\mathbf{o}^{L-4,i} + b_1'^{L-3}\mathbf{1}_{P_{L-4}}$. Then, the output $\tilde{o}'(\mathbf{x}_i)$ is

$$\begin{aligned}
\tilde{o}'(\mathbf{x}_i) = &W'^L(1,1:P_{L-2}) \\
&\sigma\big(W'^{L-1}(1:P_{L-2},1:P_{L-3}) \\
&\quad \sigma\big(W'^{L-2}(1:P_{L-3},1:P_{L-4})\mathbf{o}'^{L-3,i} + b_1'^{L-2}\mathbf{1}_{P_{L-3}}\big) + b_1'^{L-1}\mathbf{1}_{P_{L-2}}\big) + \\
&W'^L(1,P_{L-2}+1:2P_{L-2}) \\
&\sigma\big(W'^{L-1}(P_{L-2}+1:2P_{L-2},P_{L-3}+1:2P_{L-3}) \\
&\quad \sigma\big(W'^{L-2}(P_{L-3}+1:2P_{L-3},1:P_{L-4})\mathbf{o}'^{L-3,i} + b_2'^{L-2}\mathbf{1}_{P_{L-3}}\big) + b_2'^{L-1}\mathbf{1}_{P_{L-2}}\big) + b_1'^L.
\end{aligned} \tag{32}$$

For notational simplicity, denote $W'^L(1,i)\,(i \in [P_{L-2}])$ as $w^i$, and note that $v_i^j := W^{L-3}(j,\cdot)\mathbf{o}^{L-4,i}$, $(j \in [P_{L-4}])$. Using (25), we have

$$\begin{aligned}
\tilde{o}'(\mathbf{x}_i) = \sum_{j=1}^{P_{L-2}} \Big[ &w^j \sigma\Big(-\sigma\big((-v_i^j - b_1'^{L-3}) + \eta + b_1'^{L-3}\big) + \eta + b_1^{L-3}\Big) + \\
&w^j \sigma\Big(\sigma\big((v_i^j + b_1'^{L-3}) - \eta - b_1'^{L-3}\big) + \eta + b_1^{L-3}\Big) - w_j\big(\eta + b_1^{L-3}\big)\Big] + b_1^L
\end{aligned} \tag{33}$$

$$= \sum_{j=1}^{P_{L-2}} w^j \cdot z_j' + b_1^L,$$

where

$$\begin{aligned}
z_j' := &\sigma\Big(-\sigma\big((-v_i^j - b_1'^{L-3}) + \eta + b_1'^{L-3}\big) + \eta + b_1^{L-3}\Big) + \\
&\sigma\Big(\sigma\big((v_i^j + b_1'^{L-3}) - \eta - b_1'^{L-3}\big) + \eta + b_1^{L-3}\Big) - \big(\eta + b_1^{L-3}\big).
\end{aligned} \tag{34}$$

We also denote

$$z_j := \sigma(v_i^j + b_1^{L-3}), \tag{35}$$

and thus the partial output obtained from $\theta$ is

$$\tilde{o}(\mathbf{x}_i) = \sum_{j=1}^{P_{L-2}} w^j \cdot z_j + b_1^L. \tag{36}$$

There are three possible cases for the value of $v_i^j$. Note that we already have $\eta + b_1^{L-3} \geq 0$ if $\eta \geq -b_1^{L-3}$.

1) $v_i^j \geq \eta$.
In this case, we have $v_i^j - \eta \geq 0$ and $v_i^j + b_1^{L-3} \geq v_i^j - \eta \geq 0$, then

$$
\begin{aligned}
z_j' &= \sigma(\eta + b_1^{L-3}) + \sigma(v_i^j - \eta + \eta + b_1^{L-3}) - (\eta + b_1^{L-3}) \\
&= v_i^j + b_3^{L-3} = z_j > 0, \quad \forall j \in [P_{L-2}].
\end{aligned}
\tag{37}
$$

2) $-b_1^{L-3} \leq v_i^j < \eta$.
We have $-v_i^j + \eta > 0$. By (34) and $v_i^j + b_1^{L-3} \geq 0$, we then have

$$
\begin{aligned}
z_j' &= \sigma\left(-\left(-v_i^j + \eta\right) + \eta + b_1^{L-3}\right) + \sigma\left(\eta + b_1^{L-3}\right) - \left(\eta + b_1^{L-3}\right) \\
&= \sigma(v_i^j + b_1^{L-3}) = z_j \geq 0, \quad \forall j \in [P_{L-2}].
\end{aligned}
\tag{38}
$$

3) $v_i^j < -b_1^{L-3}$.
In this case, we have $v_i^j < \eta$, $v_i^j + b_1^{L-3} < 0$, then

$$
\begin{aligned}
z_j' &= \sigma\left(v_i^j - \eta + \eta + b_1^{L-3}\right) + \sigma\left(\eta + b_1^{L-3}\right) - \left(\eta + b_1^{L-3}\right) \\
&= \sigma\left(v_i^j + b_1^{L-3}\right) = 0 = z_j, \quad \forall j \in [P_{L-2}].
\end{aligned}
\tag{39}
$$

In all three possible cases, we have obtained $z_j' = z_j, \forall j \in [P_{L-2}]$. Therefore, $\tilde{o}'(\mathbf{x}_i) = \sum_{j=1}^{P_{L-2}} w^j \cdot z_j + b_1^L = \tilde{o}(\mathbf{x}_i)$, $\forall i \in [N]$. We can obtain $o_k(\mathbf{x}_i) = o_k'(\mathbf{x}_i)$ ($\forall i \in [N], \forall k \in [d_y], k \neq 1$) for remaining output components since $W'^L(2 : d_y, 1 : 2P_{L-2}) = 0$. The outputs are thus not changed, so does the training loss.

We now discuss the case of $\eta < -b_1^{L-3}$. Denote $w^i := W^L(1, 2P_{L-2} + i), \forall i \in [P_{L-2}]$. The output $\tilde{o}'(\mathbf{x}_i)$ that is affected by the parameter setting in (26) is as follows,

$$
\begin{aligned}
\tilde{o}'(\mathbf{x}_i) = &W'^L(1, 1 : P_{L-2}) \\
&\sigma\left(W'^{L-1}(1 : P_{L-2}, 1 : P_{L-3})\right. \\
&\sigma\left(W'^{L-2}(1 : P_{L-3}, 1 : P_{L-4})\mathbf{o}'^{L-3,i} + b_1'^{L-2}\mathbf{1}_{P_{L-3}}\right) + b_1'^{L-1}\mathbf{1}_{P_{L-2}}\right) + \\
&W'^L(1, P_{L-2} + 1 : 2P_{L-2}) \\
&\sigma\left(W'^{L-1}(P_{L-2} + 1 : 2P_{L-2}, P_{L-3} + 1 : 2P_{L-3})\right. \\
&\sigma\left(W'^{L-2}(P_{L-3} + 1 : 2P_{L-3}, 1 : P_{L-4})\mathbf{o}'^{L-3,i} + b_2'^{L-2}\mathbf{1}_{P_{L-3}}\right) + b_2'^{L-1}\mathbf{1}_{P_{L-2}}\right) + \\
&W'^L(1, 2P_{L-2} + 1 : 3P_{L-2}) \\
&\sigma\left(W'^{L-1}(2P_{L-2} + 1 : 3P_{L-2}, 2P_{L-3} + 1 : 3P_{L-3})\right. \\
&\sigma\left(W'^{L-2}(2P_{L-3} + 1 : 3P_{L-3}, 1 : P_{L-4})\mathbf{o}'^{L-3,i} + b_3'^{L-2}\mathbf{1}_{P_{L-3}}\right) + b_3'^{L-1}\mathbf{1}_{P_{L-2}}\right) + b_1'^L.
\end{aligned}
\tag{40}
$$

Using (26), we have

$$
\begin{aligned}
\tilde{o}'(\mathbf{x}_i) = &\sum_{j=1}^{P_{L-2}} w^j \left[\sigma\left(\sigma\left(v_i^j + b_1'^{L-3} - \eta - b_1'^{L-3}\right)\right)\right. \\
&- \sigma\left(\sigma\left(v_i^j + b_1'^{L-3} - \eta - b_1'^{L-3}\right)\right) \\
&+ \sigma\left(\sigma\left(v_i^j + b_1'^{L-3} - \eta - b_1'^{L-3}\right) + \eta + b_1^{L-3}\right)\right] + b_1'^L. \\
&= \sum_{j=1}^{P_{L-2}} w^j z_j' + b_1^L,
\end{aligned}
\tag{41}
$$

where

$$
\begin{aligned}
z'_j := & \sigma\left(\sigma\left(v_i^j + b_1'^{L-3} - \eta - b_1'^{L-3}\right)\right) \\
& - \sigma\left(\sigma\left(v_i^j + b_1'^{L-3} - \eta - b_1'^{L-3}\right)\right) \\
& + \sigma\left(\sigma\left(v_i^j + b_1'^{L-3} - \eta - b_1'^{L-3}\right) + \eta + b_1^{L-3}\right).
\end{aligned}
\tag{42}
$$

There are also three possible cases for the value of $v_i^j$. Note that we already have $\eta + b_1^{L-3} < 0$ when $\eta < -b_1^{L-3}$.

1) $v_i^j \geq -b_1^{L-3}$.

In this case, using $v_i^j - \eta > 0$, $v_i^j + b_1^{L-3} \geq 0$ and (42), we have

$$
\begin{aligned}
z'_j &= \left(v_i^j - \eta\right) - \left(v_i^j - \eta\right) + \sigma\left(v_i^j + b_1^{L-3}\right) \\
&= v_i^j + b_1^{L-3} = z_j \geq 0, \quad \forall j \in [p_{L-2}].
\end{aligned}
\tag{43}
$$

2) $\eta \leq v_i^j < -b_1^{L-3}$.

Using $v_i^j - \eta \geq 0$, $v_i^j + b_1^{L-3} < 0$, we have

$$
\begin{aligned}
z'_j &= \left(v_i^j - \eta\right) - \left(v_i^j - \eta\right) + \sigma\left(v_i^j + b_1^{L-3}\right) \\
&= 0 = z_j, \quad \forall j \in [p_{L-2}].
\end{aligned}
\tag{44}
$$

3) $v_i^j < \eta$.

Using $v_i^j - \eta < 0$, $v_i^j + b_1^{L-3} < 0$, we have

$$
z'_j = \sigma\left(\eta + b_1^{L-3}\right) = 0 = z_j = \sigma\left(v_i^j + b_1^{L-3}\right), \quad \forall j \in [p_{L-2}].
\tag{45}
$$

Therefore, in all possible cases of $\eta$ and $v_i^j$, the output $o'_1(\mathbf{x}_i) = o_1(\mathbf{x}_i), \forall i \in [N]$. We have $o_k(\mathbf{x}_i) = o'_k(\mathbf{x}_i)$ ($\forall i \in [N], \forall k \in [d_y], k \neq 1$) for remaining output components. As a result, the training loss satisfies $R(\theta) = R(\theta')$.

$\square$

## G.4 PROOF OF LEMMA 10

*Proof.* We will show that by perturbing $\theta'$, the training loss can be decreased. We use different perturbation schemes to show this for different cases of $\eta$.

If $\eta \geq -b_1^{L-3}$, we only perturb $b_2'^{L-2} = -\eta - b_1'^{L-3}$ as

$$
b_2'^{L-2} \to b_2'^{L-2} + \delta b = -\eta - b_1'^{L-3} + \delta b.
\tag{46}
$$

Under this perturbation, the output of each sample is perturbed as follows according to different cases of $v_i^j$.

1) $v_i^j \geq \eta$.

In this case, after perturbation, by modifying (34) we have

$$
z'_j = \sigma\left(\sigma(v_i^j - \eta + \delta b) + \eta + b_1^{L-3}\right).
$$

Since $\eta$ is at the midpoint of a gap between adjacent $v_i^j$s (as indicated in the proof of Lemma 2), there is $v_i^j > \eta$, hence for sufficiently small perturbation $\delta b$, $v_i^j - \eta + \delta b > 0$ and $v_i^j + b_3^{L-3} + \delta b > 0$ hold. Therefore,

$$
z'_j = v_i^j + b_1^{L-3} + \delta b = z_j + \delta b, \quad \forall j \in [P_{L-2}].
\tag{47}
$$

2) $-b_1^{L-3} \leq v_i^j < \eta$.

Since $v_i^j - \eta + \delta b < 0$, the perturbation $\delta b$ does not pass through the ReLU activation,

$$
z'_j = v_i^j + b_1^{L-3} = z_j, \quad \forall j \in [P_{L-2}].
\tag{48}
$$

3) $v_i^j < -b_1^{L-3}$.

In this case, $v_i^j - \eta + \delta b < 0$, then

$$z_j' = 0 = z_j, \quad \forall j \in [P_{L-2}]. \tag{49}$$

Combining the above three cases, it is found that the output $o_1'(\mathbf{x}_i)$ is perturbed only if $v_i^j > \eta$. Correspondingly, the perturbation of output is

$$\delta o_1'(\mathbf{x}_i) = \delta \tilde{o}'(\mathbf{x}_i) = \sum_{j=1}^{P_{L-2}} w^j \cdot \delta b \cdot I\left(v_i^j > \eta\right), \tag{50}$$

where $I()$ is the indicator function. Other components of output do not change under this perturbation since $W'^L(2:d_y, 1:2P_{L-2}) = 0$. The training loss is perturbed as

$$\delta R' := R(\theta' + \delta\theta') - R(\theta') = \frac{1}{N}\sum_{i=1}^{N} \frac{\partial l(\mathbf{o}'(\mathbf{x}_i), \mathbf{y}_i)}{\partial o_1} \cdot \delta o_1'(\mathbf{x}_i) = \frac{1}{N}\sum_{i=1}^{N} \frac{\partial l(\mathbf{o}(\mathbf{x}_i), \mathbf{y}_i)}{\partial o_1} \cdot \delta o_1'(\mathbf{x}_i),$$

here $\mathbf{o}(\mathbf{x}_i) = \mathbf{o}'(\mathbf{x}_i)$ is used to get $\frac{\partial l(\mathbf{o}'(\mathbf{x}_i),\mathbf{y}_i)}{\partial o_1} = \frac{\partial l(\mathbf{o}(\mathbf{x}_i),\mathbf{y}_i)}{\partial o_1}$. Therefore, we have

$$\delta R' = \frac{1}{N}\sum_{i=1}^{N} u_i \cdot \sum_{j=1}^{P_{L-2}} w^j \cdot \delta b \cdot I\left(v_i^j > \eta\right). \tag{51}$$

For all samples $\mathbf{x}_i$ with $v_i^j > \eta$ ($\forall j \in [P_{L-2}]$), by Lemma 2, only at the $h$th neuron in the first feature map of layer $L-3$, $\sum_{i=1}^{n} u_i \neq 0$ and $v_i^j > \eta$ ($\forall i \in [n]$) both hold. At other locations in this feature map, because $\eta$ is the biggest split threshold among all locations, even if $v_i^j > \eta$ holds, there is $\sum_{i \in \{1,2,\cdots,N|v_i^j > \eta\}} u_i = 0$ (see (8)). We then have

$$\delta R' = \frac{1}{N}\delta b \cdot \sum_{j=1}^{P_{L-2}} w^j \cdot \sum_{i \in \{1,2,\cdots,N|v_i^j > \eta\}} u_i = \frac{1}{N}\delta b \cdot w^h \cdot \sum_{i=1}^{n} u_i \tag{52}$$

Setting $\mathrm{Sgn}(\delta b) = -\mathrm{Sgn}\left(w^h \cdot \sum_{i=1}^{n} u_i\right)$ if $w^h \neq 0$, where $\mathrm{Sgn}()$ is the sign function, we obtain

$$\delta R' < 0. \tag{53}$$

Therefore, the training loss is decreased by the perturbation, resulting in

$$R(\theta) > R(\theta' + \delta\theta'). \tag{54}$$

Thus, $\theta$ is a spurious local minimum.

If $w^h = 0$, we perturb it as well by $w^h \to \delta w$, then

$$\delta R' = \frac{1}{N}\delta b \cdot \delta w \cdot \sum_{i=1}^{n} u_i \tag{55}$$

Setting $\mathrm{Sgn}(\delta b) = -\mathrm{Sgn}\left(\delta w \sum_{i=1}^{n} u_i\right)$ can still decrease the training loss, and thus the conclusion that $\theta$ is a spurious local minimum still holds.

If $\eta < -b_1'^{L-3}$, we only pertur $b_1'^{L-2} = -\eta - b_1'^{L-3}$ as

$$b_1'^{L-2} \longrightarrow b_1'^{L-2} + \delta b = -\eta - b_1'^{L-3} + \delta b. \tag{56}$$

With this perturbation, the output of each sample is perturbed in three different cases according to the value of $v_i^j$.

1) $v_i^j \geq -b_1^{L-3}$.

In this case, by modifying (42) we have

$$z_j' = \left(v_i^j - \eta + \delta b\right) - \left(v_i^j - \eta\right) + \sigma\left(v_i^j + b_1^{L-3}\right)$$
$$= z_j + \delta b, \quad \forall j \in [P_{L-2}]. \tag{57}$$

2) $\eta \leq v_i^j < -b_1^{L-3}$.

In this case, since a gap exists between $\eta$ and $v_i^j$ when splitting samples with $\eta$ in Lemma 2, we have $\eta < v_i^j$ and consequently $v_i^j - \eta + \delta b > 0$ for sufficiently small $\delta b$. Therefore,

$$z_j' = \left(v_i^j - \eta + \delta b\right) - \left(v_i^j - \eta\right) = \delta b = z_j + \delta b, \ \ \forall j \in [P_{L-2}]. \tag{58}$$

3) $v_i^j < \eta$.

Using $v_i^j - \eta + \delta b < 0$, we have

$$z_j' = \sigma\left(\eta + b_1^{L-3}\right) = 0 = z_j, \ \ \forall j \in [P_{L-2}]. \tag{59}$$

The perturbation $\delta b$ has been blocked by ReLU units.

Only in the first and second cases, the output $o_1'\left(\mathbf{x}_i\right)$ is affected by the perturbation $\delta b$. Thus, we have

$$z_j' = z_j + \delta b, \ \ \text{if } v_i^j > \eta, \ \ \forall i \in [N], j \in [P_{L-2}]. \tag{60}$$

The output is perturbed as

$$\delta o_1'\left(\mathbf{x}_i\right) = \sum_{j=1}^{P_{L-2}} w^j \delta z_j' = \sum_{j=1}^{P_{L-2}} w^j \cdot \delta b \cdot I\left(v_i^j > \eta\right). \tag{61}$$

Remaining output components keep unchanged. The training loss is perturbed as follows,

$$\delta R' = \frac{1}{N} \sum_{i=1}^{N} u_i \delta o_1'\left(x_i\right) = \frac{1}{N} \delta b \cdot \sum_{j=1}^{P_{L-2}} w^j \sum_{i \in \left\{1,2,\cdots,N | v_i^j > \eta\right\}} u_i. \tag{62}$$

Because $\eta$ is the biggest split threshold among all locations, only for the $h$th location, there is $\sum_{i \in \{1,2,\cdots,N | v_i^j > \eta\}} u_i \neq 0$. We then have

$$\delta R' = \frac{1}{N} \delta b \cdot w^h \cdot \sum_{i=1}^{n} u_i, \tag{63}$$

which is the same as (52). By setting $\delta b$ with appropriate sign, we can obtain $\delta R' < 0$ and conclude that $\theta$ is a spurious local minimum.

If a perturbation of $W^L(1, \cdot)$ is required to split data samples as shown in Lemma 11, the training loss at $\theta$ is perturbed as

$$R\left(\theta + \delta\theta\right) = R\left(\theta\right) + \frac{\partial R\left(\theta\right)}{\partial W^L(1, \cdot)} \delta W^L(1, \cdot) = R\left(\theta\right) \tag{64}$$

due to the optimality condition $\frac{\partial R(\theta)}{\partial W^L(1,\cdot)} = \mathbf{0}^T$ since $\theta$ is a local minimum. The perturbation $\delta\theta$ only modifies $W^L(1, \cdot)$ and does not change other parameters. Therefore, by starting from $\theta + \delta\theta$ and setting the parameters of the last three layers as in Lemma 9, we can obtain a point $\theta''$ with $R\left(\theta + \delta\theta\right) = R\left(\theta''\right)$. Thus we have $R\left(\theta\right) = R\left(\theta''\right)$. We then perturb the biases of $\theta''$ as in Lemma 10 to obtain a loss lower than $R\left(\theta\right)$, which shows that $\theta$ is a spurious local minimum.

$\square$

### G.5 PROOF OF LEMMA 3

*Proof.* We prove by induction. We will often perturb the network parameters to tune $o_i$ or $u_i$ ($i \in [N]$), which does not change empirical loss, as shown in (64).

Let us first consider some special cases.

### G.5.1 Special Cases: **N = 1** or **M = 1**

If there is only one data sample, i.e., $N = 1$, there will be only one group at each location. Then, by Lemma 11, we have $\mathbf{u} = (u_1) \neq 0$, thus Lemma 3 holds.

We then consider the case of only one location: $M = 1$. If there are some groups with $\sum_{i \in I_q^1} u_i \neq 0$, Lemma 3 already holds. If there exists a group consisting of a single sample with $u_i = 0$, according to Lemma 11, we can perturb network parameters to get $u_i' \neq 0$ and thus Lemma 3 holds. Otherwise, all groups have multiple samples and there are $\sum_{i \in I_q^1} u_i = 0$ for all groups, and by $R(\theta) > 0$ and consequently $\mathbf{u} \neq 0$, there must exist some group $k$ such that there exists $i \in I_k^1$ and $u_i \neq 0$. All samples in group $k$ have identical outputs, and by Lemma 11, we can separate this sample with $u_i \neq 0$ from other samples in this group by perturbation, i.e., $o_i' \neq o_j'$ ($\forall j \in I_k^1, j \neq i$). By doing so, we can obtain a isolated sample with $u_i' \neq 0$, and thus Lemma 3 is satisfied.

### G.5.2 Induction Hypothesis

Now we discuss the cases for $N \geq 2$ and $M > 1$, and use induction to prove.

In order to give our induction hypothesis, we first give some definitions. For multiple locations ($M > 1$), there are usually a number of groups at each location. Recall that $I_q^j$ denotes the set of indices of samples in the $q$th group of the $j$th location. We can select one group with $\sum_{i \in I_{q_j}^j} u_i$ (the sum of $u_i$s in this group) from each location $j$ and form a sequence $\sum_{i \in I_{q_1}^1} u_i, \sum_{i \in I_{q_2}^2} u_i, \cdots, \sum_{i \in I_{q_M}^M} u_i$. If all elements in this sequence are equal and nonzero, i.e., $\sum_{i \in I_{q_1}^1} u_i = \sum_{i \in I_{q_2}^2} u_i = \cdots = \sum_{i \in I_{q_M}^M} u_i \neq 0$, we call it a complete and identical sequence of groups (**CI sequence**). If all elements in a sequence are nonzero but not equal, or partial elements in this sequence equal to zero, we call it a **non-CI sequence**.

Given a CI sequence, after adding a sample $u_k$ into the group $q_j$ at each location $j$, if $\sum_{i \in I_{q_1}^1} u_i = \sum_{i \in I_{q_2}^2} u_i = \cdots = \sum_{i \in I_{q_M}^M} u_i = -u_k$, the sequence becomes $\sum_{i \in I_{q_1}^1} u_i = 0, \sum_{i \in I_{q_2}^2} u_i = 0, \cdots, \sum_{i \in I_{q_M}^M} u_i = 0$ (now the set $I_{q_j}^j$ ($j \in [M]$) contains the new sample index $k$). Therefore, if for all groups and all locations, there exists only one CI sequence, and there are $\sum_{i \in I_{q_j}^j} u_i = 0$ for all groups not in this CI sequence, then it is possible that for all $j \in [M]$ and $q \in [g_j]$, $\sum_{i \in I_q^j} u_i = 0$ after adding a new sample that can annihilate this CI sequence, thus Lemma 3 is violated.

In order to prove Lemma 3, it suffices to prove that for every possible group configuration groups with $\sum_{i \in I_q^j} u_i \neq 0$ exist after removing CI sequences. Sometimes only CI sequences exist, like $(u_i) - (u_i) - \cdots - (u_i)$ ($i \in [N]$), and when adding new samples, there is a risk of annihilating such CI sequences. We can avoid such situations using perturbations of network parameters (this will be explained in sections G.5.3 and G.5.4). Consequently, we give the following induction hypothesis.

**For multiple locations, either non-CI sequences exist after removing each CI sequence (setting each element $\sum_{i \in I_{q_j}^j} u_i$ in it to zero), or the removal of CI sequences can be prevented.**

By this induction hypothesis, after adding a new sample $\mathbf{x}_{m+1}$ into one existing group at each location, groups with $\sum_{i \in I_{q_j}^j} u_i \neq 0$ still exist, and Lemma 3 holds accordingly.

As base cases, we will prove that this induction hypothesis holds for $N = 2$. We then prove that if it holds for $N = m$ ($m \geq 2$), it will also hold when adding a new sample $\mathbf{x}_{m+1}$, as induction step.

### G.5.3 Base Cases: $N = 2$

$M = 2$. First, we discuss the cases of two locations. When $N = 2$, all possible group configurations are discussed in the following and we will prove that induction hypothesis holds for each case.

CASE 1.   Suppose the groups at two locations are

$$\begin{aligned} \text{location } 1 &: (u_1, u_2), \\ \text{location } 2 &: (u_1, u_2). \end{aligned} \tag{65}$$

If $u_1 + u_2 \neq 0$, then the sequence $(u_1, u_2) - (u_1, u_2)$ (the first element in the sequence is $(u_1, u_2)$ from location 1, and the second one is $(u_1, u_2)$ from location 2) is a CI sequence. However, this CI sequence cannot be vanished by adding a new sample since we can use perturbation of network parameters to prevent it from happening, described later in the induction step.

If $u_1 + u_2 = 0$, by Lemma 11, we can perturb network parameters to obtain $u_1' + u_2' = 0$ at first.

CASE 2.   Suppose the groups at two locations are

$$\begin{aligned} \text{location } 1 &: (u_1, u_2), \\ \text{location } 2 &: (u_1), (u_2), \end{aligned} \tag{66}$$

where groups on the left have bigger $v_i^j$s than those on the right at each location.

By Lemma 11, we can perturb network parameters to obtain $u_1' \neq 0$ and $u_2' \neq 0$ for squared loss and cross-entropy loss, then the sequences $(u_1', u_2') - (u_1')$ and $(u_1', u_2') - (u_2')$ are both non-CI sequences, thus the induction hypothesis holds. If $(u_1, u_2)$ in location 1 is separated as $(u_1'), (u_2')$ after perturbation, then it can be treated by the following case 3.

CASE 3.   Suppose the groups at two locations are

$$\begin{aligned} \text{location } 1 &: (u_1), (u_2) \\ \text{location } 2 &: (u_1), (u_2). \end{aligned} \tag{67}$$

or

$$\begin{aligned} \text{location } 1 &: (u_1), (u_2) \\ \text{location } 2 &: (u_2), (u_1). \end{aligned} \tag{68}$$

For these cases, by Lemma 11, if $u_1 = u_2$, we can perturb the parameters to obtain $u_1' \neq u_2'$ for squared loss and cross-entropy loss, then the sequences $(u_1') - (u_2')$ and $(u_2') - (u_1')$ are both non-CI sequences.

The sequences $(u_1) - (u_1)$ and $(u_2) - (u_2)$ are CI sequences. Again, by appropriate perturbations when adding new samples in the induction step, such CI sequences will not be removed and thus Lemma 3 holds.

$M \geq 3$.   If there are more than two locations, we can select two locations as a subset, which must belong to one of the above three possible cases. The full sequences of all locations contain those sequences in this two-location subset as sub-sequences, and consequently groups with $\sum_{i \in I_{q_j}^j} u_i \neq 0$ still exist in full sequences. Therefore, induction hypothesis holds for $M \geq 3$.

### G.5.4   INDUCTION STEP

At last, we prove that if the induction hypothesis holds for $N = m$ samples, then it holds for $N = m + 1$ samples. Suppose sample $\mathbf{x}_{m+1}$ is to be added given current arrangement of groups. Note that we can have $u_{m+1} \neq 0$ according to Lemma 11. There are the following three possible situations.

**1)**   $\mathbf{x}_{m+1}$ is added as a new group at every location.

This is equivalent to adding a new CI sequence. CI and non-CI sequences formed by previous groups still exist, thus induction hypothesis holds.

**2)**   $\mathbf{x}_{m+1}$ is inserted into an existing group at every location. More specifically, this can be divided into the following four cases.

A)  $\mathbf{x}_{m+1}$ is inserted into an existing group satisfying $\sum_{i \in I_q^j} u_i = 0$ at every location.

This is equivalent to adding a new CI sequence into existing groups. Since CI or non-CI sequences still exist, the induction hypothesis holds.

B)  $\mathbf{x}_{m+1}$ is inserted into existing CI or non-CI sequences.

If inserting $\mathbf{x}_{m+1}$ into an existing CI sequence, it will either be a new CI sequence, or annihilated by this addition. By induction hypothesis, non-CI sequences may still exist, hence induction hypothesis holds again after inserting $\mathbf{x}_{m+1}$. Another possible situation is that CI sequences exist, such as $(u_i) - (u_i) - \cdots - (u_i)$ $(i \in [N])$, and when adding new samples, there is a risk of annihilating such CI sequences, and if no remaining non-CI sequences exist, then no groups with $\sum_{i \in I_q^j} u_i \neq 0$ exist. We can perturb the outputs of samples to avoid such situation using a perturbation of network parameters. For example, if $u_1 + u_3 = 0$, then adding $u_3$ into the sequence $(u_1) - (u_1) - \cdots - (u_1)$ will cause its disappearance. We will have $o_1 = o_3$ since sample 1 and sample 3 always appear in the same group at every location after the addition. After perturbation that tunes $o_1$ and $o_3$ differently to get $o_1' \neq o_3'$, as shown in Lemma 11, sample 1 and sample 3 then cannot appear in the same group at every location. The annihilation of sequence $(u_1) - (u_1) - \cdots - (u_1)$ is thus avoided and induction hypothesis holds.

If inserting $\mathbf{x}_{m+1}$ into an existing non-CI sequence, it will still be a non-CI sequence by definition. Therefore, induction hypothesis holds.

C)  $\mathbf{x}_{m+1}$ is inserted into an existing CI sequence at some locations and into an existing non-CI sequence at remaining locations.

The involved CI sequence becomes a non-CI one, and the involved non-CI sequence may become a non-CI sequence, a CI sequence or a zero sequence. Anyway, non-CI sequences still exist and cannot be removed by adding a sample, and thus induction hypothesis holds.

D)  $\mathbf{x}_{m+1}$ is inserted into an existing CI or non-CI sequence at some locations and into groups with $\sum_{i \in I_q^j} u_i = 0$ at remaining locations.

In this case, non-CI sequences will exist after inserting $\mathbf{x}_{m+1}$, thus induction hypothesis holds.

**3)**  $\mathbf{x}_{m+1}$ is inserted into existing groups at some locations $j \in J_1$, and added as new groups at remaining locations $j \in J_2$. Equivalently, we can regard new groups in $J_2$ (before adding $\mathbf{x}_{m+1}$) as existing groups with $\sum_{i \in I_q^j} u_i = 0$, thus the above case D) can be applied directly, and induction hypothesis is maintained.

### G.5.5  INDUCTION CONCLUSION

We have proved that if the induction hypothesis holds for $N = m$ samples, then it holds for $N = m+1$ samples. By induction hypothesis, there exists $j \in [M], q \in [g_j]$, $\sum_{i \in I_q^j} u_i \neq 0$, and consequently a split of data samples somewhere in feature maps is always possible.

$\square$

### G.6  PROOF OF LEMMA 12

*Proof.* For cross-entropy loss, the training loss for a sample $(\mathbf{x}_i, \mathbf{y}_i)$ is

$$l\left(\mathbf{o}(\mathbf{x}_i), \mathbf{y}_i\right)) = -\sum_{j=1}^{d_y} y_{i,j} \log \frac{e^{-o_{i,j}}}{\sum_{k=1}^{d_y} e^{-o_{i,k}}}, \tag{69}$$

where $o_{i,j}$ is the $j$th component of $\mathbf{o}_i$. Its derivative is

$$u_i = \frac{\partial l\left(\mathbf{o}(\mathbf{x}_i), \mathbf{y}_i\right)}{\partial o_{i,1}} = \frac{e^{-o_{i,1}}}{\sum_{k=1}^{d_y} e^{-o_{i,k}}} - y_{i,1}. \tag{70}$$

Let $p_i = \frac{e^{-o_{i,1}}}{\sum_{k=1}^{d_y} e^{-o_{i,k}}}$, we have

$$u_i = p_i - y_{i,1}. \tag{71}$$

Denote $\mathbf{w}^T := W^L(1, \cdot)$, $o := o_{i,1}$. Under the perturbation $\delta\mathbf{w}$, by Taylor expansion we have

$$\begin{aligned}
u_i' = u_i &+ \frac{\partial u_i}{\partial o}\left[(I^{L-1,i}W^{L-1}\cdots I^{1,i}W^1\mathbf{x}_i)\cdot\delta\mathbf{w}\right] \\
&+ \frac{1}{2}\frac{\partial^2 u_i}{\partial o^2}\left[(I^{L-1,i}W^{L-1}\cdots I^{1,i}W^1\mathbf{x}_i)\cdot\delta\mathbf{w}\right]^2 + O(\|\delta\mathbf{w}\|^3).
\end{aligned} \tag{72}$$

Note that for $\mathbf{o}^{L-1,i} = I^{L-1,i}W^{L-1}\cdots I^{1,i}W^1\mathbf{x}_i$, we have $\mathbf{o}^{L-1,i}(1:2P_{L-2}) = 0$ $(\forall i \in [N])$ due to the parameter setting in Lemma 6, thus perturbation $\delta\mathbf{w}(1:2P_{L-2})$ has no effect in (72).

For cross-entropy loss, the first and second order derivatives of $u_i$ are

$$\frac{\partial u_i}{\partial o} = p_i - p_i^2, \tag{73}$$

$$\frac{\partial^2 u_i}{\partial o^2} = \frac{\partial^2 u_i}{\partial o \partial p_i}\frac{\partial p_i}{\partial o} = (1 - 2p_i)\frac{\partial u_i}{\partial o}, \tag{74}$$

respectively.

Let $Q^i := I^{L-1,i}W^{L-1}\cdots I^{1,i}W^1$, we now prove that if $Q^i\mathbf{x}_i = \alpha Q^j\mathbf{x}_j$ ($\alpha \neq 0, \alpha \neq 1$), then by setting $\delta\mathbf{w}$ appropriately, one can always make $u_i' \neq u_j'$. Note that $\frac{\partial u_i}{\partial o} = p_i - p_i^2 \neq 0$ since $p_i \neq 0$ and $p_i \neq 1$ for finite inputs. Also note the requirement $I^{L-1,i} \neq 0$ in Lemma 6, we have $Q^i\mathbf{x}_i \neq \mathbf{0}$, $i \in [N]$. We can set $\delta\mathbf{w} = \epsilon\frac{Q^i\mathbf{x}_i}{\|Q^i\mathbf{x}_i\|}$ such that $(Q^i\mathbf{x}_i)\cdot\delta\mathbf{w} \neq 0$ and $(Q^j\mathbf{x}_j)\cdot\delta\mathbf{w} \neq 0$, where $\epsilon$ is a sufficiently small positive number. By (72), if $\frac{\partial u_i}{\partial o} \neq \frac{1}{\alpha}\frac{\partial u_j}{\partial o}$, then $\frac{\partial u_i}{\partial o}(Q^i\mathbf{x}_i)\cdot\delta\mathbf{w} \neq \frac{\partial u_j}{\partial o}(Q^j\mathbf{x}_j)\cdot\delta\mathbf{w}$, hence we have $u_i' \neq u_j'$ up to the first-order Taylor expansion.

If $Q^i\mathbf{x}_i = \alpha Q^j\mathbf{x}_j$ and $\frac{\partial u_i}{\partial o} = \frac{1}{\alpha}\frac{\partial u_j}{\partial o}$, we need to consider the second-order term in (72), and if $\frac{\partial^2 u_i}{\partial o^2}\alpha^2 \neq \frac{\partial^2 u_j}{\partial o^2}$, we have $u_i' \neq u_j'$ up to the second-order Taylor approximation. Otherwise, we have $\frac{\partial u_i}{\partial o} = \frac{1}{\alpha}\frac{\partial u_j}{\partial o}$ and $\frac{\partial^2 u_i}{\partial o^2}\alpha^2 = \frac{\partial^2 u_j}{\partial o^2}$, which result in

$$(p_i - p_i^2)\alpha = p_j - p_j^2 \tag{75}$$

and

$$(1 - 2p_i)\alpha = 1 - 2p_j, \tag{76}$$

respectively.

We next show that (75) and (76) cannot both hold for each case of $y_{i,1} \in \{0, 1\}$, and thus we have $u_i' \neq u_j'$ up to either the first-order or the second-order Taylor approximation.

If $u_i = p_i$, $u_j = p_j$ or $u_i = p_i - 1$, $u_j = p_j - 1$, then $u_i = u_j$ implies $p_i = p_j$. From (75) and $\alpha \neq 1$, contradiction is resulted and hence $u_i' \neq u_j'$. If samples $\mathbf{x}_i$ and $\mathbf{x}_j$ have distinct labels, say without loss of generality, $u_i = p_i - 1$, $u_j = p_j$, then $u_i = u_j$ leads to $p_j = p_i - 1$. Substituting it into (75) and (76), we obtain

$$(p_i - p_i^2)\alpha = (p_i - p_i^2) + 2(p_i - 1) \tag{77}$$

and

$$(1 - 2p_i)\alpha = (1 - 2p_i) + 2, \tag{78}$$

respectively. (77) and (78) can be transformed into

$$p_i(1 - p_i)\alpha = (1 - p_i)(p_i - 2) \tag{79}$$

and
$$(1 - p_i)(2\alpha - 2) = \alpha + 1, \tag{80}$$
respectively. By $1 - p_i \neq 0$ , (79) results in
$$p_i \alpha = p_i - 2. \tag{81}$$
Solving (80) and (81), we obtain $\alpha = -1, p_i = 1$. However, $p_i = 1$ is impossible for finite inputs, therefore, we have $u'_i \neq u'_j$ after the perturbation.

$\square$

### G.7 PROOF OF LEMMA 13

*Proof.* We now prove that if $u_i + u_j = 0$ and $Q^i \mathbf{x}_i = \alpha Q^j \mathbf{x}_j$ ($\alpha \neq 0, \alpha \neq 1$), then by setting $\delta \mathbf{w}$ appropriately, one can always obtain $u'_i + u'_j \neq 0$.

By (72), if $\frac{\partial u_i}{\partial o} \neq -\frac{1}{\alpha} \frac{\partial u_j}{\partial o}$, then $\frac{\partial u_i}{\partial o}(Q^i \mathbf{x}_i) \cdot \delta \mathbf{w} \neq -\frac{\partial u_j}{\partial o}(Q^j \mathbf{x}_j) \cdot \delta \mathbf{w}$, hence $u'_i + u'_j \neq 0$ up to the first-order Taylor expansion.

If $Q^i \mathbf{x}_i = \alpha Q^j \mathbf{x}_j$ and $\frac{\partial u_i}{\partial o} = -\frac{1}{\alpha} \frac{\partial u_j}{\partial o}$, we need to consider the second-order term in (72), and if $\frac{\partial^2 u_i}{\partial o^2} \alpha^2 \neq -\frac{\partial^2 u_j}{\partial o^2}$, we have $u'_i + u'_j \neq 0$ up to the second-order Taylor approximation. Otherwise, we have $\frac{\partial u_i}{\partial o} = -\frac{1}{\alpha} \frac{\partial u_j}{\partial o}$ and $\frac{\partial^2 u_i}{\partial o^2} \alpha^2 = -\frac{\partial^2 u_j}{\partial o^2}$, which result in
$$-(p_i - p_i^2)\alpha = p_j - p_j^2 \tag{82}$$
and
$$-(1 - 2p_i)\alpha + (1 - 2p_j) = 0, \tag{83}$$
respectively.

We next show that (82) and (83) cannot both hold for every case of $y_{i,1} \in \{0, 1\}$.

If $u_i = p_i$, $u_j = p_j$, then $u_i + u_j = 0$ implies $p_j = -p_i$. Substituting it into (82) and (83) and with simple calculations, we can obtain
$$-p_i \alpha - p_i = 1 - \alpha \tag{84}$$
and
$$2p_i \alpha + 2p_i = \alpha - 1. \tag{85}$$
The solution is $\alpha = 1, p_i = 0$, which contradict $\alpha \neq 1, p_i \neq 0$.

If $u_i = p_i - 1$, $u_j = p_j - 1$, then $u_i + u_j = 0$ implies $p_j = 2 - p_i$. In this case, the solution to (82) and (83) is $\alpha = 1, p_i = 1$, again contradiction is resulted.

If samples $\mathbf{x}_i$ and $\mathbf{x}_j$ have distinct labels, say without loss of generality, $u_i = p_i - 1$, $u_j = p_j$, then $u_i + u_j = 0$ leads to $p_j = 1 - p_i$. The solution to (82) and (83) will be $\alpha = -1$ and $p_i$ can be arbitrary. However, $\alpha = -1$ implies $Q^i \mathbf{x}_i = -Q^j \mathbf{x}_j$. Due to $I^{L-1,i} \neq 0$ by the requirement in Lemma 6, we have for all $i \in [N]$ that all components of $Q^i \mathbf{x}_i$ are nonnegative and some components of it must be positive, and thus $Q^i \mathbf{x}_i = -Q^j \mathbf{x}_j$ leads to contradiction. Therefore, $\alpha = -1$ is impossible.

In summary, for various cases of $y_{i,1} \in \{0, 1\}$, we all have $u'_i + u'_j \neq 0$.

$\square$

### G.8 PROOF OF LEMMA 11

*Proof.*

1. Proof of $o'_i \neq o'_j$ ($i, j \in [N]$, $i \neq j$) for squared loss and cross-entropy loss.

We perturb $W^L(1, \cdot)$ to make $o'_i \neq o'_j$. Under this perturbation, we have
$$o'_i = o_i + (I^{L-1,i} W^{L-1} \cdots I^{1,i} W^1 \mathbf{x}_i) \cdot \delta \mathbf{w} = o_i + (Q^i \mathbf{x}_i) \cdot \delta \mathbf{w}. \tag{86}$$
Since $\mathbf{o}^{L-1,i} \neq \mathbf{o}^{L-1,j}$ by Assumption 1, we have $Q^i \mathbf{x}_i \neq Q^j \mathbf{x}_j$. Then, if $o_i = o_j$, we have $o'_i - o'_j = (Q^i \mathbf{x}_i - Q^j \mathbf{x}_j) \cdot \delta \mathbf{w}$. Let $\delta \mathbf{w} = \epsilon(Q^i \mathbf{x}_i - Q^j \mathbf{x}_j)$, where $\epsilon$ is a sufficiently small positive number, there is $o'_i - o'_j = \epsilon \left\| Q^i \mathbf{x}_i - Q^j \mathbf{x}_j \right\|^2 \neq 0$. Thus, $o'_i \neq o'_j$.

2. Proof of $u_i' \neq 0$ $(i \in [N])$.

For squared loss, we perturb $W^L(1, \cdot)$ to make $u_i' \neq 0$ if $u_i = 0$. With this perturbation, we have

$$u_i' = u_i + \frac{\partial u_i}{\partial o} \left[ (I^{L-1,i} W^{L-1} \cdots I^{1,i} W^1 \mathbf{x}_i) \cdot \delta \mathbf{w} \right]. \tag{87}$$

We have $u_i = o_i - y_{i,1}$ for squared loss $l(\mathbf{o}_i, \mathbf{y}_i) = \frac{1}{2} \|\mathbf{o}_i - \mathbf{y}_i\|^2$, then

$$u_i' = (Q^i \mathbf{x}_i) \cdot \delta \mathbf{w}. \tag{88}$$

Setting $\delta \mathbf{w} = \epsilon(Q^i \mathbf{x}_i)$ results in $u_i' = \epsilon \|Q^i \mathbf{x}_i\|^2$. By $Q^i \mathbf{x}_i \neq \mathbf{0}$ due to the requirement $I^{L-1,i} \neq 0$ in Lemma 6, we have $u_i' \neq 0$.

For cross-entropy loss function, the derivative $u_i$ is given in (70). For finite inputs and $y_{i,1} \in \{0,1\}$, we have $\frac{e^{-o_{i,1}}}{\sum_{k=1}^{d_y} e^{-o_{i,k}}} \neq 0$ and $\frac{e^{-o_{i,1}}}{\sum_{k=1}^{d_y} e^{-o_{i,k}}} \neq 1$, thus $u_i \neq 0$.

3. Proof of $u_i' \neq u_j'$ or $u_i' + u_j' \neq 0$ $(i,j \in [N], i \neq j)$.

For squared loss, we perturb $W^L(1, \cdot)$ to make $u_i' \neq u_j'$ or $u_i' + u_j' \neq 0$. Under this perturbation, if $u_i = u_j$, by (87) we have $u_i' - u_j' = (Q^i \mathbf{x}_i - Q^j \mathbf{x}_j) \cdot \delta \mathbf{w}$. Let $\delta \mathbf{w} = \epsilon(Q^i \mathbf{x}_i - Q^j \mathbf{x}_j)$, there is $u_i' - u_j' = \epsilon \|Q^i \mathbf{x}_i - Q^j \mathbf{x}_j\|^2 \neq 0$. Therefore, $u_i' \neq u_j'$ $(i,j \in [N], i \neq j)$.

If $u_i + u_j = 0$, we have $u_i' + u_j' = (Q^i \mathbf{x}_i + Q^j \mathbf{x}_j) \cdot \delta \mathbf{w}$. Note that $Q^i \mathbf{x}_i \neq -Q^j \mathbf{x}_j$ due to $I^{L-1,i} \neq 0$ and $I^{L-1,j} \neq 0$ by the requirement in Lemma 6. Let $\delta \mathbf{w} = \epsilon(Q^i \mathbf{x}_i + Q^j \mathbf{x}_j)$, we then have $u_i' + u_j' = \epsilon \|Q^i \mathbf{x}_i + Q^j \mathbf{x}_j\|^2 \neq 0$. Therefore, $u_i' + u_j' \neq 0$ $(i,j \in [N], i \neq j)$.

For cross-entropy loss function, recall that $\mathbf{w}^T := W^L(1, \cdot)$ and $Q^i := I^{L-1,i} W^{L-1} \cdots I^{1,i} W^1$. Under perturbation $\delta \mathbf{w}$, by the first-order Taylor expansion in (72), using (73) we have

$$u_i' = u_i + \frac{\partial u_i}{\partial o} \left[ (I^{L-1,i} W^{L-1} \cdots I^{1,i} W^1 \mathbf{x}_i) \cdot \delta \mathbf{w} \right] = u_i + (p_i - p_i^2) \left[ (Q^i \mathbf{x}_i) \cdot \delta \mathbf{w} \right]. \tag{89}$$

Note that $Q^i \mathbf{x}_i \neq \mathbf{0}$, $Q^j \mathbf{x}_j \neq \mathbf{0}$, and $p_i - p_i^2 \neq 0$ since $p_i \neq 0$ and $p_i \neq 1$. If $Q^i \mathbf{x}_i$ is not parallel to $Q^j \mathbf{x}_j$, then $(p_i - p_i^2)(Q^i \mathbf{x}_i) \pm (p_j - p_j^2)(Q^j \mathbf{x}_j) \neq \mathbf{0}$. Therefore, we have $u_i' \neq u_j'$ or $u_i' + u_j' \neq 0$ $(i,j \in [N], i \neq j)$, respectively.

If $Q^i \mathbf{x}_i$ is parallel to $Q^j \mathbf{x}_j$, i.e., $Q^i \mathbf{x}_i = \alpha Q^j \mathbf{x}_j$ $(\alpha \neq 0)$, we can obtain $u_i' \neq u_j'$ or $u_i' + u_j' \neq 0$ $(i,j \in [N], i \neq j)$ by Lemma 12 or Lemma 13, respectively. $\alpha = 1$ is not allowed by Assumption 1.

4. $\mathbf{u} = (u_1, u_2, \cdots, u_N)^T \neq \mathbf{0}$ if $R = \frac{1}{N} \sum_{i=1}^N l(\mathbf{o}_i, \mathbf{y}_i) > 0$.

For squared loss, $u_i = o_i - y_{i,1}$. If $\mathbf{o}_i - \mathbf{y}_i = 0$ for every sample $\mathbf{x}_i$ $(i \in [N])$, then the empirical loss $R = \frac{1}{N} \sum_{i=1}^N l(\mathbf{o}_i, \mathbf{y}_i) = \frac{1}{2N} \sum_{i=1}^N \|\mathbf{o}_i - \mathbf{y}_i\|^2 = 0$, contradicting the assumption $R > 0$. Therefore, there must be some nonzero components in vector $(\mathbf{o}_i - \mathbf{y}_i)$ for some $i \in [N]$. Without loss of generality, let some components in $(o_1 - y_{1,1}, o_2 - y_{2,1}, \cdots, o_N - y_{N,1})$ be nonzero, hence $\mathbf{u} \neq \mathbf{0}$.

For cross-entropy loss, according to (70), $u_i \neq 0$ $(\forall i \in [N])$ for finite inputs, thus $\mathbf{u} = (u_1, u_2, \cdots, u_N)^T \neq \mathbf{0}$.

5. Nonzero gaps between $o_i'$s or $u_i'$s can still exist after subsequent perturbations, for squared loss and cross-entropy loss.

We discuss the gaps between $o_i'$s in detail. Given some samples with identical outputs, say without loss of generality, $o_1 = o_2 = \cdots = o_s$, by Lemma 11 we can use perturbation of parameters $\delta \mathbf{w}^T := \delta W^L(1, \cdot)$ to obtain $o_1' \neq o_2'$, thus a gap exists between $o_1'$ and $o_2'$. Without loss of generality, suppose after the first perturbation, the outputs of samples are ordered as $o_1' > o_2' > \cdots > o_n' = o_{n+1}' = \cdots = o_{n+m}' > o_{n+m+1}' > \cdots > o_s'$. We now want to perturb the outputs of samples again to avoid $o_n' = o_{n+1}'$ using $\delta \mathbf{w}'$.

Under the second perturbation $\delta \mathbf{w}'$, the output of each sample is $o_i'' = (\mathbf{w}' + \delta \mathbf{w}') \cdot Q^i \mathbf{x}_i := o_i' + \delta \mathbf{w}' \cdot Q^i \mathbf{x}_i$. The gaps between samples are changed to $o_i'' - o_{i+1}'' = o_i' - o_{i+1}' + \delta \mathbf{w}' \cdot (Q^i \mathbf{x}_i - $

$Q^{i+1}\mathbf{x}_{i+1})$ $(i \in [s-1])$. We set $\delta\mathbf{w}'$ small enough as follows to make sure the nonzero gaps in $o_i' - o_{i+1}'$ $(i \in [s-1])$ do not disappear.

For the gap $o_i'' - o_{i+1}''$, we have $o_i'' - o_{i+1}'' \geq o_i' - o_{i+1}' - \|\delta\mathbf{w}'\| \|Q^i\mathbf{x}_i - Q^{i+1}\mathbf{x}_{i+1}\|$ $(i \in [s-1])$. We already have $o_i' - o_{i+1}' > 0$ for nonzero gaps, and if $\delta\mathbf{w}'$ are small enough such that

$$\forall i \in [s-1], \; o_i' - o_{i+1}' - \|\delta\mathbf{w}'\| \|Q^i\mathbf{x}_i - Q^{i+1}\mathbf{x}_{i+1}\| > 0, \tag{90}$$

or equivalently

$$\forall i \in [s-1], \; \|\delta\mathbf{w}'\| < \min_{(i\in[s-1],\, o_i'-o_{i+1}'>0)} \frac{o_i' - o_{i+1}'}{\|Q^i\mathbf{x}_i - Q^{i+1}\mathbf{x}_{i+1}\|}, \tag{91}$$

then we will have $\forall i \in [s-1], \; o_i'' - o_{i+1}'' > 0$. Note that $\|Q^i\mathbf{x}_i - Q^{i+1}\mathbf{x}_{i+1}\| \neq 0$. The nonzero gaps between neighboring samples still exist.

Additional perturbations can be treated in the same spirit. For a dataset with $N$ samples, at most $N$ perturbations are needed. The total perturbation is obtained by adding the perturbations in all steps, i.e., $\delta\mathbf{w} + \delta\mathbf{w}' + \cdots$.

The gaps between $u_i'$s can be treated similarly using small enough perturbations.

$\square$

## G.9 PROOF OF THEOREM 3

*Proof.* The parameter matrix $W^1(1:P_0, 1:P_0)$ of the first convolutional layer can be implemented by a filter like $\mathbf{w}_1^1 = (0,0,1,0,0)$ (for $s_0 = 5$). With the parameter setting in Theorem 3, the output of the $k$th neuron in the first convolution layer is

$$o_k^1 = \sigma\left(x_i^k + c_p\right), \; \forall i \in [N] \tag{92}$$

Then, we have

$$o_k^1(\mathbf{x}_i) = \sigma\left(x_i^k + c_p\right) = x_i^k + c_p > 0, \; \forall i \in I \tag{93}$$

due to (10). The outputs of remaining neurons in the first feature map of the first convolutional layer do not matter since they will be annihilated later by $W^L$. Given $x_i^k + c_p < 0$ $(\forall i \notin I)$, we have

$$o_k^1(\mathbf{x}_i) = 0, \; i \in [N], i \notin I. \tag{94}$$

The filters and biases of higher convolutional layers $(l \in \{2, 3, \cdots, L-1\})$ for the first predictor propagate $o^l(1:P_{l-1})$ (the outputs in the first feature map of each layer) forward without changing them.

Sufficiently large positive constants $c_l$ $(l \in [L-1])$ are added to the biases of each layer such that the ReLU units in the second subnetwork are all activated and their outputs are not truncated. The first convolutional layer will be added $c_1$ to each of its output neuron, and the change of outputs of higher convolutional layers can be recursively computed as

$$\begin{aligned} \Delta o^1(i) &= c_1, \; i \in [P_0 + 1 : n_1], \\ \Delta o^l(i) &= \sum_{j \in [P_{l-2}+1:n_{l-1}]} W^l(i,j)\Delta o^{l-1}(j) + c_l, \; i \in [P_{l-1}+1 : n_l], \, l \in \{2, 3, \cdots, L-1\}, \\ \Delta o^L(i) &= \sum_{j \in [P_{L-2}+1:n_{L-1}]} W^L(i,j)\Delta o^{L-1}(j), \; i \in [d_y]. \end{aligned} \tag{95}$$

The effect of introducing $c_l$ $(l \in [L-1])$ is cancelled at the output layer by subtracting $\Delta o^L(i)$ from $o^L(i)$ $(i \in [d_y])$. This is equivalent to subtracting $\Delta o^L(i)$ from $b_i^L$.

By the parameter setting, the output of each sample is

$$\mathbf{o}(\mathbf{x}_i) = W_2\hat{\mathbf{x}}_i + \sigma\left(x_i^k + c_p\right) \cdot \left(\lambda_1, \lambda_2, \cdots, \lambda_{d_y}\right)^T, \; \forall i \in [N], \tag{96}$$

where the expression of $W_2\hat{\mathbf{x}}_i$ is given in (20). This implies

$$\mathbf{o}\left(\mathbf{x}_i\right) = W_2\hat{\mathbf{x}}_i + \left(x_i^k + c_p\right)\left(\lambda_1, \lambda_2, \cdots, \lambda_{d_y}\right)^T, \quad i \in I, \tag{97}$$

and

$$\mathbf{o}\left(\mathbf{x}_i\right) = \tilde{\mathbf{y}}_i, \quad l\left(\mathbf{o}\left(\mathbf{x}_i\right), \mathbf{y}_i\right) = 0, \quad i \in I \tag{98}$$

if there is a single index in set $I$, where $\tilde{\mathbf{y}}_i$ is defined in section 4. If there are multiple elements in set $I$,

$$\sum_{i \in I} l\left(\mathbf{o}\left(\mathbf{x}_i\right), \mathbf{y}_i\right) = \sum_{i \in I} l\left(W_2\hat{\mathbf{x}}_i + \left(x_i^k + c_p\right)\left(\lambda_1, \lambda_2, \cdots, \lambda_{d_y}\right)^T, \mathbf{y}_i\right) \tag{99}$$

is minimized.

For samples $\{\mathbf{x}_i, \ i \notin I\}$, we have

$$\mathbf{o}\left(\mathbf{x}_i\right) = W_2\hat{\mathbf{x}}_i, \forall i \in [N], \ i \notin I. \tag{100}$$

We now show that for the CNN given in Theorem 3, $\theta_1 = \left(W^l, \mathbf{b}^l\right)_{l=1}^L$ is a local minimum. This can be seen by perturbing the parameters and showing the non-decreaseness of loss. The training loss is

$$\begin{aligned} R &= \frac{1}{N}\sum_{I=1}^N l\left(\mathbf{o}\left(\mathbf{x}_i\right), \mathbf{y}_i\right) \\ &= \frac{1}{N}\sum_{i \in I} l\left(W_2\hat{\mathbf{x}}_i + \left(x_i^k + c_p\right)\left(\lambda_1, \lambda_2, \cdots, \lambda_{d_y}\right)^T, \mathbf{y}_i\right) + \frac{1}{N}\sum_{\substack{i \in [N], \\ i \notin I}} l\left(W_2\hat{\mathbf{x}}_i, \mathbf{y}_i\right). \end{aligned} \tag{101}$$

Under perturbation $\left(\delta W^l, \delta \mathbf{b}^l\right)_{l=1}^L$, since there is a gap between $x_i^k$ ($i \in [N]$) and the threshold $\frac{1}{2}\left(x_{i_*}^k + x_{j_*}^k\right)$, the ReLU unit in (92) keeps its activation status for every sample when the perturbation of $b_1^1 = c_p$ is sufficiently small. Therefore, the output of each sample is still in the form of (96). Accordingly, if there are multiple elements in set $I$,

$$\begin{aligned} R' :=&\frac{1}{N}\sum_{i=1}^N l\left(\mathbf{o}\left(\mathbf{x}_i\right) + \delta\mathbf{o}\left(\mathbf{x}_i\right), \mathbf{y}_i\right) \\ =&\frac{1}{N}\sum_{i \in I} l\left(W_2\hat{\mathbf{x}}_i + \left(x_i^k + c_p\right)\left(\lambda_1, \lambda_2, \cdots, \lambda_{d_y}\right)^T + \delta\mathbf{o}\left(\mathbf{x}_i\right), \mathbf{y}_i\right) \\ &+\frac{1}{N}\sum_{\substack{i \in [N] \\ i \notin I}} l\left(W_2\hat{\mathbf{x}}_i + \delta\mathbf{o}\left(\mathbf{x}_i\right), \mathbf{y}_i\right), \end{aligned}$$

where $\delta\mathbf{o}\left(\mathbf{x}_i\right)$ $(i \in I)$ and $\delta\mathbf{o}\left(\mathbf{x}_i\right)$ $(i \notin I)$ are caused by the perturbations of parameters in both $W_1$ and $W_2$. However, since $\left(\lambda_1, \lambda_2, \cdots, \lambda_{d_y}\right)$ minimize $\sum_{i \in I} l\left(W_2\hat{\mathbf{x}}_i + \left(x_i^k + c_p\right)\left(\lambda_1, \lambda_2, \cdots, \lambda_{d_y}\right)^T, \mathbf{y}_i\right)$, and the space of $\mathbf{o}\left(\mathbf{x}_i\right)$ has been fully explored by $\left(\lambda_1, \lambda_2, \cdots, \lambda_{d_y}\right)$ during minimization, we have

$$\begin{aligned} &\sum_{i \in I} l\left(W_2\hat{\mathbf{x}}_i + \left(x_i^k + c_p\right)\left(\lambda_1, \lambda_2, \cdots, \lambda_{d_y}\right)^T + \delta\mathbf{o}\left(\mathbf{x}_i\right), \mathbf{y}_i\right) \\ &\geq \sum_{i \in I} l\left(W_2\hat{\mathbf{x}}_i + \left(x_i^k + c_p\right)\left(\lambda_1, \lambda_2, \cdots, \lambda_{d_y}\right)^T, \mathbf{y}_i\right). \end{aligned} \tag{102}$$

On the other hand, since the size of each convolutional layer satisfies $n_l - P_{l-1} \geq d_y$ ($l \in [L-1]$), the matrix $\hat{W}^L(\cdot, P_{L-2}+1 : n_{L-1})\hat{W}^{L-1}(P_{L-2}+1 : n_{L-1}, P_{L-3}+1 : n_{L-2})\cdots\hat{W}^1(P_0 + 1 : n_1, \cdot) \in \mathbb{R}^{d_y \times d_x}$ is full rank and can generate the full space of $d_y \times d_x$ matrices around $\theta$, thus $\sum_{\substack{i \in [N], \\ i \notin I}} l\left(W_2\hat{\mathbf{x}}_i, \mathbf{y}_i\right)$ is minimized by $W_2$ in (20). Therefore,

$$\sum_{\substack{i \in [N], \\ i \notin I}} l\left(W_2\hat{\mathbf{x}}_i + \delta\mathbf{o}\left(\mathbf{x}_i\right), \mathbf{y}_i\right) \geq \sum_{\substack{i \in [N], \\ i \notin I}} l\left(W_2\hat{\mathbf{x}}_i, \mathbf{y}_i\right). \tag{103}$$

Using (102) and (103),

$$R' \geq \frac{1}{N} \sum_{i \in I} l\left(W_2\hat{\mathbf{x}}_i + \left(x_i^k + c_p\right)\left(\lambda_1, \lambda_2, \cdots, \lambda_{d_y}\right)^T, \mathbf{y}_i\right) + \frac{1}{N} \sum_{\substack{i \in [N], \\ i \notin I}} l\left(W_2\hat{\mathbf{x}}_i, \mathbf{y}_i\right)$$

$$= \frac{1}{N} \sum_{I=1}^{N} l\left(\mathbf{o}\left(\mathbf{x}_i\right), \mathbf{y}_i\right) = R. \tag{104}$$

For the case of single element in set $I$, $R' \geq R$ can be shown easily due to (98). Therefore, $\theta_1 = \left(W^l, \mathbf{b}^l\right)_{l=1}^{L}$ is a local minimum.

$\square$

