# OpenReview forum: "Spurious Local Minima Provably Exist for Deep Convolutional Neural Networks"
_ICLR.cc/2023/Conference — Submitted to ICLR 2023_

### Official Review · Reviewer_wvat · 2022-10-23

**Confidence:** 4
**Correctness:** 4
**Technical Novelty And Significance:** 2
**Empirical Novelty And Significance:** Not applicable
**Recommendation:** 3

**Clarity, Quality, Novelty And Reproducibility:**

-Clarity: This paper is generally well-structured. However, the proof sketch might be a bit long with too many notations and minor structural problems, so it might need some improvements such as the ones suggested in the "Weaknesses" section above.

-Quality and Reproducibility: The theoretical results in this paper appear to be correct, and the experimental details are provided in the appendix, making it possible to reproduce the experimental results.

-Novelty: To the best of my knowledge, the techniques introduced in this paper to extend the previous framework to convolutional layers are novel.

**Strength And Weaknesses:**

-Strengths:

1. This paper showed the existence of spurious local minima in deep convolutional neural networks, and this existence is mainly due to the use of convolutional layers. This indicates that convolutional layers can introduce spurious local minima themselves when equipped in a fully-connected neural network.

2. The network structure used for the theoretical proof is realistic and the ReLU activation pattern of the spurious local minima is non-trivial. The model structure can include fully-connected layers, convolutional layers, and max/average-pooling layers, which match the main ingredients of the convolutional neural networks used in practice.

3. This paper introduces new techniques to extend the framework in (He et al., 2020) to convolutional neural networks. Specifically, Lemma 2/Lemma 5 extends the output split to the convolutional case by solving a combinatorial problem, and Lemma 3/Lemma 4 uses a different way (3 consecutive convolutional layers) to perform local perturbation.

-Weaknesses:

1. The new proof techniques used in this paper seem to be specific about the setting in this paper and could not be easily generalized to solve other problems. The framework for constructing local minima mainly follows that of (He et al., 2020), and the additional technical lemmas seem to only work for the specific setting of convolutional layers. This might limit the technical significance of this paper.

2. It appears somewhat unclear what is the main message from the main theorems in this paper. Since the construction of spurious local minima of convolutional layers seems harder than fully-connected ones, does that mean convolutional layers are less likely to introduce spurious local minima? It would be better if the authors could include more implications and discussions of their theoretical results in this paper.

3. The notation in this paper is heavy and somewhat hard to follow, and it might be better if the authors could use more intuitive explanations/figures and fewer formulas to improve the understanding of the readers. Moreover, the organization of the formulas and lemmas can be further improved. For example, The margin around equations (9) and (10) seems too large. Section 3.2 could also be re-arranged to a place closer to Lemma 2 to help with the proof sketch.

**Summary Of The Paper:**

This paper theoretically proved the existence of spurious local minima in deep convolutional neural networks by construction. Specifically, for a convolutional neural network whose last three layers before the final fully-connected layer are all convolutional layers, the authors constructed a spurious local minimum for square loss or cross-entropy loss under some non-degeneracy assumptions for the input data and model weights. The construction and proof in this paper mainly follow the framework in (He et al., 2020) which deals with fully-connected layers, and the authors solved two main technical challenges (limited receptive field, and parameter sharing) to extend their proof to the convolutional case. Experiments of a 7-conv-layer CNN on CIFAR-10 are also provided to validate the theoretical results.

**Summary Of The Review:**

I tend to lean towards rejecting this paper. Although this paper introduces new techniques to construct spurious local minima for convolutional neural networks, these techniques might not be easily generalized to other settings, and the main message of the results might be somewhat unclear. Besides, the clarity of this paper also needs improvement.

---

> ### Author Response · Authors · 2022-11-18
> **Response to Reviewer wvat (Part 1 of 2)**
>
> Thank you very much for your insightful review and constructive suggestions. Below are our point-to-point responses to your comments.  We have also updated the final version accordingly. We sincerely hope that you can take into account the responses we have made and reevaluate the merit of this paper.
>
> Q1: The new proof techniques used in this paper seem to be specific about the setting in this paper and could not be easily generalized to solve other problems. The framework for constructing local minima mainly follows that of (He et al., 2020), and the additional technical lemmas seem to only work for the specific setting of convolutional layers. This might limit the technical significance of this paper.
>
> A1: Thank you very much for your comments. In our new version, we explicitly claim that our construction of spurious local minima is general and applies to CNNs with two consecutive convolutional layers (wherever there are in the CNN), which is easily satisfied by practical CNN models. After presenting our results for the case of having two consecutive convolutional layers on the top, we discuss the general case in Section 3.3. The idea is as follows.
>
> For the general case when the two consecutive convolutional layers we utilized are not located at the top and there are some convolutional, average pooling, or fully connected layers between them and the output layer, we can still construct spurious local minima using similar idea. The only difference is the setting of parameters for layers above the two consecutive convolutional layers, and we set them such that the output of the two consecutive convolutional layers are propagated unchanged (except pooling operations in possible subsequent pooling layers) to the first fully connected layer, which plays the role of layer $ L $ in Lemmas 1 and 4, and then the output of the first fully connected layer is forwarded unchanged to the output neurons. For such constructed $ \theta $ and $ \theta^{\prime} $, we can show that $ \theta $ is still a local minimum, there is still $ R(\theta^{\prime}) = R(\theta) $ and the empirical risk can be decreased by perturbing $ \theta^{\prime} $. The details are given in Appendix L.
>
>
> Q2: It appears somewhat unclear what is the main message from the main theorems in this paper. Since the construction of spurious local minima of convolutional layers seems harder than fully-connected ones, does that mean convolutional layers are less likely to introduce spurious local minima? It would be better if the authors could include more implications and discussions of their theoretical results in this paper.
>
> A2: Thank you very much for your insightful comments and suggestions. We revised our paper accordingly. We add a remark in Section 3.3 to discuss the implications of our results, given as follows.
>
> Our construction of spurious local minima shows that for practical datasets and CNNs each local minimum of the subnetwork is associated with a spurious local minimum. Since the output of a CNN with ReLU activations is a piece-wise linear function, and from the perspective of fitting data samples with piece-wise linear output [1], the local minima of subnetworks of a CNN (and consequently the spurious local minima of the CNN) may be common due to the abundance of different fitting patterns. Furthermore, as suggested by [2], CNNs have more expressivity than fully connected NNs per parameter in terms of the number of linear regions produced. The ability of producing more linear regions implies more fitting patterns, thus we conjecture that CNNs are more likely to produce spurious local minima than fully connected NNs of the same size. We leave the exploration of these ideas to our future work.
>
> We rewrite some parts of the introduction to highlight our contributions and ideas, shown as follows. We prove that infinite spurious local minima exist in the loss landscape of deep CNNs with squared loss or cross-entropy loss.  This is in contrast to the "no spurious local minima" property of deep linear networks.  The construction of spurious local minima in this paper is general and applies to practical dataset and CNNs containing two consecutive convolutional layers, which is satisfied by popular CNN architectures. The construction of spurious local minima for CNNs faces some technical challenges, and the construction for fully connected deep networks cannot be directly extended to CNNs. Our main contribution in this paper is to tackle these technical challenges.
>
> References:
>
> 1. Bo Liu. Spurious local minima are common for deep neural networks with piecewise linear activations. IEEE Transactions on Neural Networks and Learning Systems, pp. 1–13, 2022. doi: 10.1109/TNNLS.2022.3204319.
>
> 2. Huan Xiong, Lei Huang, Mengyang Yu, Li Liu, Fan Zhu, and Ling Shao. On the number of linear regions of convolutional neural networks. In International Conference on Machine Learning, 2020.

---

> ### Author Response · Authors · 2022-11-18
> **Response to Reviewer wvat (Part 2 of 2)**
>
> Q3: The notation in this paper is heavy and somewhat hard to follow, and it might be better if the authors could use more intuitive explanations/figures and fewer formulas to improve the understanding of the readers. Moreover, the organization of the formulas and lemmas can be further improved. For example, The margin around equations (9) and (10) seems too large. Section 3.2 could also be re-arranged to a place closer to Lemma 2 to help with the proof sketch.
>
> A3: Thank you very much for your comments and suggestions. We revised our paper accordingly.
>
> All technical details (regarding the construction of local minima $ \theta $, the construction of $ \theta^{\prime} $ with $ R(\theta^{\prime}) = R(\theta) $, the perturbation scheme, and result in Theorem 2) in the main text have been rewritten in informal and intuitive ways, and a new figure (Fig.1 in new version) is added to illustrate them. The formal versions of these technical details are moved to appendices.
>
> We simplify the notations, especially those appeared in the main text, and move some formulas and notations in the original main text to appendices.
>
> The auxiliary lemma (now Lemma 3) in original Section 3.2 has been placed to be closer to Lemma 2.
>
>
> Thank you again for your insightful comments and constructive suggestions.

---

### Official Review · Reviewer_JdM7 · 2022-10-25

**Confidence:** 4
**Clarity, Quality, Novelty And Reproducibility:** Please see above.
**Correctness:** 2
**Technical Novelty And Significance:** 3
**Empirical Novelty And Significance:** 3
**Recommendation:** 3

**Strength And Weaknesses:**

Studying the loss landscape, especially the existence of bad local minima is important for understanding neural network training. Most existing results are limited to fully-connected networks, hence an attempt to expand our understanding to CNN is valuable.

The proof of this paper consists of two major steps: 1) the construction of a local minimum, and 2) proof of the constructed local minimum to be spurious. However, in the first step (Lemma 1), the authors seem to intrinsically consider a fully-connected network. Below are detailed comments.

- The authors study the matrix-vector product formulation (2) for CNN. Due to weight sharing, some entries in matrix W and bias b represent the same network weights. That is, during the optimization, some entries in W and b should be kept identical.

- However, in the construction of Lemma 1, entries of $W$ and $b$ seem to be freely designed. If this is the case, it reduces to the fully-connected networks, bypassing the most challenging issue of CNN. Otherwise, the authors should justify how the shared weight constraints are satisfied. For example, the form of W in (2) does not allow assigning nonzero values to all diagonal entries. Then, how is the construction of $W^{L-1}$ in Lemma 1 possible?

Lemma 1 (construction of local minimum) serves as a foundation of this paper. If the problem of Lemma 1 can not be addressed properly, the paper has not really proven the existence of spurious local minima for CNN.

Other comments:

- Lemma 1 requires the matrix product $W^L I^{L-1,i} W^{L-1}\cdots W^1$ to be full-rank. This may not be possible since some of the rows in $W^{L-2}$ (and some of the columns in $W^{L}$) are set to 0. Are there any dimension constraints (on $n_{L-1}, n_{L-2}, n_{L-3}$) missing in Lemma 1?

- In the last sentence of page 3, $i^*_p$ may not be able to keep constant in its neighborhood if two or more features are equal.

- Assumption 1 needs further justification. If neural collapse happens, the inputs to the final layer may be identical for different samples in the same class.


**Summary Of The Paper:**

This paper constructs bad local minima for convolutional neural networks (CNN) with ReLU activation. Specifically, the paper establishes the following results:

1) If a neural network (termed "subnetwork") has a local minimum, then a network concatenating the subnetwork and 2 convolutional layers has a local minimum.
2) There exists another point with a smaller training loss than the above local minimum. Thus, the above local minimum is not a global minimum.
3) Construct a concrete example of the subnetwork, so as to validate the existence of a bad local minimum.

This is the first paper that addresses the existence of bad local minima for CNN with ReLU activation.

**Summary Of The Review:**

Extend an existing technique to construct a spurious local minimum for CNN, which is the first attempt in this field. However, the construction of the local minimum is not sufficiently justified to be a CNN point.

---

> ### Author Response · Authors · 2022-11-18
> **Response to Reviewer JdM7 (Part 1 of 2)**
>
> Thank you very much for your insightful review and constructive suggestions. Below are our point-to-point responses to your comments.  We have also updated the final version accordingly. We sincerely hope that you can take into account the responses we have made and reevaluate the merit of this paper.
>
> Q1: However, in the construction of Lemma 1, entries of $ W $ and $ b $ seem to be freely designed. If this is the case, it reduces to the fully-connected networks, bypassing the most challenging issue of CNN. Otherwise, the authors should justify how the shared weight constraints are satisfied. For example, the form of $ W $ in (2) does not allow assigning nonzero values to all diagonal entries. Then, how is the construction of $ W^{L-1} $ in Lemma 1 possible?
>
> A1: Thank you very much for your comments. To clarify how the shared weight constraints are satisfied and the "identity" matrices are implemented, we rewrite the example of matrix $ W^l $ for convolution operations in Section 2.2. Given a one-dimensional input $ \\mathbf{o}^{l-1}=(a,b,c,d,e)^T $ and a filter $ \\mathbf{w}_1=(w^1_1,w^2_1,w^3_1)^T$,
>
> the input becomes $ (0,a,b,c,d,e,0)^T $ after padding with zeros. Here, the input dimension is 5, the filter size is $ s_{l-1}=3 $ with $ T_l=1 $, and the patch size is also $ s_{l-1}=3 $. After padding a zero on both sides of original input, there are in total 5 overlapped patches, i.e., $ P_{l-1}=5 $. The output dimension is $ n_l= T_l P_{l-1}=5$. The convolution operation is  equivalent to
>
> \begin{equation*}
> \\mathbf{o}^l = \\sigma \left [  \begin{pmatrix}
> w^2_1 & w^3_1 & 0 & 0 & 0 \\\\
> w^1_1 &  w^2_1 & w^3_1 & 0 & 0  \\\\
> 0 & w^1_1 &  w^2_1 & w^3_1 & 0 \\\\
> 0 & 0 & w^1_1 &  w^2_1 & w^3_1 \\\\
> 0 & 0 & 0 & w^1_1 &  w^2_1
> \end{pmatrix} \begin{pmatrix}
> a\\\\
> b\\\\
> c\\\\
> d\\\\
> e\\\\
> \end{pmatrix}  +
> \begin{pmatrix}
> b^l_1\\\\
> b^l_1\\\\
> b^l_1\\\\
> b^l_1\\\\
> b^l_1
> \end{pmatrix} \right ] .
> \end{equation*}
>
> Thus, the input dimension and output dimension are equal. The shared weight constraint is reflected in the above equation by the fact that each row of the weight matrix uses the same filter $ (w^1_1,w^2_1,w^3_1) $.
> One can let $ (w^1_1=0,w^2_1=1,w^3_1=0)$ to set the weight matrix to a identity one.
>
> For the case of two-dimensional input with $ 3 \times 3 $ filters, each input feature map is padded by inserting a zero along its four sides. The identity forward propagation in layers $ L-1 $ and $ L-2 $ (see the new Fig.1(a)) can be implemented by setting corresponding submatrices in $  W^{L-1}$ and $ W^{L-2}$ to identity ones, which is equivalent to setting corresponding shared filters to the form of $ (0,\cdots,1,\cdots,0)^T$( with a single nonzero entry in the middle), and setting biases $ \mathbf{b}^{L-1},\mathbf{b}^{L-2} $ to zeros.
>
>
> Q2: Lemma 1 requires the matrix product $ W^{L} I^{L-1, i} W^{L-1}  \cdots W^{1} $ to be full-rank. This may not be possible since some of the rows in $  W^{L-2}$ (and some of the columns in $  W^{L}$) are set to 0. Are there any dimension constraints (on $ n_{L-1}, n_{L-2}, n_{L-3} $) missing in Lemma 1?
>
> A2: Thank you very much for your insightful comments. We have explained the structure of matrices $ W^l $ in A1. In our new version, we discard the full-rank requirement and directly enforce the nondegenerate condition: $ \hat{I}^{l, i} \ne 0 $ for all $ l \in [L-1] $ and $ i \in [N] $, which is much less restrictive and does not affect our proof in Appendix G.1.
>
>
> Q3: In the last sentence of page 3, $ i^*_p $ may not be able to keep constant in its neighborhood if two or more features are equal.
>
> A3: Thank you very much for your insightful comments. You are right. In our new version, we remove max-pooling operations and only consider average-pooling layers. The first reason for this is the constant $ i^*_p $ problem you pointed out (though can be fixed by adding constraint on local minimum $ \theta $). The second and more important reason is that we add proofs to show that there always exists local minimum $ \theta $ satisfying both Assumption 1 and $ R (\theta) > 0 $, which require linear average pooling operations.
>
>
> Q4: Assumption 1 needs further justification. If neural collapse happens, the inputs to the final layer may be identical for different samples in the same class.
>
> A4: Thank you very much for your insightful comments. Below we give the justification for Assumption 1 and we have revised our paper accordingly. For fully connected deep neural networks, [1,2] have assumed that all data points are distinct, i.e., $ \forall i,j \in [N] \ \text{and} \ i \ne j, \ \mathbf{x}_i \ne \mathbf{x}_j $. Under this condition, we show in Lemma 7 in Appendix C that for practical datasets and CNNs there always exists local minimum $ \theta $ for which Assumption 1 holds.

---

> ### Author Response · Authors · 2022-11-18
> **Response to Reviewer JdM7 (Part 2 of 2)**
>
> The idea is as follows and the detailed proof is given in Appendix C. We can set the parameters $ \left(W^{l},\\mathbf{b}^{l}\right)_{l=1}^{L}  $ such that each convolutional layer before layer $ L-2 $ forwards its input unchanged and the training loss of the CNN (more specifically, the subnetwork in Lemma 1) is then  expressed as
>
> $ R(\\theta) = \\frac{1}{N} \\sum_{i=1}^{N}  \left( W^{fc} W^{pool} \\mathbf{x}_i, \\mathbf{y}_i \right) $,
>
> where $ W^{pool} $ is a constant matrix that characterizes the average pooling operations from input to the fully connected layer, and $ W^{fc} $ characterizes the fully connection computation in layer $ L $.  The CNN has been reduced to a linear classifier and the corresponding point $ \theta $ in parameter space becomes a local minimum when minimizing
>
> $ \frac{1}{N} \sum_{i=1}^{N}  \left( W^{fc} W^{pool} \mathbf{x}_i, \mathbf{y}_i \right) $
>
> with respect to $  W^{fc} $.
> The only operation that can affect the  distinctness of $ \mathbf{o^{L-1,i}} \ (i \in [N]) $ at layer $ L-1 $ is then the pooling operation. If the distinctness of samples $ \mathbf{x}_i \ (i \in [N]) $ is preserved after all pooling operations of the CNN ( $ W^{pool}  $) when directly applied to $ \mathbf{x}_i $, which is usually the case, then $ \forall i,j \in [N] \ \text{and} \ i \ne j, \ \mathbf{o}^{L-1,i} \ne \mathbf{o}^{L-1,j} $, hence Assumption 1 holds for $ \theta $.
>
> Thank you very much for reminding me of the neural collapse phenomenon in which the features of the final hidden layer tend to collapse to class feature means. As a result, our construction of spurious local minima does not include those which may arise during the ending training phase when the loss is driving towards zero. We add this discussion in our new version.
>
> References:
>
> 1. Fengxiang He, Bohan Wang, and Dacheng Tao. Piecewise linear activations substantially shape the loss surfaces of neural networks. In International Conference on Learning Representations, 2020.
>
> 2. Bo Liu, Zhaoying Liu, Ting Zhang, and Tongtong Yuan. Non-differentiable saddle points and sub-optimal local minima exist for deep relu networks. Neural Networks, 144:75–89, 2021.
>
> Thank you again for your insightful comments and helpful suggestions.

---

> ### Comment · Reviewer_JdM7 · 2022-12-14
> **Problem of Lemma 1**
>
> Thanks for the authors' reply. I understand how to set $W^L$ as an identity matrix now.
>
> I have a question regarding the proof of Lemma 6. What does "the space of $o(x_i)$ has been fully explored by..." mean？ $o(x_i)$ should be a vector, it spans a one-dimensional subspace, but what does it mean by "fully explored"?
>
> I guess the authors mean that if we define an input matrix $\mathbf{O} = [o(x_1),\cdots, o(x_N)]$, the output can be seen as a row combination of $\mathbf{O}$. However, as we perturb the last two convolutional layers, the output may no longer be a row combination of $\mathbf{O}$, because the activation patterns could become different for data points.
>
> Could the authors resolve this?

---

### Official Review · Reviewer_h5wB · 2022-10-25

**Confidence:** 3
**Correctness:** 3
**Technical Novelty And Significance:** 3
**Empirical Novelty And Significance:** Not applicable
**Recommendation:** 5

**Clarity, Quality, Novelty And Reproducibility:**

- The paper is heavy in notation and in many cases it uses multiple equivalent symbols to denote the same thing. For example, the output of the network for input $x_i$ is denoted by four symbols: $o_i, o(x_i), o^L(x_i), o^{L,i}$. I'm not sure if this improves readability; rather, it is likely to hurt readability.

- It is good to be explicit about the construction, but sometimes too many details can hurt. At first glance, Lemma 3 looked just overwhelming to me. Perhaps it would be better to avoid spelling out the explicit details in the statement?

- Other than the aforementioned issues, the paper is not too difficult to follow.

- As mentioned above, the paper seems to have developed novel techniques to overcome the difficulties that arise in the analysis of CNNs. Unfortunately, I didn't have a chance to check the details of the proofs though.

**Strength And Weaknesses:**

- Existence/absence of bad local minima is an important topic in the literature, and it is true that the investigation has mostly focused on fully-connected networks. This paper is one of the first attempts to characterize spurious local minima that arise from convolutional networks, and the authors seem to have overcome several technical barriers that arise in CNNs.

- However, while I got the main idea, reading the paper left me with some questions, mostly related to Theorem 1 and its proof. If all the issues are clarified I would be happy to raise my score.

- I'm confused about the relationship between the number of neurons $n_l = T_l P_{l-1}$ and the number of patches $P_l$. As one can see in Eq (1) and (2), the output $o^l \in \mathbb R^{n_l}$ is plugged in as the input for the next layer. Considering that we apply zero-padding and stride is equal to 1, it seems to me that the number of patches at layer $l$ would be the same as $P_l = n_l$. However, it seems that the paper is assuming $P_l = P_{l-1}$ by zero-padding (e.g., see $P_{L-2} = P_{L-3} = P_{L-4}$ right above Lemma 1). To me this looks like a contradiction unless we have $T_l = 1$. Can you clarify?

- In Lemma 1, we make portions of $W^{L-1}$ and $W^{L-2}$ equal to "identity" matrices by filling in the diagonal entries with 1. However, I wonder whether this is always possible? For example, if we look at Eq (2), there are certain entries that are *forced to be zero* by the structure of the weight matrix that arises from convolution. Also, it is mentioned that the conclusion of Lemma 1 still holds if the top layers are pooling layers. How? Take average pooling for example. One does not have the freedom to choose $W^{L-1}$ and $W^{L-2}$ arbitrarily; the matrices are just fixed, based on the number of patches and filter length.

- Lemma 1 requires that $\hat W^L \hat I^{L-1,i} \cdots \hat W^1$ is full-rank for all $i$. While this may look reasonable for networks of general size, for certain cases I think this condition can become a big restriction. For example, consider the case where all $\hat W^l$ are square matrices. Then necessarily all the diagonal matrices $\hat I^{l,i}$ must set their diagonal entry to one in order to satisfy the full-rank condition. This means that all ReLUs must be turned on for all datasets, which boils down to a linear model.

- For the local minimum $\theta$ constructed by Lemma 1, the rest of the proof assumes that the local minimum has $R (\theta) > 0$. How can you safely assume that? The conclusion that the local minimum $\theta$ is spurious becomes vacuous if this assumption $R (\theta) > 0$ actually does not hold for any local minimum. In other words, the theorem does not *disprove* the possibility that all the local minima are globally optimal because it relies on this assumption. Would there be a way to show the existence of $\theta$ that always satisfies $R (\theta) > 0$? Maybe by using the construction idea from Theorem 2?

- Why should Eq (10) necessarily hold? If $\eta$ is chosen based on $h$ with the biggest split threshold, why should the sum in Eq (10) suddenly become zero for all $j \neq h$?

- In Theorem 2, it is assumed that for any target $y_{i^*}$ there exists $\tilde y_{i^*}$ such that the loss $l(\tilde y_{i^*}, y_{i^*}) = 0$. What happens for losses for which such $\tilde y_{i^*}$ does not exist, e.g., cross-entropy loss? Also, in Theorem 2, is $R (\theta) > 0$ always satisfied?


**Summary Of The Paper:**

This paper theoretically investigates spurious local minima (i.e., local minima that are not globally optimal) that arise in the empirical risk of deep convolutional neural networks.

- By embedding a local minimum of a subnetwork to a larger network, the paper constructs a local minimum $\theta$ of the larger network (Lemma 1). Then, it is shown that a different point $\theta'$ can be perturbed in a way that the empirical risk at $\theta'$ is strictly smaller than $\theta$ (Lemmas 2, 3, and 4). This proves that the local minimum $\theta$ is spurious (Theorem 1).

- Theorem 1 relies on the existence of a local minimum of a subnetwork. Theorem 2 provides a construction of a local minimum that is "nontrivial"---in the sense that different ReLU activation patterns exist for different data points, unlike existing results.


**Summary Of The Review:**

This paper develops some novel techniques to show the existence of bad local minima in the empirical risk of CNNs. If fully correct, this looks like a good contribution to the literature. However, some issues (mostly with proof and notation) are raised and I set the recommendation to weak reject for now. If the authors clarify these issues through the rebuttal period, I would be happy to raise my score.

---

> ### Author Response · Authors · 2022-11-18
> **Response to Reviewer h5wB (Part 1 of 3)**
>
> Thank you very much for your thorough review and constructive comments. Below are our point-to-point responses to your comments.  We have also updated the final version accordingly.
>
> Q1: I'm confused about the relationship between the number of neurons $ n_l = T_l P_{l-1} $ and the number of patches $ P_l $. As one can see in Eq (1) and (2), the output $ \mathbf{o}^l \in \mathbb{R}^{n_l} $ is plugged in as the input for the next layer. Considering that we apply zero-padding and stride is equal to 1, it seems to me that the number of patches at layer $ l $  would be the same as $  P_l=n_l $. However, it seems that the paper is assuming $ P_l=P_{l-1} $ by zero-padding (e.g., see $ P_{L-2}=P_{L-3}=P_{L-4} $ right above Lemma 1). To me this looks like a contradiction unless we have $ T_l=1 $. Can you clarify?
>
> A1: Thank you very much for your insightful comments. We add some sentences in our new version to clarify the concepts. In a convolutional layer $ l $, if there are multiple feature maps in input layer $ (l-1) $, a patch will include corresponding neighboring neurons from every feature map. Therefore, $ P_{l-1} \ne n_{l-1} $ if the number of feature maps in layer $ (l-1) $ is greater than 1. The number of neurons in a single feature map in the $ l $th layer is $ P_{l-1} $, thus $ n_l = T_l P_{l-1} $. The size of single feature map equals for input and output layers after zero-padding the input. Further clarification can be found in the following example given in A2.
>
>
> Q2: In Lemma 1, we make portions of $ W^{L-1} $ and $ W^{L-2} $ equal to "identity" matrices by filling in the diagonal entries with 1. However, I wonder whether this is always possible? For example, if we look at Eq (2), there are certain entries that are forced to be zero by the structure of the weight matrix that arises from convolution.
>
> A2: Thank you very much for your insightful comments. To illustrate how the "identity" matrices are implemented, we rewrite the example of matrix $ W^l $ for convolution operations in Section 2.2. Given a one-dimensional input $ \\mathbf{o}^{l-1}=(a,b,c,d,e)^T $ and a filter $ \\mathbf{w}_1=(w^1_1,w^2_1,w^3_1)^T$,
>
> the input becomes $ (0,a,b,c,d,e,0)^T $ after padding with zeros. Here, the input dimension is 5, the filter size is $ s_{l-1}=3 $ with $ T_l=1 $, and the patch size is also $ s_{l-1}=3 $. After padding a zero on both sides of original input, there are in total 5 overlapped patches, i.e., $ P_{l-1}=5 $. The output dimension is $ n_l= T_l P_{l-1}=5$. The convolution operation is  equivalent to
>
> \begin{equation*}
> \\mathbf{o}^l = \\sigma \left [  \begin{pmatrix}
> w^2_1 & w^3_1 & 0 & 0 & 0 \\\\
> w^1_1 &  w^2_1 & w^3_1 & 0 & 0  \\\\
> 0 & w^1_1 &  w^2_1 & w^3_1 & 0 \\\\
> 0 & 0 & w^1_1 &  w^2_1 & w^3_1 \\\\
> 0 & 0 & 0 & w^1_1 &  w^2_1
> \end{pmatrix} \begin{pmatrix}
> a\\\\
> b\\\\
> c\\\\
> d\\\\
> e\\\\
> \end{pmatrix}  +
> \begin{pmatrix}
> b^l_1\\\\
> b^l_1\\\\
> b^l_1\\\\
> b^l_1\\\\
> b^l_1
> \end{pmatrix} \right ] .
> \end{equation*}
> Thus, the input dimension and output dimension are equal.  One can let $ (w^1_1=0,w^2_1=1,w^3_1=0)$ to set the weight matrix to a identity one.
>
> For the case of two-dimensional input with $ 3 \times 3 $ filters, each input feature map is padded by inserting a zero along its four sides. The identity forward propagations in layers $ L-1 $ and $ L-2 $ (see the new Fig.1(a)) can be implemented by setting corresponding submatrices in $  W^{L-1}$ and $ W^{L-2}$ to identity ones, which is equivalent to setting corresponding filters to the form of $ (0,\cdots,1,\cdots,0)^T$( with a single nonzero entry in the middle), and setting biases $ \mathbf{b}^{L-1},\mathbf{b}^{L-2} $ to zeros.
>
>
> Q3: Also, it is mentioned that the conclusion of Lemma 1 still holds if the top layers are pooling layers. How? Take average pooling for example. One does not have the freedom to choose $ W^{L-1} $ and $ W^{L-2} $ arbitrarily; the matrices are just fixed, based on the number of patches and filter length.
>
> A3: Thank you very much for your insightful comments. In our new version, we explicitly claim that our construction of spurious local minima is general and applies to CNNs with two consecutive convolutional layers (wherever there are in the CNN), which is easily satisfied by practical CNN models. After presenting our results for the case of having two consecutive convolutional layers ($ (L-1) $ and $ (L-2) $) on the top, we discuss the general case in Section 3.3.

---

> ### Author Response · Authors · 2022-11-18
> **Response to Reviewer h5wB (Part 2 of 3)**
>
> For the general case when the two consecutive convolutional layers we utilized are not located at the top and there are some convolutional, average pooling, or fully connected layers between them and the output layer, we can still construct spurious local minima using similar idea. The only difference is the setting of parameters for layers above the two consecutive convolutional layers, and we set them such that the output of the two consecutive convolutional layers are propagated unchanged (except pooling operations in possible subsequent pooling layers) to the first fully connected layer, which plays the role of layer $ L $ in Lemmas 1 and 4, and then the output of the first fully connected layer is forwarded unchanged to the output neurons. For such constructed $ \theta $ and $ \theta^{\prime} $, we can show that $ \theta $ is still a local minimum, there is still $ R(\theta^{\prime}) = R(\theta) $ and the empirical risk can be decreased by perturbing $ \theta^{\prime} $. The details are given in Appendix L.
>
>
> Q4: Lemma 1 requires that $ \hat{W}^{L} \hat{I}^{L-1, i} \hat{W}^{L-1}  \cdots \hat{W}^{1} $ is full-rank for all $ i $. While this may look reasonable for networks of general size, for certain cases I think this condition can become a big restriction. For example, consider the case where all are square matrices. Then necessarily all the diagonal matrices must set their diagonal entry to one in order to satisfy the full-rank condition. This means that all ReLUs must be turned on for all datasets, which boils down to a linear model.
>
> A4: Thank you very much for your insightful comments. You are right. We realized that the full-rank requirement is sometimes a big restriction. In our new version, we discard the full-rank requirement and directly enforce the nondegenerate condition: $ \hat{I}^{l, i} \ne 0 $ for all $ l \in [L-1] $ and $ i \in [N] $, which does not affect our proof. This is much less restrictive than the full-rank requirement.
>
>
> Q5: For the local minimum $ \theta $ constructed by Lemma 1, the rest of the proof assumes that the local minimum has $ R (\theta) > 0 $. How can you safely assume that? The conclusion that the local minimum $ \theta $ is spurious becomes vacuous if this assumption $ R (\theta) > 0 $ actually does not hold for any local minimum. In other words, the theorem does not disprove the possibility that all the local minima are globally optimal because it relies on this assumption. Would there be a way to show the existence of $ \theta $ that always satisfies $ R (\theta) > 0 $? Maybe by using the construction idea from Theorem 2?
>
> A5: Thank you very much for your insightful comments. In our new version, we point out that for training data that cannot be fit by linear models (also assumed by He et al. (2020); Liu et al. (2021)) and popular CNN architectures, there always exists point $ \theta $ such that both Assumption 1 and $ R(\theta) > 0 $ hold.
>
> The idea is as follows and the details are given in Lemma 8 in Appendix C.
> We can set the parameters $ \left(W^{l},\\mathbf{b}^{l}\right)_{l=1}^{L}  $ such that each convolutional layer before layer $ L-2 $ forwards its input unchanged and the training loss of the CNN (more specifically, the subnetwork in Lemma 1) is then  expressed as
>
> $ R(\\theta) = \\frac{1}{N} \\sum_{i=1}^{N}  \left( W^{fc} W^{pool} \\mathbf{x}_i, \\mathbf{y}_i \right) $,
>
> where $ W^{pool} $ is a constant matrix that characterizes the average pooling operations from input to the fully connected layer, and $ W^{fc} $ characterizes the fully connection computation in layer $ L $.  The CNN has been reduced to a linear classifier and the corresponding point $ \theta $ in parameter space becomes a local minimum when minimizing
>
> $ \frac{1}{N} \sum_{i=1}^{N}  \left( W^{fc} W^{pool} \mathbf{x}_i, \mathbf{y}_i \right) $
>
> with respect to $  W^{fc} $.
>
> Let	$ W^\star \in \mathbb{R}^{d_y \times d_x}$ is a local minimizer of
>
> $ \frac{1}{N} \sum_{i=1}^{N}  \left( W \mathbf{x}_i, \mathbf{y}_i \right) $,
>
> where $ l $ is the loss function. For training data that cannot be fit by linear models, we have
> $ \frac{1}{N} \sum_{i=1}^{N}  \left( W^\star \mathbf{x}_i, \mathbf{y}_i \right) > 0$.
>
> Since $\frac{1}{N} \sum_{i=1}^{N}  \left( W^\star \mathbf{x}_i, \mathbf{y}_i \right)=$
>
> $\min_{W \in \mathbb{R}^{d_y \times d_x}} \frac{1}{N} \sum_{i=1}^{N}  \left( W \mathbf{x}_i, \mathbf{y}_i \right)$, we have $ R(\theta) >0 $.

---

> ### Author Response · Authors · 2022-11-18
> **Response to Reviewer h5wB (Part 3 of 3)**
>
> Q6: Why should Eq (10) necessarily hold? If  is chosen based on  with the biggest split threshold, why should the sum in Eq (10) suddenly become zero for all $ j \ne h $?
>
> A6: Thank you very much for your comments. We explain this in our revised  paper. Since $\eta:=\eta^h$ for the $ h $th location is the biggest threshold among all locations $ j \in \left[M\right] $,  by Eq (7) (the Eq (9) in original version) shown below,
> \begin{equation*}
> \begin{aligned}
> v_{1}^{j}, v_{2}^{j}, \cdots, v_{n}^{j} > \eta^j,   \ \ v_{n+1}^{j}, v_{n+2}^{j}, \cdots, v_{N}^{j} < \eta^j , \ \ \sum_{i=1}^{n} u_{i} \ne 0 ,
> \end{aligned}
> \end{equation*}
> then for each location $ j \ne h $, $ \sum_{i=1}^{n} u_{i} \ne 0 $ (the sum of $ u_{i} $s for samples whose $ v_{i}^{j}$s are greater than $\eta:=\eta^h$) is not satisfied since $ \eta^j < \eta^h$, thus we have $\forall j \in [M], \ j \ne h$,
>
> $\\forall j \\in [M], \\ j \\ne h, \\ \\ \\ \\sum_{i \\in \\left\\{  1,2,\\cdots,N   |  v^j_i > \eta  \\right\\}    } u_i = 0. $
>
>
> Q7: In Theorem 2, it is assumed that for any target $\\textbf{y}_{i^{*}} $,
>
> there exists $ y^\\prime_{i^\\star}$  such that the loss
>
> $l ( y^\prime_{i^\star}, y_{i^\star} )=0$.
>
> What happens for losses for which such $\textbf{y}_{i^{*}} $ does not exist, e.g., cross-entropy loss? Also, in Theorem 2, is $ R(\theta) >0 $ always satisfied?
>
> A7: Thank you very much for your insightful comments. We revised our paper accordingly. If loss function is the squared loss for which loss can be equal to zero,
> we use our original analytical expression to solve $ \lambda_j $. If loss function is the cross-entropy loss, $ \lambda_j $s are determined by minimizing the loss shown in Eq (16) (originally Eq (21)).
>
> Yes, in Theorem 2 we also need to make sure $ R (\theta) > 0 $. With the same ideas as dicussed in A5, when using the local minimum $ \theta_1 $ constructed in Theorem 2 as the subnetwork of $ \theta $ in Lemma 1, we can construct the second subnetwork in Theorem 2 such that there always exists point $ \theta_1 $ (and consequently $ \theta $) for which $ R (\theta) > 0 $ holds for popular datasets and CNN architectures. We revised Section 4 accordingly.
>
> Q8: The paper is heavy in notation and in many cases it uses multiple equivalent symbols to denote the same thing. For example, the output of the network for input $ x_i $ is denoted by four symbols: $ \mathbf{o}_{i} $,
>
> $ \mathbf{o}(\mathbf{x}_{i}) $,
>
> $ \\mathbf{o}^{L}(\\mathbf{x}_{i}) $,
> $ \\mathbf{o}^{L, i} $. I'm not sure if this improves readability; rather, it is likely to hurt readability.
>
> A8: Thank you very much for your suggestions. We simplify the notations, and move some formulas and notations in the original main text to appendices.
>
>
> Q9: It is good to be explicit about the construction, but sometimes too many details can hurt. At first glance, Lemma 3 looked just overwhelming to me. Perhaps it would be better to avoid spelling out the explicit details in the statement?
>
> A9: Thank you very much for your suggestions. We rewrite some parts of our paper to highlight the ideas. All lemmas (regarding the construction of local minima $ \theta $, the construction of $ \theta^{\prime} $ with $ R(\theta^{\prime}) = R(\theta) $, and the perturbation scheme) in the main text have been rewritten in informal ways to describe their ideas intuitively, with a new figure (Fig.1 in new version) to illustrate them. The explicit details are now given in appendices.
>
>
> Thank you again for your insightful comments and helpful suggestions.

---

> > ### Comment · Reviewer_h5wB · 2022-12-02
> > **Response acknowledged**
> >
> > I appreciate the authors for their response! I can also tell that they put a lot of effort into updating the paper. Most of your replies are clarifying and the updates seem to have made the paper clearer (although I haven't gone through the entire updated paper). Please allow me leave one comment on my remaining doubt:
> >
> > A5: I get your point that you can construct a provably spurious local minimum provided that the dataset cannot be perfectly fitted with linear models, but to the best of my understanding, this specific construction does not guarantee that the local minima considered by Lemma 1 all provably satisfy $R > 0$. Rather, the construction by Lemmas 7 and 8 in fact make the network "linear" by turning on all the ReLUs, which is in contrast to the paper's initial claim in page 2 that "In comparison, we deal with spurious local minima for CNNs in this work, and the constructed spurious local minima can be nontrivial in which nonlinear predictors are generated and some ReLU neurons are inactive"?

---

> > > ### Author Response · Authors · 2022-12-03
> > > **Re: Response acknowledged**
> > >
> > > Thank you very much for your insightful comment.
> > > Yes, it is not guaranteed that all local minima considered by Lemma 1 satisfy $ R(\theta) >0 $, and we only consider those local minima with $ R(\theta) >0 $ later in Lemma 2 and Theorem 1 to demonstrate the existence of spurious local minima.
> > > We already constructed a specific local minima in Lemma 8 with $ R(\theta) >0 $, in which all ReLU units in the subnetwork of Lemma 1 are turned on. Actually, the local minima in Lemma 1 can be general, depending on data, in the sense that some ReLU neurons in the subnetwork are inactive for some samples (consequently the resulted models are nonlinear). Such nontrivial subnetworks of $ \theta $ with $ R(\theta) >0 $ can be constructed similar to $ \theta_1 $ in Theorem 2.
> > > By taking the "linear" CNN constructed in Lemma 8 as the second subnetwork of $ \theta_1 $,  we will have $ R(\theta) >0 $ when samples $\\left \\{ \mathbf{x}_i | i \notin  I  \\right \\}  $ cannot be perfectly fit by linear models, and at the same time some  ReLU neurons in the first subnetwork of $ \theta_1 $ are inactive for $\mathbf{x}_i \ (i \notin  I ) $  and the activation patterns for samples $\mathbf{x}_i \ (i \notin  I ) $ and $\mathbf{x}_j \ (j \in  I ) $ are different.
> > >
> > > We will revise the next version of our paper accordingly to clarify this. Thank you again for your comment.

---

### Official Review · Reviewer_L56V · 2022-10-31

**Confidence:** 3
**Correctness:** 3
**Technical Novelty And Significance:** 3
**Empirical Novelty And Significance:** Not applicable
**Recommendation:** 5

**Clarity, Quality, Novelty And Reproducibility:**

The paper is very technical and written in a nonintuitive way -- for example,
when the authors describe their results and techniques in the introduction of
the paper, it reads like a direct translation of some of the proof techniques
into natural language rather than as an elucidation of the ideas behind the
proof (why such a result is true, how one might arrive at it, etc.). This makes
the effort required to penetrate the paper for a nonspecialist unduly high.

In general (following comments in the previous section), the preliminaries
section 2 is written in an unclear and disorganized style that is overly
general and unnecessarily confusing.  For example, the definition of the max
pooling operation (3) is not clearly made with respect to its interaction with
distinct filters $T_{\ell}$ in a layer; the use of a matrix $W^{\ell}$ for all
operations does not accurately show the distinct structures of each operation
(average pool; convolution; fully connected) as functions of the actual weight
parameters of the network and architecture parameters, and hence makes it seem
unnecessary to define such intricate notation here (just put it in an appendix
instead, if it is not essential to understanding the main body of the paper?);
the discussion of feature map padding is done vaguely in an unhelpful way (one
wants to know here **what restrictions does this place on the various
parameters of the network**? In (2), the input feature map dimension is 5 and
the output dimension is 6. If I pad the input by 1, now the input dimension is
6 and the output dimension is 8. It does not seem to be possible to pad the
input feature map to make the input and output sizes equal!). I would recommend
the authors polish the writing and only introduce notation that will be
necessary to discuss the main results that are presented in the subsequent
sections of the main paper.


The technical presentation of the argument reads as though it is poorly
organized, as well. Lemma 1 is a technical result that seems rather abstruse,
and it also applies to a very limited setting (only Conv + ReLU layers); the
lemma has unnatural technical assumptions on nondegeneracy, and after
presenting it the authors then state that "the conclusion in Lemma 1 still
holds" for more general settings without any proof or connection to a result in
the appendices. Why not present this result informally, with the necessary
links to the fully general versions in the appendices made? The second
paragraph of section 3.1 says "for ease of presentation", but this seems to be
off to me -- the result presented is still very technical, and the
simplifications seem to rather have the effect of making it unclear what
networks the authors' result actually applies to, rather than the stated
intention. The same could be suggested for the following material that leads to
Theorem 1, although here the technicality may be essential.



**Strength And Weaknesses:**

## Strengths

The authors argue that their approach to constructing local minima is modular,
in that the task is reduced to studying a local minimum of a subnetwork (which
can then be embedded into larger networks). This may be useful for obtaining
general conclusions. They mention on page 5 that this is new in the context of
CNNs (standard for feedforward nets).

The analysis seems highly technical and intricate, involving detailed
index-level manipulations of the filters in the network. (This also makes it
challenging to verify). The result considers fairly general CNN architectures
and various loss functions, including regression and classification losses.

## Weaknesses

The claims of novelty in the introduction (end of first paragraph, end of third
paragraph) are not correct: the related work section even discusses a work of
Du et al. on one-layer CNNs that shows spurious minima (and see similar works
not cited, such as https://arxiv.org/abs/1909.03172). This is not hard to fix:
you can claim the first result on spurious minimizers in deep CNNs / CNNs with
several channels, etc. I would also recommend an additional literature search
for possible related references in this context, since the one I mention here
was missed.

Claim about argmax at bottom of page 3 and top of page 4: I do not think this
is correct in general -- consider the case where there are two distinct indices
$i$ which achieve the argmax corresponding to different filters, and suppose
the coordinates $o_p^{l-1}(i)$ are open mappings (is there any reason for them
not to be?). Then every neighborhood of the parameters $W, b$ contains points
at which the argmax is taken uniquely on a distinct coordinate involved in the
tie; it implies that there is no locally-valid way to define a *constant* matrix
that selects the right element of the max-pooling operation. The assumption that the
coordinate maps are open is only sufficient; it should be possible to construct
counterexamples under more general conditions specific to the definition of the
feature maps.

The result relies on a technical assumption (Assumption 1) that seems to
significantly limit the scope of the theory (or even make it vacuous).
Assumption 1 is a nondegeneracy assumption on the network features at a local
minimizer (this is used to prove that the minimizer is spurious). As a result
of the dependence on this assumption, it does not seem that the theory makes
any predictions about the existence or non-existence of spurious minimizers in
any actual network architectures. Based on the way Section 4 and Theorem 2-3
are written, I understand that even in the authors' construction of a concrete
local minimum, it is still necessary to make assumption 1 in order to conclude
that the local minimum is spurious. In the rebuttal, I would appreciate if the
authors could enlighten me on whether assumptions like this are standard in
other works that study spurious minimizers in neural networks (say, in the
well-studied feedforward setting). From my perspective, I would find it hard to
understand why the natural extension from the results mentioned in the related
work section by Du et al. and the one I mentioned above for concrete spurious
minimizers non-overlapping-receptive-field and one-layer Convnets would not be
something for single-layer and overlapping receptive field networks, or
multi-layer and non-overlapping receptive field networks, rather than the
extremely general setting the authors consider here (given that there are no
concrete conclusions in this setting in the presented work).

## Minor / Questions

Bottom of page 1 and elsewhere: "perturbated" -> "perturbed" ("perturbate" does
not sound correct)


**Summary Of The Paper:**


The authors study the loss landscape of multilayer convolutional neural
networks, and in particular consider the construction of spurious minimizers of
networks trained on regression/classification tasks. They consider networks
with fully connected, convolutional, max-pooling, and average pooling layers,
as well as ReLU nonlinearities. Their main results (1) construct spurious
minimizers in a network assuming that a subnetwork construction is satisfied
and a certain nondegeneracy assumption holds at the local minimizer obtained
from this subnetwork construction; (2) show in a specific setting (slightly
nonstandard architecture) that a subnetwork can explicitly be constructed to
give a local minimizer.


**Summary Of The Review:**


The strength of the assumptions made in the theory seems to significantly limit
its scope. As a result, although the work is technically impressive, it does
not seem to shed much novel light on when and why spurious minimizers exist in
convolutional neural networks trained on various practical tasks. The notation
and writing could do with additional polishing.

---

> ### Author Response · Authors · 2022-11-18
> **Response to Reviewer L56V (Part 1 of 4)**
>
> Thank you very much for your thorough review and constructive comments. Below are our point-to-point responses to your comments.  We have also updated the final version accordingly.
>
> Q1: The claims of novelty in the introduction (end of first paragraph, end of third paragraph) are not correct: the related work section even discusses a work of Du et al. on one-layer CNNs that shows spurious minima (and see similar works not cited, such as https://arxiv.org/abs/1909.03172). This is not hard to fix: you can claim the first result on spurious minimizers in deep CNNs / CNNs with several channels, etc. I would also recommend an additional literature search for possible related references in this context, since the one I mention here was missed.
>
>
> A1: Thank you very much for your comments and suggestions. We revised our paper accordingly.
>
> In our new version, we claim that our result is the first one on spurious minimizers in deep CNNs.
>
> We also added some new references, including the paper you recommended and another paper on one-hidden-layer CNNs, and some papers on the local convergence of gradient descent for sufficiently over-parameterized deep networks including CNNs.
>
>
> Q2: Claim about argmax at bottom of page 3 and top of page 4: I do not think this is correct in general -- consider the case where there are two distinct indices  which achieve the argmax corresponding to different filters, and suppose the coordinates $ o^{l-1}_p(i) $ are open mappings (is there any reason for them not to be?). Then every neighborhood of the parameters $ W,b $ contains points at which the argmax is taken uniquely on a distinct coordinate involved in the tie; it implies that there is no locally-valid way to define a constant matrix that selects the right element of the max-pooling operation. The assumption that the coordinate maps are open is only sufficient; it should be possible to construct counterexamples under more general conditions specific to the definition of the feature maps.
>
> A2: Thank you very much for your insightful comments. You are right. In our new version, we remove max-pooling operations and only consider average-pooling layers. The first reason for this is the constant matrix problem you pointed out (though can be fixed by adding constraint on local minimum $ \theta $). The second and more important reason is that we add proofs to show that there always exists local minimum $ \theta $ satisfying both Assumption 1 and $ R (\theta) > 0 $, which require linear average pooling operations.
>
> Q3: The result relies on a technical assumption (Assumption 1) that seems to significantly limit the scope of the theory (or even make it vacuous). Assumption 1 is a nondegeneracy assumption on the network features at a local minimizer (this is used to prove that the minimizer is spurious). As a result of the dependence on this assumption, it does not seem that the theory makes any predictions about the existence or non-existence of spurious minimizers in any actual network architectures. Based on the way Section 4 and Theorem 2-3 are written, I understand that even in the authors' construction of a concrete local minimum, it is still necessary to make assumption 1 in order to conclude that the local minimum is spurious. In the rebuttal, I would appreciate if the authors could enlighten me on whether assumptions like this are standard in other works that study spurious minimizers in neural networks (say, in the well-studied feedforward setting).
>
> A3: Thank you very much for your insightful comments. For fully connected deep neural networks, [1,2] have assumed that all data points are distinct, i.e., $ \forall i,j \in [N] \ \text{and} \ i \ne j, \ \mathbf{x}_i \ne \mathbf{x}_j $. Under this condition, we further show in Lemma 7 in Appendix C that for practical datasets and CNNs there always exists local minimum $ \theta $ for which Assumption 1 holds.
>
> The idea is as follows and the detailed proof is given in Appendix C. We can set the parameters $ \left(W^{l},\\mathbf{b}^{l}\right)_{l=1}^{L}  $ such that each convolutional layer before layer $ L-2 $ forwards its input unchanged and the training loss of the CNN (more specifically, the subnetwork in Lemma 1) is then  expressed as
>
> $ R(\\theta) = \\frac{1}{N} \\sum_{i=1}^{N}  \left( W^{fc} W^{pool} \\mathbf{x}_i, \\mathbf{y}_i \right) $,
>
> where $ W^{pool} $ is a constant matrix that characterizes the average pooling operations from input to the fully connected layer, and $ W^{fc} $ characterizes the fully connection computation in layer $ L $.  The CNN has been reduced to a linear classifier and the corresponding point $ \theta $ in parameter space becomes a local minimum when minimizing
>
> $ \frac{1}{N} \sum_{i=1}^{N}  \left( W^{fc} W^{pool} \mathbf{x}_i, \mathbf{y}_i \right) $
>
> with respect to $  W^{fc} $.

---

> ### Author Response · Authors · 2022-11-18
> **Response to Reviewer L56V (Part 2 of 4)**
>
> The only operation that can affect the  distinctness of $ \mathbf{o^{L-1,i}} \ (i \in [N]) $ at layer $ L-1 $ is then the pooling operation. If the distinctness of samples $ \mathbf{x}_i \ (i \in [N]) $ is preserved after all pooling operations of the CNN ( $ W^{pool}  $) when directly applied to $ \mathbf{x}_i $, which is usually the case, then $ \forall i,j \in [N] \ \text{and} \ i \ne j, \ \mathbf{o}^{L-1,i} \ne \mathbf{o}^{L-1,j} $, hence Assumption 1 holds for $ \theta $.
>
> Yes, when using the local minimum constructed in Theorem 2-3 as the subnetwork of $ \theta $ in Lemma 1, we still need to make sure that Assumption 1 holds. Using the same idea as discussed above, we can construct the second subnetwork in Theorem 2-3 such that there always exists point $ \theta $ for which  Assumption 1 holds for popular datasets and CNN architectures. We revised Section 4 accordingly.
>
> References:
>
> 1. Fengxiang He, Bohan Wang, and Dacheng Tao. Piecewise linear activations substantially shape the loss surfaces of neural networks. In International Conference on Learning Representations, 2020.
>
> 2. Bo Liu, Zhaoying Liu, Ting Zhang, and Tongtong Yuan. Non-differentiable saddle points and sub-optimal local minima exist for deep relu networks. Neural Networks, 144:75–89, 2021.
>
> Q4:  From my perspective, I would find it hard to understand why the natural extension from the results mentioned in the related work section by Du et al. and the one I mentioned above for concrete spurious minimizers non-overlapping-receptive-field and one-layer Convnets would not be something for single-layer and overlapping receptive field networks, or multi-layer and non-overlapping receptive field networks, rather than the extremely general setting the authors consider here (given that there are no concrete conclusions in this setting in the presented work).
>
> A4: [3,4,5] showed the existence of spurious local minima for one-hidden-layer CNNs with a single non-overlapping filter, Gaussian input and squared loss. Given non-overlapping filter and Gaussian input, the population risk with squared loss function can be formulated as an analytical expression with respect to the single filter $ \mathbf{w} $ by integrating out the Gaussian input, which facilitates the analysis of loss landscape. Thus, the techniques used in [3,4,5] cannot be extended to the general case of empirical risk with arbitrary input samples discussed in this paper.  We revised the related work section accordingly.
>
> References:
>
> 3. Simon S. Du, Jason D. Lee, Yuandong Tian, Barnabás Póczos, and Aarti Singh. Gradient descent learns one-hidden-layer cnn: Don’t be afraid of spurious local minima. In International Conference on Machine Learning, 2018.
>
> 4. Mo Zhou, Tianyi Liu, Yan Li, Dachao Lin, Enlu Zhou, and Tuo Zhao. Toward understanding the importance of noise in training neural networks. In International Conference on Machine Learning, 2019.
>
> 5. Alon Brutzkus and Amir Globerson. Globally optimal gradient descent for a convnet with gaussian inputs. In International Conference on Machine Learning, 2017.
>
>
> Q5: Bottom of page 1 and elsewhere: "perturbated" -> "perturbed" ("perturbate" does not sound correct).
>
> A5: Thank you very much for your suggestion. We revised our paper accordingly.
>
> Q6: The paper is very technical and written in a nonintuitive way -- for example, when the authors describe their results and techniques in the introduction of the paper, it reads like a direct translation of some of the proof techniques into natural language rather than as an elucidation of the ideas behind the proof (why such a result is true, how one might arrive at it, etc.). This makes the effort required to penetrate the paper for a nonspecialist unduly high.
>
> A6: Thank you very much for your suggestions. We rewrite some parts of the introduction to highlight our contributions and ideas. Also, all lemmas (regarding the construction of local minima $ \theta $, the construction of $ \theta^{\prime} $ with $ R(\theta^{\prime}) = R(\theta) $, and the perturbation scheme) in the main text have been rewritten in informal ways to describe their ideas intuitively, with a new figure (Fig.1 in new version) to illustrate them. Moreover, we move some formulas and notations in the original main text to appendices.

---

> ### Author Response · Authors · 2022-11-18
> **Response to Reviewer L56V (Part 3 of 4)**
>
> Q7: In general (following comments in the previous section), the preliminaries section 2 is written in an unclear and disorganized style that is overly general and unnecessarily confusing. For example, the definition of the max pooling operation (3) is not clearly made with respect to its interaction with distinct filters $ T_l $ in a layer; the use of a matrix $ W^l $ for all operations does not accurately show the distinct structures of each operation (average pool; convolution; fully connected) as functions of the actual weight parameters of the network and architecture parameters, and hence makes it seem unnecessary to define such intricate notation here (just put it in an appendix instead, if it is not essential to understanding the main body of the paper?);
>
> A7: Thank you very much for your suggestions. We revised our paper accordingly. The preliminary on CNNs is rewritten more accurately. We rewrite the example of matrix $ W^l $ for convolution operations to make it more consistent with subsequent sections. We add a new Appendix A, with examples, to accurately describe the forms of parameter matrices for average pooling and fully connected operations.
>
> In the definition of the max pooling operation (3) (now for the definition of average pooling), if layer $ l $ is a pooling layer, each patch used in the pooling operation include only the neighboring neurons within a single feature map in layer $ (l-1) $. The computation in (3) is for the output of the $ p $th neuron $ o^l(p) $ in a single feature map of layer $ l $, and every feature map performs the same computation. $ T_l $ is for convolutional layers. We revised our paper accordingly.
>
>
> Q8: the discussion of feature map padding is done vaguely in an unhelpful way (one wants to know here what restrictions does this place on the various parameters of the network? In (2), the input feature map dimension is 5 and the output dimension is 6. If I pad the input by 1, now the input dimension is 6 and the output dimension is 8. It does not seem to be possible to pad the input feature map to make the input and output sizes equal!). I would recommend the authors polish the writing and only introduce notation that will be necessary to discuss the main results that are presented in the subsequent sections of the main paper.
>
> A8: Thank you very much for your comments and suggestions. To illustrate the role of zero-padding, we rewrite the example of matrix $ W^l $ for convolution operation in Section 2.2. Given a one-dimensional input $ \\mathbf{o}^{l-1}=(a,b,c,d,e)^T $ and a filter $ \\mathbf{w}_1=(w^1_1,w^2_1,w^3_1)^T$,
>
> the input becomes $ (0,a,b,c,d,e,0)^T $ after padding with zeros. Here, the input dimension is 5, the filter size is $ s_{l-1}=3 $ with $ T_l=1 $, and the patch size is also $ s_{l-1}=3 $. After padding a zero on both sides of original input, there are in total 5 overlapped patches, i.e., $ P_{l-1}=5 $. The output dimension is $ n_l= T_l P_{l-1}=5$. The convolution operation is  equivalent to
>
> \begin{equation*}
> \\mathbf{o}^l = \\sigma \left [  \begin{pmatrix}
> w^2_1 & w^3_1 & 0 & 0 & 0 \\\\
> w^1_1 &  w^2_1 & w^3_1 & 0 & 0  \\\\
> 0 & w^1_1 &  w^2_1 & w^3_1 & 0 \\\\
> 0 & 0 & w^1_1 &  w^2_1 & w^3_1 \\\\
> 0 & 0 & 0 & w^1_1 &  w^2_1
> \end{pmatrix} \begin{pmatrix}
> a\\\\
> b\\\\
> c\\\\
> d\\\\
> e\\\\
> \end{pmatrix}  +
> \begin{pmatrix}
> b^l_1\\\\
> b^l_1\\\\
> b^l_1\\\\
> b^l_1\\\\
> b^l_1
> \end{pmatrix} \right ] .
> \end{equation*}
>
> Thus, the input dimension and output dimension are equal. One can let $ (w^1_1=0,w^2_1=1,w^3_1=0)$ to set the weight matrix to a identity one.
>
> For the case of two-dimensional input with $ 3 \times 3 $ filters, each input feature map is padded by inserting zeros along its four sides.
>
> Q9: The technical presentation of the argument reads as though it is poorly organized, as well. Lemma 1 is a technical result that seems rather abstruse, and it also applies to a very limited setting (only Conv + ReLU layers); the lemma has unnatural technical assumptions on nondegeneracy, and after presenting it the authors then state that "the conclusion in Lemma 1 still holds" for more general settings without any proof or connection to a result in the appendices. Why not present this result informally, with the necessary links to the fully general versions in the appendices made? The second paragraph of section 3.1 says "for ease of presentation", but this seems to be off to me -- the result presented is still very technical, and the simplifications seem to rather have the effect of making it unclear what networks the authors' result actually applies to, rather than the stated intention. The same could be suggested for the following material that leads to Theorem 1, although here the technicality may be essential.

---

> ### Author Response · Authors · 2022-11-18
> **Response to Reviewer L56V (Part 4 of 4)**
>
> A9: Thank you very much for your comments and suggestions. We revised our paper accordingly.
>
> All technical details (regarding the construction of local minima $ \theta $, the construction of $ \theta^{\prime} $ with $ R(\theta^{\prime}) = R(\theta) $, the perturbation scheme, and result in Theorem 2) in the main text have been rewritten in informal and intuitive ways, and a new figure (Fig.1 in new version) is added to illustrate them. The formal versions of these technical details are moved to appendices.
>
> In our new version, we explicitly claim that our construction of spurious local minima is general and applies to CNNs with two consecutive convolutional layers (wherever there are in the CNN), which is easily satisfied by practical CNN models. After presenting our results for the case of having two consecutive convolutional layers on the top, we discuss the general case in Section 3.3.
>
> For the general case when the two consecutive convolutional layers we utilized are not located at the top and there are some convolutional, average pooling, or fully connected layers between them and the output layer, we can still construct spurious local minima using similar idea. The only difference is the setting of parameters for layers above the two consecutive convolutional layers, and we set them such that the output of the two consecutive convolutional layers are propagated unchanged (except pooling operations in possible subsequent pooling layers) to the first fully connected layer, which plays the role of layer $ L $ in Lemmas 1 and 4, and then the output of the first fully connected layer is forwarded unchanged to the output neurons. For such constructed $ \theta $ and $ \theta^{\prime} $, we can show that $ \theta $ is still a local minimum, there is still $ R(\theta^{\prime}) = R(\theta) $ and the empirical risk can be decreased by perturbing $ \theta^{\prime} $. The details are given in Appendix L.
>
> For the assumptions on nondegeneracy, in A3 we have shown that for practical datasets and CNNs, there always exists local minimum $ \theta $ for which Assumption 1 holds. We also remove the full rank requirement in original Lemma 1 and directly enforce the nondegenerate condition: $  \hat{I}^{l, i} \ne 0 $ for all $ l \in [L-1] $ and $ i \in [N] $, which is less restrictive and does not affect our proof. We show in Appendix C that nondegenerate cases always exist since we can fix sufficiently big biases such that all ReLU neurons are turned on and then train the subnetwork.
>
>
> Q10: it does not seem to shed much novel light on when and why spurious minimizers exist in convolutional neural networks trained on various practical tasks. The notation and writing could do with additional polishing.
>
> A10: Thank you very much for your comments and suggestions. We revised our paper accordingly.
>
> Our construction of spurious local minima is general and applies to CNNs with two consecutive convolutional layers, wherever there are in the CNN. We give an example of spurious local minima when discussing Assumption 1 and $ R (\theta) > 0 $, in which the CNN is reduced to a linear classifier. We also add a remark in Section 3.3 to discuss the implications of our results, shown as follows.
>
> Our construction of spurious local minima shows that for practical datasets and CNNs each local minimum of the subnetwork is associated with a spurious local minimum. Since the output of a CNN with ReLU activations is a piece-wise linear function, and from the perspective of fitting data samples with piece-wise linear output [6], the local minima of subnetworks of a CNN (and consequently the spurious local minima of the CNN) may be common due to the abundance of different fitting patterns. Furthermore, as suggested by [7], CNNs have more expressivity than fully connected NNs per parameter in terms of the number of linear regions produced. The ability of producing more linear regions implies more fitting patterns, thus we conjecture that CNNs are more likely to produce spurious local minima than fully connected NNs of the same size. We leave the exploration of these ideas to our future work.
>
> We simplify the notations, especially those appeared in the main text, and polish the writing.
>
> References:
>
> 6. Bo Liu. Spurious local minima are common for deep neural networks with piecewise linear activations. IEEE Transactions on Neural Networks and Learning Systems, pp. 1–13, 2022. doi: 10.1109/TNNLS.2022.3204319.
>
> 7. Huan Xiong, Lei Huang, Mengyang Yu, Li Liu, Fan Zhu, and Ling Shao. On the number of linear regions of convolutional neural networks. In International Conference on Machine Learning, 2020.
>
> Thank you again for your insightful comments and helpful suggestions.

---

> > ### Comment · Reviewer_L56V · 2022-12-06
> > **thanks**
> >
> > Thanks to the authors for the very detailed rebuttal and for numerous revisions to the submission that have improved its clarity. Let me respond to a few specific points here:
> > - [Q3] Thank you for clarifying this. I recommend that the authors restructure this section of the paper to make it clear that the assumption holds in certain practical cases -- i.e. "lead" with the presentation of the main spurious minimizer result in the setting where it holds, and relegate the very precise technical details of when and why (with the assumption A.1) to the appendix. This would improve the clarity of the presentation and allow readers to immediately infer the result and its significance.
> > - [Q8] Thank you for this clarification, this makes sense.
> > I think the revisions have greatly improved the paper, but the amount of change may necessitate an additional round of reviewing. I also feel that a technical venue or journal may be most appropriate for the submission given its highly technical focus.

---

> > > ### Author Response · Authors · 2022-12-07
> > > **Re: thanks**
> > >
> > > Thank you very much for your comments and suggestions. We will restructure Section 3 according to your suggestions in the next version of this paper.

---

### Decision · Program_Chairs · 2023-01-20

**Decision:**

Reject

**Justification For Why Not Higher Score:**

Both the network settings and the technical assumptions significantly limit the scope of the theory. The rebuttal did not fully address these issues.

**Justification For Why Not Lower Score:**

N/A

**Metareview: Summary, Strengths And Weaknesses:**

This paper studies the existence of spurious local minimum in deep CNN. It is shown that a type of CNNs with two consecutive convolutional layers before the output layer have spurious local minima. Reviewers raised two major issues regarding this paper. First, the condition that a subnetwork has a specific type of local minimum may be too strong, or even vacuous. Second, the network structure considered in this paper is substantially different from practical CNNs.
For the first issue, two reviewers mentioned that Assumption 1 and the degeneracy assumption in Lemma 6 (Lemma 1 in the original version) need more justifications. The rebuttal argued that Assumption 1 can be satisfied by using the techniques in the literature and showed that it was possible for practical image data set. The rebuttal also replaced the non-degeneracy condition in Lemma 6 by a weaker version. However, it is still not clear whether this degeneracy condition can be met by the local minimum of the subnetwork. Based my own reading, the authors' claim highly relies on the fact that the subnetwork has a local minimum. It is also implied that the subnetwork can be made rather general as long as it admits local minima satisfying several pre-set conditions. As the reviewers posted questions on the validity of these conditions, the authors should rigorously show under what assumptions on the subnetwork the local minima can meet all these conditions.
For the second issue, one reviewer commented that the framework for constructing local minima mainly follows that of (He et al., 2020), and the additional technical lemmas seem to only work for the specific setting of convolutional layers (with two convolutional layers before the output layer). The revised version claimed that the results could be directly extended to general CNN structures with two consecutive convolutional layers located anywhere (not necessarily the last two layers). After reading revised version, I am not fully convinced by the arguments. If the two convolutional layers are middle layers, the proof of Lemma 6 no longer applies. The perturbation of the two middle layers (under the weight construction of Lemma 6) may yield new activation patterns that the remaining subnetwork cannot achieve.
In summary, both the network settings and the technical assumptions significantly limit the scope of the theory. The rebuttal did not fully address these issues. I think the paper is working on an interesting problem, but due to these reasons, I can only recommend rejection for this version. Hope the authors can address the issues in the future.